# Consistency of Neural Causal Partial Identification

**Jiyuan Tan**
Management Science and Engineering
Stanford University
Stanford, CA 94305
jiyuantan@stanford.edu

**Jose Blanchet**
Management Science and Engineering
Stanford University
Stanford, CA 94305
jose.blanchet@stanford.edu

**Vasilis Syrgkanis**
Management Science and Engineering
Stanford University
Stanford, CA 94305
vsyrgk@stanford.edu

## Abstract

Recent progress in Neural Causal Models (NCMs) showcased how identification and partial identification of causal effects can be automatically carried out via training of neural generative models that respect the constraints encoded in a given causal graph [52, 3]. However, formal consistency of these methods has only been proven for the case of discrete variables or only for linear causal models. In this work, we prove the consistency of partial identification via NCMs in a general setting with both continuous and categorical variables. Further, our results highlight the impact of the design of the underlying neural network architecture in terms of depth and connectivity as well as the importance of applying Lipschitz regularization in the training phase. In particular, we provide a counterexample showing that without Lipschitz regularization this method may not be asymptotically consistent. Our results are enabled by new results on the approximability of Structural Causal Models (SCMs) via neural generative models, together with an analysis of the sample complexity of the resulting architectures and how that translates into an error in the constrained optimization problem that defines the partial identification bounds.

## 1 Introduction

Identifying causal quantities from observational data is an important problem in causal inference which has wide applications in economics [1], social science [20], health care [34, 18], and recommendation systems [9]. One common approach is to transform causal quantities into statistical quantities using the ID algorithm [49] and deploy general purpose methods to estimate the statistical quantity. However, in the presence of unobserved confounding, typically the causal quantity of interest will not be point-identified by observational data, unless special mechanisms are present in the data generating process (e.g. instruments, unconfounded mediators, proxy controls). In the presence of unobserved confounding, the ID algorithm will typically fail to return a statistical quantity and declare the causal quantity as non-identifiable.

One remedy to this problem, which we focus on in this paper, is partial identification, which aims to give informative bounds for causal quantities based on the available data. At a high level, partial identification bounds can be defined as follows: find the maximum and the minimum value that a target causal quantity can take, among all Structural Causal Models (SCMs) that give rise to the same observed data distribution and respect the given causal graph (as well as any other structural

constraints that one is willing to impose). Note that in the presence of unobserved confounding, there will typically exist many structural mechanisms that could give rise to the same observational distribution but have vastly different counterfactual distributions. Hence, partial identification can be formulated as solving a max and a min optimization problem [3]

$$\max_{\mathcal{M} \in \mathcal{C}} \setminus \min_{\mathcal{M} \in \mathcal{C}} \theta(\mathcal{M}), \tag{1}$$
$$\text{subject to } P^{\mathcal{M}}(\boldsymbol{V}) = P^{\mathcal{M}^*}(\boldsymbol{V}), \quad \text{and} \quad \mathcal{G}_{\mathcal{M}} = \mathcal{G}_{\mathcal{M}^*},$$

where $\theta(\mathcal{M})$ is the causal quantity of interest, $\mathcal{M}^*$ is the true model, $\boldsymbol{V}$ is the set of observed nodes, $P^{\mathcal{M}}(\boldsymbol{V})$ is the distribution of $\boldsymbol{V}$ in SCM $\mathcal{M}$, $\mathcal{C}$ is a collection of causal models and $\mathcal{G}_{\mathcal{M}}$ is the causal graph of $\mathcal{M}$ (see Section 2 for formal definitions). Two recent lines of work explore the optimization approach to partial identification. The first line deals with discrete Structure Causal Models (SCMs), where all observed variables are finitely supported. In this case, (1) becomes a Linear Programming (LP) or polynomial programming problem and tight bounds can be obtained [37, 4, 5, 27, 44, 45, 40, 8, 54, 56, 14]. The second line of work focuses on continuous models and explores ways of solving (1) in continuous settings using various techniques [24, 32, 29, 38, 3, 41].

Recently, Xia et al. [51] formalized the connection between SCMs and generative models (see also [33] for an earlier version of a special case of this connection). This work showcased that SCMs can be interpreted as neural generative models, namely Neural Causal Models (NCMs), that follow a particular architecture that respects the constraints encoded in a given causal graph. Hence, counterfactual quantities of SCMs can be learned by optimizing over the parameters of the underlying generative models. However, there could be multiple models that lead to the same observed data distribution, albeit have different counterfactual distributions. Xia et al. [51] first analyze the approximation power of NCMs for discrete SCMs and employ the max/min approach to verify the identifiability of causal quantities, without the need to employ the ID algorithm. Balazadeh et al. [3] and Hui et al. [30], extend the method in [51] and re-purpose it to perform partial identification by solving the min and max problem in the partial identification formulation over neural causal models.

However, for SCMs with general random variables and functional relationships, the approximation error and consistency of this optimization-based approach to partial identification via NCMs has not been established. In particular, two problems remain open. First, given an arbitrary SCM, it is not yet known if we can find an NCM that produces approximately the same intervention distribution as the original one. Although Xia et al.[51] show it is possible to represent any discrete SCM by an NCM, their construction highly relies on the discrete assumption and cannot be directly generalized to the general case. Moreover, Xia et al. [51] use step functions as the activation function in their construction, which may create difficulties in the training process since step functions are discontinuous with a zero gradient almost everywhere.

Second, since we only have access to $n$ samples from the true distribution, $P^{\mathcal{M}}(\boldsymbol{V})$, we need to replace the constraints in (1) with their empirical version that uses the empirical distribution of samples $P_n^{\mathcal{M}^*}(\boldsymbol{V})$ in place of the population distribution and looks for NCMs, whose implied distribution lies within a small distance from the empirical distribution. Moreover, even the NCM distribution is typically only accessible through sampling, hence we will need to generate $m_n$ samples from the NCM and use the empirical distribution of the $m_n$ samples from the NCM in place of the true distribution implied by the NCM. Thus, in practice, we will use a constraint of the form $d(P_n^{\mathcal{M}^*}(\boldsymbol{V}), P_{m_n}^{\mathcal{M}}(\boldsymbol{V})) \leqslant \alpha_n$, where $d$ is some notion of distribution distance and $\alpha_n$ accounts for the sampling error. It is not clear that this approach is consistent, converges to the correct partial identification bounds, when the sample size $n$ grows to infinity. Balazadeh et al. [3] only show the consistency of this approach for linear SCMs. Consistency results concerning more general SCMs is still lacking in the neural causal literature.

In this paper, we establish representation and consistency results for general SCMs. Our contributions are summarized as follows.

- We show that under suitable regularity assumptions, given any Lipschitz SCM, we can approximate it using an NCM such that the Wasserstein distance between any interventional distribution of the NCM and the original SCM is small. Each random variable of the SCM is allowed to be continuous or categorical. We specify two architectures of the Neural Networks (NNs) that can be trained using common gradient-based optimization algorithms (Theorem 2, Theorem 3 and Corollary 1).

- To construct the NCM approximation, we develop a novel representation theorem of probability measures (Proposition 1) that may be of independent interest. Proposition 1 implies that under certain assumptions, probability distributions supported on the unit cube can be simulated by pushing forward a multivariate uniform distribution.

- We discover the importance of Lipschitz regularization by constructing a counterexample where the neural causal approach is not consistent without regularization (Proposition 2).

- Using Lipschitz regularization, we prove the consistency of the neural causal approach (Theorem 4).

**Related Work** There exists a rich literature on partial identification of average treatment effects (ATE) [37, 4, 5, 27, 44, 45, 40, 8, 54, 56, 14, 7, 39, 26]. Balke and Pearl [4, 5] first give an algorithm to calculate bounds on the ATE in the Instrumental Variable (IV) setting. They show that regardless of the exact distribution of the latent variables, it is sufficient to consider discrete latent variables as long as all endogenous variables are finitely supported. Moreover, they discover that (1) is an LP problem with a closed-form solution. This LP-based technique was generalized to several special classes of SCMs [44, 8, 27, 46]. For general discrete SCMs, [14, 56, 55] consider transforming the problem (1) into a polynomial programming problem. Xia et al. [51] discover the connection between generative models and SCMs. They show that NCMs are expressive enough to approximate discrete SCMs. By setting $\mathcal{C}$ in problem (1) to be the collection of NCMs, they apply NCMs for identification and estimation.

For causal models with continuous random variables, the constraints in (1) become integral equations, which makes the problem more difficult. One approach is to discretize the constraints. [24] uses stochastic process representation of the causal model in the continuous IV setting and transforms the problem into a semi-infinite program. [32] relaxes the constraints to finite moment equations and solves the problem by the Augmented Lagrangian Method (ALM). The other approach is to use generative models to approximate the observational distribution and use some metric to measure the distance between distributions. [33] first propose to use GAN to generate images. Later, [29] uses Wasserstein distance in the constraint and transforms the optimization problem into a min and max problem. Similarly, [3] solves the optimization problem using Sinkhorn distance to avoid instability during training. They propose to estimate the Average Treatment Derivative (ATD) and use ATD to obtain a bound on the ATE. They also prove the consistency of the proposed estimator for linear SCMs. [41] uses a linear combination of basis functions to approximate response functions. [26] uses sieve to solve the resulting optimization problem in the IV setting. [19] proposes a neural network framework for sensitivity analysis under unobserved confounding.

**Organization of this paper** In Section 2, we introduce the notations and some basic concepts used throughout the paper. Next, in Section 3, we demonstrate how to construct an NCM so that they can approximate a given SCM arbitrarily well. Two kinds of architecture are given along with an approximation error analysis. In Section 4, we highlight the importance of Lipschitz regularization by giving a counterexample that is not consistent. Then, leveraging the previous approximation results, we are able to prove the consistency of this approach under regularization. Finally, we compare our method with the traditional polynomial programming method empirically in Section 4.1.

## 2 Preliminary

First, we introduce the definition of an SCM. Throughout the paper, we use bold symbols to represent sets of random variables.

**Definition 1.** *(Structural Causal Model) A Structural Causal Model (SCM) is a tuple $\mathcal{M} = (\boldsymbol{V}, \boldsymbol{U}, F, P(\boldsymbol{U}), \mathcal{G}_0)$, where $\boldsymbol{V} = \{V_i\}_{i=1}^{n_V}$ is the set of observed variables; $\boldsymbol{U} = \{U_j\}_{j=1}^{n_U}$ is the set of latent variables; $P(\boldsymbol{U})$ is the distribution of latent variables; $\mathcal{G}_0$ is an acyclic directed graph whose nodes are $\boldsymbol{V}$. The values of each observed variable $V_i$ are generated by*

$$V_i = f_i \left( Pa(V_i), \boldsymbol{U}_{V_i} \right), \quad \text{where } V_i \notin Pa(V_i) \text{ and } \boldsymbol{U}_{V_i} \subset \boldsymbol{U}, \tag{2}$$

*where $F = (f_1, \cdots, f_{n_V})$, $Pa(V)$ is the set of parents of $V$ in graph $\mathcal{G}_0$ and $\boldsymbol{U}_{V_i}$ is the set of latent variables that affect $V_i$. $V_i$ takes either continuous values $\mathbb{R}^{d_i}$ or categorical in $[n_i]$. We extend graph $\mathcal{G}_0$ by adding bi-directed arrows between any $V_i, V_j \in \mathcal{G}_0$ if there exists a correlated latent variable pair $(U_k, U_l), U_k \in \boldsymbol{U}_{V_i}, U_l \in \boldsymbol{U}_{V_j}$. We call the extended graph $\mathcal{G}_{\mathcal{M}}$ the causal graph of $\mathcal{M}$.*

When we write $\text{Pa}(V_i)$, we refer to the parents of $V_i$ in the $\mathcal{G}_0$. To connect with the literature, the causal graph we define is a kind of Acyclic Directed Mixed Graph (ADMG), which is often used to represent SCMs with unobserved variables [49]. Note that we allow one latent variable to enter several nodes, which differs from the common definition. We use $n_U$ and $n_V$ to denote the number of latent variables and observable variables. Let $\boldsymbol{T} \subset \boldsymbol{V}$ be a set of treatment variables. The goal is to estimate causal quantities under a given intervention $\boldsymbol{T} = \boldsymbol{t}$. Formally, the structural equations of the intervened model are

$$T_i = t_i, \quad \forall T_i \in \boldsymbol{T},$$
$$V_i(t) = f_i\left(\text{Pa}(V_i), \boldsymbol{U}_{V_i}\right), \quad \forall V_i \notin \boldsymbol{T}.$$

We denote $V_i(\boldsymbol{t})$ to be the value of $V_i$ under the intervention $\boldsymbol{T} = \boldsymbol{t}$ and $P^{\mathcal{M}}(\boldsymbol{V}(\boldsymbol{t}))$ to be the distribution of $\boldsymbol{V}(\boldsymbol{t})$ in $\mathcal{M}$. The notion of a $C^2$ component [51] is defined as follows.

**Definition 2** ($C^2$-Component). *For a causal graph $\mathcal{G}$, a subset $C \subset \boldsymbol{V}$ is $C^2$-component if each pair $V_i, V_j \in C$ is connected by a bi-directed arrow in $\mathcal{G}$ and $C$ is maximal.*

We provide a concrete example in Appendix A to explain all these notions. We will make the following standard assumption about the independence of latent variables. Note that since we allow latent variables to enter in multiple structural equations, this is more a notational convention and not an actual assumption. Also note that under this convention a bi-directed arrow essentially represents the existence of a common latent parental variable.

**Assumption 1.** *All the latent variables in $\boldsymbol{U}$ are independent.*

To deal with categorical variables, we assume that latent variables that influence categorical variables contain two parts: the shared confounding that influences the propensity functions and the independent noise that generates categorical distributions.

**Assumption 2.** *The set of latent variables consists of two parts $\boldsymbol{U} = \{U_1, \cdots, U_{n_U}\} \cup \{G_{V_i} : V_i \text{ is categorical}\}$. Precisely, if $V_i \in \boldsymbol{V}$ is a categorical variable, the data generation process of $V_i$ satisfies $V_i = \arg\max_{k \in [n_i]} \left\{ g_k^{V_i} + \log\left(f_i\left(Pa(V_i), \boldsymbol{U}_{V_i}\right)\right)_k \right\} \sim Categorical(f_i / \|f_i\|_1)$, where $G_{V_i} = (g_1^{V_i}, \cdots, g_{n_i}^{V_i})$ are i.i.d. standard Gumbel variables, $\boldsymbol{U}_{V_i} \subset \{U_1, \cdots, U_{n_U}\}$.*

This convention is without loss of generality at this point, but will be useful when introducing Lispchitz restrictions on the structural equation functions. The Gumbel variables in the assumption can be replaced by any random variables that can generate categorical variables. It can be proven that all discrete SCMs satisfy this assumption. Note that we implicitly assume that all categorical variables $V_i$ are supported on $[n_i]$ for some $n_i$. It is straightforward to generalize all results to any finite support. Next, we introduce Neural Causal Models (NCMs).

**Definition 3.** *(Neural Causal Model) A Neural Causal Model (NCM) is a special kind of SCM where $\boldsymbol{U} = \{U_1, \cdots, U_{n_U}\} \cup \{G_{V_i} : V_i \text{ categorical}\}$, all $U_i$ are i.i.d. multivariate uniform variables, $G_{V_i}$ are i.i.d Gumbel variables and functions in (2) are Neural Networks (NNs).*

The definition of NCMs we use is slightly different from that in [52] because we need to deal with mixed variables in our models. In (1), we usually take $\mathcal{C}$ to be the set of all SCMs. However, it is difficult to search over all SCMs since (1) becomes an infinite-dimensional polynomial programming problem. As an alternative, we can search over all NCMs. One quantity of common interest in causal inference is the Average Treatment Effect (ATE).

**Definition 4.** *(Average Treatment Effect). For SCM $\mathcal{M}$, the ATE at $\boldsymbol{T} = \boldsymbol{t}$ with respect to $\boldsymbol{T} = \boldsymbol{t}_0$ is given by $ATE_{\mathcal{M}}(\boldsymbol{t}) = \mathbb{E}_{\boldsymbol{u} \sim P(\boldsymbol{U})}[Y(\boldsymbol{t}) - Y(\boldsymbol{t}_0)]$.*

Partial identification can be formulated as estimating the solution to the optimization problems (1) [3]. The max and min values $\overline{F}$ and $\underline{F}$ define the interval $[\underline{F}, \overline{F}]$ which is the smallest interval we can derive from the observed data without additional assumptions. In particular, if $\overline{F} = \underline{F}$, then the causal quantity is point-identified.

**Notations.** We use $\|\cdot\|, \|\cdot\|_\infty$ for the 1-norm and $\infty$-norm and $[n]$ for $\{1, \cdots, n\}$. Bold letters represent sets of random variables. We let $\mathcal{F}_L(K_1, K_2)$ be the class of Lipschitz $L$-continuous functions $f : K_1 \to K_2$. We may omit the domain and use $\mathcal{F}_L$ when the domain is clear from context. Let $\mathcal{H}(K_1, K_2)$ be the set of homeomorphisms from $K_1$ to $K_2$, i.e., injective and continuous

maps in both directions. We define $\epsilon(\mathcal{F}_1, \mathcal{F}_2) = \sup_{f_2 \in \mathcal{F}_2} \inf_{f_1 \in \mathcal{F}_1} \|f_2 - f_1\|$ for function classes $\mathcal{F}_1, \mathcal{F}_2$. We use standard asymptotic notation $O(\cdot), \Omega(\cdot)$. Given a measure $\mu$ on $\mathbb{R}^{d_1}$ and a measurable function $f : \mathbb{R}^{d_1} \to \mathbb{R}^{d_2}$, the push-forward measure $f_{\#}\mu$ is defined as $f_{\#}\mu(B) = \mu\left(f^{-1}(B)\right)$ for all measurable sets $B$. We use $P(X)$ to represent the distribution of random variable $X$. Let $\Delta_n = \left\{(p_1, \cdots, p_n) : \sum_{i=1}^{n} p_i = 1, p_i \geqslant 0\right\}$ be the probability simplex. We use Categorical$(\boldsymbol{p}), \boldsymbol{p} \in \Delta_n$ to represent categorical distribution with event probability $\boldsymbol{p}$. We let $W(\cdot, \cdot)$ be the Wasserstein-1 distance and $S_\lambda(\mu, \nu)$ be the Sinkhorn distance [12] with regularization parameter $\lambda > 0$.

# 3   Approximation Error of Neural Causal Models

In this section, we study the expressive power of NCMs, which serves as a key ingredient in proving the consistency result. In particular, given an SCM $\mathcal{M}^*$, we want to construct an NCM $\hat{\mathcal{M}}$ such that the two causal models produce similar interventional results. Unlike in the discrete case [4, 51], latent distributions can be extremely complicated in general cases. The main challenge is how to design the structure of NCMs to ensure strong approximation power.

In the following, we first derive an upper bound on the Wasserstein distance between two causal models sharing the same causal graph. Using this result, we decompose the approximation error into two parts: the error caused by approximating structural functions via neural networks and the error of approximating the latent distributions. Then, we design different architectures for these two parts.

**Decomposing the Approximation Error**   First, we present a canonical representation of an SCM, which essentially states that we only need to consider the case where each latent variable $U_i$ corresponds to a $C^2$ component of $\mathcal{G}$.

**Definition 5** (Canonical representation). *A SCM $\mathcal{M}$ with causal graph $\mathcal{G}$ has canonical form if*

1. *The set of latent variables consists of two sets,*

$$\boldsymbol{U} = \left\{U_C : C \text{ is a } C^2\text{-component of } \mathcal{G}\right\} \cup \left\{G_{V_i} = (g_1^{V_i}, \cdots, g_{n_i}^{V_i}) : V_i \text{ is categorical}\right\},$$

   *where $U_C$ and $g_j^{V_i}$ are independent and $g_j^{V_i}$ are standard Gumbel variables.*

2. *The structure equations have the form*

$$V_i = \begin{cases} f_i\left(Pa(V_i), \boldsymbol{U}_{V_i}\right), & V_i \text{ is continuous,} \\ \arg\max_{k \in [n_i]}\left\{g_k^{V_i} + \log\left(f_i\left(Pa(V_i), \boldsymbol{U}_{V_i}\right)\right)_k\right\}, & V_i \text{ is categorical, } \|f_i\|_1 = 1, \end{cases}$$

$$(3)$$

   *where $\boldsymbol{U}_{V_i} = \{U_C : V_i \in C, C \text{ is a } C^2\text{-component of } \mathcal{G}\}$ and $(x)_k$ is the $k$-th coordinate of the vector $x$. We further assume that $f_i$ are normalized for categorical variables.*

*Given a function class $\mathcal{F}$, the SCM class $\mathcal{M}(\mathcal{G}, \mathcal{F}, \boldsymbol{U})$ consists of all canonical SCM models with causal graph $\mathcal{G}$ such that $f_i \in \mathcal{F}, i \in [n_V]$.*

Proposition 4 in the appendix shows that any SCM satisfying Assumption 1,2 can be represented in this way and we provide an example in Appendix B.1 to illustrate how to obtain the canonical representation for a given SCM. Therefore, we restrict our attention to the class $\mathcal{M}(\mathcal{G}, \mathcal{F}, \boldsymbol{U})$. For two SCM classes $\mathcal{M}(\mathcal{G}, \mathcal{F}, \boldsymbol{U}), \mathcal{M}(\mathcal{G}, \hat{\mathcal{F}}, \hat{\boldsymbol{U}})$, we want to study how well we can represent the models in the first class by the second class. The Wasserstein distance between the intervention distributions is used to measure the quality of the approximation. To approximate the functions in the structural equations, we need to make the following regularity assumptions on the functions.

**Assumption 3.** *If $V_i$ is continuous, $f_i$ in (3) are $L_f$-Lipschitz continuous. If $V_i$ is categorical, the propensity functions $f_i(Pa(V_i), \boldsymbol{U}_{V_i}) \triangleq \mathbb{P}(V_i = j | Pa(V_i), \boldsymbol{U}_{V_i}), j \in [n_i]$ are $L_f$-Lipschitz continuous. There exists a constant $K > 0$ such that $\max_{i \in [n_V], j \in [n_U]}\{\|V_i\|_\infty, \|U_j\|_\infty\} \leqslant K$.*

The following theorem summarizes our approximation error decomposition.

**Theorem 1.** *Given any SCM model $\mathcal{M} \in \mathcal{M}(\mathcal{G}, \mathcal{F}_L, \boldsymbol{U})$, let the treatment variable set be $\boldsymbol{T} = \{T_k\}_{k=1}^{n_t}$ and suppose that Assumption 1, 2 and 3 hold for $\mathcal{M}$ with Lipschitz constant $L$, constant $K$.*

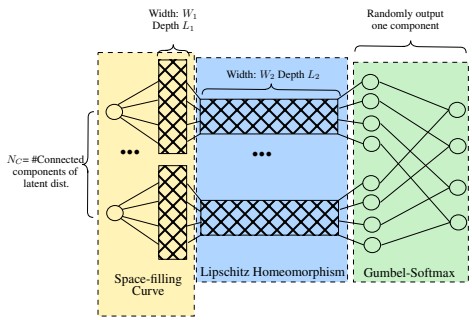
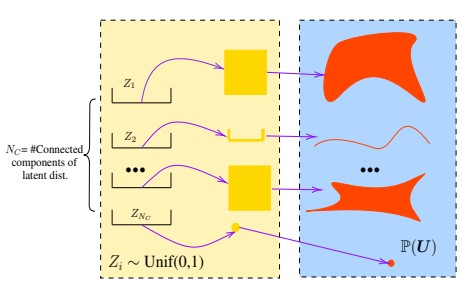

(a) Architecture of wide neural network for $4-$dimensional output. The first (yellow) part approximates the distribution on different connected components of the support using the results from [43]. The width and depth of each block in this part are $W_1$ and $L_1$. The second (blue) part transforms the distributions on the unit cube to the distributions on the support. The width and depth of each block in the blue part are $W_2$ and $L_2$. The third (green) part is the Gumbel-Softmax layer. It combines the distributions on different connected components of the support together and outputs the final distribution.

(b) This figure demonstrates the first two parts of our architecture. Each interval in the yellow box corresponds to one coordinate of input in the left figure. We first push forward uniform distributions to different cubes. Then, using Assumption 4, we adapt the shape of the support and push the measure from unit cubes to the original support of $P(\boldsymbol{U})$. In this way, we can approximate complicated measures by pushing forward uniform variables.

*For any intervention $\boldsymbol{T} = \boldsymbol{t}$ and $\hat{\mathcal{M}} \in \mathcal{M}(\mathcal{G}, \hat{\mathcal{F}}, \hat{\boldsymbol{U}})$, we have*

$$W\left(P^{\mathcal{M}}(\boldsymbol{V}(\boldsymbol{t})), P^{\hat{\mathcal{M}}}(\hat{\boldsymbol{V}}(\boldsymbol{t}))\right) \leqslant C_{\mathcal{G}}(L, K)\left(\sum_{i=1}^{n_V} \|f_i - \hat{f}_i\|_\infty + W\left(P^{\mathcal{M}}(\boldsymbol{U}), P^{\hat{\mathcal{M}}}(\hat{\boldsymbol{U}})\right)\right), \quad (4)$$

*where $C_{\mathcal{G}}(L, K)$ is a constant that only depends on $L, K$ and the causal graph $\mathcal{G}$ and $f_i, \hat{f}_i$ are structural functions of $\mathcal{M}$ and $\hat{\mathcal{M}}$ respectively.*

Theorem 1 separates the approximation error into two parts, which motivates us to construct the NCM in the following way. First, we approximate the functions $f_i$ in (3) by NNs $\hat{f}_i$. Then, we approximate the distribution of latent variables by pushing forward uniform and Gumbel variables using neural networks, i.e., $\hat{U}_{C_j} = \hat{g}_j(Z_{C_j})$, where $\{C_j\}$ are $C^2$ components and $Z_{C_j}$ are multi-variate uniform and Gumbel random variables. The structural equations of the resulting approximated model $\hat{\mathcal{M}}$ are

$$\hat{V}_i = \begin{cases} \hat{f}_i\left(\mathrm{Pa}(\hat{V}_i), (\hat{g}_j(Z_{C_j}))_{U_{C_j} \in \boldsymbol{U}_{V_i}}\right), & V_i \text{ is continuous,} \\ \arg\max_{k \in [n_i]}\left\{g_k + \log\left(\hat{f}_i\left(\mathrm{Pa}(\hat{V}_i), (\hat{g}_j(Z_{C_j}))_{U_{C_j} \in \boldsymbol{U}_{V_i}}\right)_k\right)\right\}, & V_i \text{ is categorical,} \end{cases} \quad (5)$$

wheres $N_{C,j}$ are constants to be specified later.

For the first part, we need to approximate Lipschitz continuous functions. For simplicity, we assume that the domain of the functions are uniform cubes. Similar arguments hold for any bounded cubes. We denote $\mathcal{NN}_{k_1,k_2}(W, L)$ to be the set of ReLu NNs with input dimension $k_1$, output dimension $k_2$, width $W$ and depth $L$. It has been shown that $\epsilon(\mathcal{NN}_{k_1,1}(2d_1 + 10, L_0), \mathcal{F}_L([0,1]^{d_1}, \mathbb{R})) \leqslant O(L_0^{-2/k_1})$ [53], where $\epsilon(\cdot, \cdot)$ denotes the approximation error defined in Section 2. For a vector valued function, we can use a wider NN to approximate each coordinate and get a similar rate.

For the second part, we approximate each $U_i$ individually by pushing forward i.i.d. multivariate uniform and Gumbel variables $\hat{U}_{C_i} = \hat{g}_i(Z_{C_i})$ since the latent variables are independent by Assumption 1. To do so, we examine under what assumptions on the measure $\mathbb{P}$ over $\mathbb{R}^n$ we can find a NN $\hat{g}$ such that $W(\hat{g}_\# \lambda, \mathbb{P})$ is small, where $\lambda(\cdot)$ is some reference measure.

## 3.1 Approximating Mixed Distributions by Wide Neural Networks

In this subsection, we will extend the results in [43] to construct a wide neural network as the push-forward map. It turns out that to get a good approximation of the targeted distribution, the shape of the support is essential.

**Assumption 4** (Mixed Distribution). *The support of measure $\mathbb{P}$ has finite connected components $C_1, C_2 \cdots C_{N_C}$, i.e., $supp(\mathbb{P}) = \bigcup_{i=1}^{N_C} C_i$, and each component $C_i$ satisfies $\mathcal{H}([0,1]^{d_i^C}, C_i) \cap \mathcal{F}_L([0,1]^{d_i^C}, C_i) \neq \emptyset$ for some $d_i^C \geqslant 0$. Recall that $\mathcal{H}(K_1, K_2)$ is the set of homeomorphisms from $K_1$ to $K_2$ defined at the end of Section 2.*

Assumption 4 encompasses almost all natural distributions. For example, distributions supported on $[0,1]^d$ and closed balls, finitely supported distributions and mixtures of them all satisfy this assumption. Assumption 4 allows us to transform the support of the targeted distribution into unit cubes and the nice geometric properties of unit cubes facilitate our construction.

Now, we briefly explain the construction of the push-forward maps. An example is provided in Appendix B.1 to illustrate the construction. The NN architecture consists of three parts (see Figure 1a). The input dimension is the same as the number of connected components of the support $N_C$. For each component $C_i$, let $H_i \in \mathcal{H}([0,1]^{d_i^C}, C_i) \cap \mathcal{F}_L([0,1]^{d_i^C}, C_i)$, where $d_i^C$ is the dimension of component $C_i$ in Assumption 4. By $\mathbb{P} = (H_i)_\# (H_i^{-1})_\# \mathbb{P}$ on $C_i$, we can approximate $(H_i^{-1})_\# \mathbb{P}$ first, which is supported on a unit cube. [43] constructs a wide NN $\hat{g}$ of width $W$ and constant depth such that $W(\hat{g}_\# \lambda, (H_i)_\#^{-1} \mathbb{P}) \leqslant O(W^{-1/d_i^C})$ where $\lambda$ is the uniform measure on $[0,1]$. Then, we approximate the Lipschitz map $H_i$ to $C_i$ to pull the distribution back to $C_i$. These are the first two parts (yellow and blue blocks in Figure 1a) of the architecture.

Suppose that the output of $i$-th coordinate in the first two parts is $\vec{v}_i$, the Gumbel-softmax layer in the third part (green box) combines different components of the support. In particular, we want to output $\vec{v}_i$ with probability $p_i = \mathbb{P}(C_i)$. Let $V = [\vec{v}_1, \cdots, \vec{v}_i]$, this can be achieved by outputting $VX$, where $X = (X_1, \cdots, X_{N_C})^\mathsf{T}$ is a one-hot random vector with $\mathbb{P}(X_i = 1) = p_i$. To use backpropagation in training, we use the Gumbel-Softmax trick [31] to (approximately) simulate such a random vector, $\hat{X}_i^\tau = \frac{\exp((\log p_i + G_i)/\tau)}{\sum_{k=1}^{N_C} \exp((\log p_k + G_k)/\tau)}$, where $\tau > 0$ is the temperature parameter (a hyperparameter) and $G_i \sim \text{Gumbel}(0,1)$ are i.i.d. standard Gumbel variables. As $\tau \to 0$, the distribution of $\hat{X}^\tau$ converges almost surely to the categorical distribution [36, Proposition 1]. In particular, when $\tau = 0$, we denote $\hat{X}_i^0 = \mathbb{I}_{i = \arg \max_j \{\log p_i + G_i\}}$. The output of the last layer is $V\hat{X}^\tau$. Note that the Gumbel-softmax function is differentiable with respect to parameter $\{\log(p_i)\}_{i=1,\cdots,N_C}$. Therefore, we can train the network with common gradient-based algorithms. Putting things together, we obtain the following theorem.

**Theorem 2.** *Given any probability measure $\mathbb{P}$ on $\mathbb{R}^d$ that satisfies Assumption 4, let $\lambda$ be the Lebesgue measure on $[0,1]^{N_C}$, where $N_C$ is defined in Assumption 4. There exists a neural network $\hat{g} = \hat{g}_3^\tau \circ \hat{g}_2 \circ \hat{g}_1$ with the above architectsure (Figure 1a) such that*

$$W(\hat{g}_\# \lambda, \mathbb{P}) \leqslant O\left(W_1^{-1/\max_i \{d_i^C\}} + L_2^{-2/\max_i \{d_i^C\}} + (\tau - \tau \log \tau)\right).$$

*Here, $\hat{g}_i, i = 1, 2$ has the separable form $\left(\hat{g}_i^1(x_1), \cdots, \hat{g}_i^{N_C}(x_{N_C})\right)$ and $\hat{g}_1^j \in \mathcal{NN}_{1,d_j^C}(W_1, \Theta(d_j^C))$, $\hat{g}_2^j \in \mathcal{NN}_{d_j^C, d}(\Theta(d \cdot d_j^C), L_2)$, $j \in [N_C]$. $\{d_j^C\}$ are the dimension of cubes in Assumption 4. $\hat{g}_3^\tau$ is the Gumbel-Softmax layer with temperature parameter $\tau > 0$.*

Note that $\hat{g}_3^\tau$ (the Gumbel-softmax layer) is actually a random function since the coefficient vector $\hat{X}^\tau$ is a random variable. In this sense, $\hat{g}_\# \lambda$ can be viewed as pushing forward uniform and Gumbel variables using a neural net.

### 3.2 Approximating Mixed Distributions by Deep Neural Networks

In this subsection, we will show that under one additional assumption on the distribution, deep ReLu networks have a stronger approximation power in approximating distributions, which means we can use fewer computational units to achieve the same worst-case theoretical approximation error.

**Assumption 5** (Lower Bound). *Suppose that $\mathbb{P}$ is supported on a compact set $K \subset \mathbb{R}^D$, there exists a constant $C_f > 0$, $f \in \mathcal{H}([0,1]^d, K) \cap \mathcal{F}_L([0,1]^d, K)$, such that for any measurable set $B \subset [0,1]^d$, $\mathbb{P}(f(B)) \geqslant C_f \lambda(B)$. Besides, if $d > 0$, $f_\# \mathbb{P}$ vanishes on the boundary $\partial[0,1]^d$.*

Assumption 5 implies that $d\lambda/d(f_\#^{-1}\mathbb{P})$ exists and is lowered bounded by a constant $C_f$. The next proposition extends the Skorohod representation theorem [48]. It shows that under Assumption 5, it

is possible to simulate any distribution on the unit cubes with Hölder continuous curves and uniform distribution on $[0, 1]$.

**Proposition 1.** *Let $\lambda$ be the Lebesgue measure on $[0, 1]$. Given any probability measure $\mathbb{P}$ that satisfies Assumption 5, there exists a continuous curve $\gamma : [0, 1] \to supp(\mathbb{P})$ such that $\gamma_\# \lambda = \mathbb{P}$. Furthermore, if $d \geqslant 1$, $\gamma$ is $1/d$-Hölder continuous.*

Results from [53] show that we can approximate any Hölder continuous $d$-dimensional function with index $\alpha$ by a deep ReLu network with depth $L$ and error $O(L^{-2\alpha/d})$. Leveraging this result, we can replace the first part of the architecture in the previous subsection with deep ReLu networks (See Figure 8 in the appendix). The remaining two parts are the same as the construction in Figure 1a. The following theorem gives a sharper bound on the approximation error compared with Theorem 2.

**Theorem 3.** *Under the Assumption 4, if in addition, $\mathbb{P}$ constrained to each component satisfies Assumption 5, there exists a neural network $\hat{g} = \hat{g}_3^\tau \circ \hat{g}_2 \circ \hat{g}_1$ with the above architecture such that*

$$W(\hat{g}_\# \lambda, \mathbb{P}) \leqslant O\left(L_1^{-2/\max_i\{d_i^C\}} + L_2^{-2/\max_i\{d_i^C\}} + (\tau - \tau \log \tau)\right),$$

*where $d_i^C$ are the dimensions of the connected components in Assumption 4, $\hat{g}_i, i = 1, 2$ has the form $\left(\hat{g}_i^1(x_1), \cdots, \hat{g}_i^{N_C}(x_{N_C})\right)$ and $\hat{g}_1^j \in \mathcal{NN}_{1,d_j^C}(\Theta(d_j^C), L_1)$. $\hat{g}_2^j, \hat{g}_3^\tau, \tau$ are the same as in Theorem 2.*

Let $N$ be the number of nonzero weights in a neural network, Theorem 3 shows that a deep NN can achieve $O(N^{-2/d})$ error while by Theorem 2 a wide network can only achieve $O(N^{-1/d})$ error. Now, we can put things together to construct NCM approximations. For simplicity, we omit the input and output dimensions of the neural network. As we mention in previous sections, our construction (5) consists of two parts, $\hat{f}_i$ approximating the structural functions $f_i$ in (3), and $\hat{g}_j(Z_j) = \hat{g}_{3,j}^\tau \circ \hat{g}_{2,j} \circ \hat{g}_{1,j}(Z_j)$ approximating the latent variables $U_j$. Let $\text{NCM}_\mathcal{G}(\mathcal{F}_0, \mathcal{F}_1, \mathcal{F}_2, \tau)$ be a collection of NCMs with structural equation (5) and

$$\hat{f}_i \in \mathcal{F}_0, \hat{g}_{1,j} = (\hat{g}_{1,j}^1, \cdots, \hat{g}_{1,j}^{N_{C,j}}), \hat{g}_{1,j}^i \in \mathcal{F}_1, \hat{g}_{2,j} = (\hat{g}_{2,j}^1, \cdots, \hat{g}_{2,j}^{N_{C,j}}), \hat{g}_{2,j}^i \in \mathcal{F}_2,$$

where $N_{C,j}$ is the number of connected components for $U_j$ and $\mathcal{F}_i$ are function classes.

**Corollary 1.** *Given a causal model $\mathcal{M}^* \in \mathcal{M}(\mathcal{G}, \mathcal{F}_L, \boldsymbol{U})$, suppose that Assumptions 1-3 hold and the distributions of $U_C$ for all $C^2$-component satisfy the assumptions of Theorem 3. Let $d_{\max}^{in}$ and $d_{\max}^{out}$ be the largest input and output dimension of $f_i$ in (3) and $d_{\max}^U$ be the largest dimension of all latent variables. There exists a neural causal model*

$$\hat{\mathcal{M}} \in NCM_\mathcal{G}(\mathcal{NN}(\Theta(d_{\max}^{in} d_{\max}^{out}), L_0), \mathcal{NN}\left(\Theta\left(d_{\max}^U\right), L_1\right), \mathcal{NN}\left(\Theta\left((d_{\max}^U)^2\right), L_2\right), \tau)$$

*with structure equations (5). For any intervention $\boldsymbol{T} = \boldsymbol{t}$, $\hat{\mathcal{M}}$ satisfies*

$$W\left(P^{\mathcal{M}^*}(\boldsymbol{V}(\boldsymbol{t})), P^{\hat{\mathcal{M}}}(\boldsymbol{V}(\boldsymbol{t}))\right) \leqslant O(L_0^{-2/d_{\max}^{in}} + L_1^{-2/d_{\max}^U} + L_2^{-2/d_{\max}^U} + (\tau - \tau \log \tau)).$$

Similar approximation results also hold for wide NN approximation, as presented in Section 3.1. The proof can be easily adapted to the wide NNs architecture.

## 4 Consistency of Neural Causal Partial Identification

In this section, we prove the consistency of the max/min optimization approach to partial identification. In the finite sample setting, we consider the following estimator $F_n$ of the optimal values of (1).

$$F_n = \underset{\hat{\mathcal{M}} \in \text{NCM}_\mathcal{G}(\mathcal{F}_{0,n}, \mathcal{F}_{1,n}, \mathcal{F}_{2,n}, \tau_n)}{\arg\min} \mathbb{E}_{t \sim \mu_T} \mathbb{E}_{\hat{\mathcal{M}}}[F(V_1(t), \cdots, V_{n_V}(t))], \tag{6}$$

$$s.t. \quad S_{\lambda_n}(P_{m_n}^{\hat{\mathcal{M}}}(\boldsymbol{V}), P_n^{\mathcal{M}^*}(\boldsymbol{V})) \leqslant \alpha_n,$$

where $P_n^{\mathcal{M}^*}, P_{m_n}^{\hat{\mathcal{M}}}$ are the empirical distribution of $P^{\mathcal{M}^*}, P^{\hat{\mathcal{M}}}$ with sample size $n, m_n, \mu_T$ is some given measure and $\mathcal{F}_{i,n}$ will be specified later. For example, the counterfactual outcome $\mathbb{E}[Y(1)]$ is a special case of the objective. Our results can be easily generalized to any linear combination of objective functions of this form. We use the Sinkhorn distance because it can be computed efficiently in practice [15]. We want to study if $F_n$ gives a useful lower bound as $n \to \infty$.

To match the observational distribution, we need to increase the width or depth of the NNs we use. As the sample size increases, the number of parameters also increases to infinity, which creates difficulty in the analysis. To obtain consistency, we need to regularize the functions while preserving their approximation power. Surprisingly, if we do not use any regularization, the following proposition implies that consistency may not hold even if the SCM is identifiable.

**Proposition 2** (Informal, see Proposition 5 for a formal version)**.** *There exists a constant $c > 0$ and an identifiable SCM $\mathcal{M}^*$ satisfying Assumptions 1-5 such that for any $\epsilon > 0$, there exists an SCM $\mathcal{M}_\epsilon$ satisfying $W(P^{\mathcal{M}^*}(\boldsymbol{V}), P^{\mathcal{M}_\epsilon}(\boldsymbol{V})) \leq \epsilon$ and $|ATE_{\mathcal{M}^*} - ATE_{\mathcal{M}_\epsilon}| > c$.*

Here, $\mathcal{M}^*$ is the ground-truth model and $\mathcal{M}_\epsilon$ are the models we use to approximate $\mathcal{M}^*$. Proposition 2 implies that we need some regularization on $\mathcal{M}_\epsilon$. Otherwise, even if the observation distributions are close, their ATEs can be far away. In particular, we may want to regularize the Lipschitz constant of the NN. Much work has been done to impose Lipschitz regularization during the training process [13, 50, 22, 42, 10]. We denote $\mathrm{Lip}(f)$ to be the Lipschitz constant of a function $f$ and define the truncated Lipschitz NN class,

$$\mathcal{NN}_{d_1,d_2}^{L_f,K}(W,L) = \{\max\{-K, \min\{f, K\}\} : f \in \mathcal{NN}_{d_1,d_2}(W,L), \mathrm{Lip}(f) \leqslant L_f\}.$$

For simplicity, we omit the dimensions and use shorthand $\mathcal{NN}^{L_f,K}(W,L)$. The next theorem gives the consistency result of the min estimator. To state the theorem, we define $F_* = \mathbb{E}_{t \sim \mu_T} \mathbb{E}_{\mathcal{M}^*}[F(V_1(t), \cdots, V_{n_V}(t))]$ to be the true value and $\underline{F}^L$ to be the optimal value of the following optimization problem over SCMs with Lipschitz constant $L$.

$$\underline{F}^L = \underset{\hat{\mathcal{M}} \in \mathcal{M}(\mathcal{G}, \mathcal{F}_L^K, \boldsymbol{U}), P(\boldsymbol{U})}{\arg\min} \mathbb{E}_{t \sim \mu_T} \mathbb{E}_{\hat{\mathcal{M}}}[F(V_1(t), \cdots, V_{n_V}(t))], \tag{7}$$

$$s.t. W(P^{\hat{\mathcal{M}}}(\boldsymbol{V}), P^{\mathcal{M}^*}(\boldsymbol{V})) = 0,$$

where the minimum is taken over $\mathcal{M}(\mathcal{G}, \mathcal{F}_L^K, \boldsymbol{U})$ with $\mathcal{F}_L^K = \{f : \|f\| \leqslant K, f \in \mathcal{F}_L\}$ and all latent distributions $P(\boldsymbol{U})$. Note that if $L$ is the Lipschitz bound $L_f$ that we assume on our structural functions, then $\underline{F}^L$ is the sharp lower bound.

**Theorem 4.** *Let $\mathcal{M}^*$ be any SCM satisfying the assumptions of Corollary 1. Suppose that the Lipschitz constant of functions in $\mathcal{M}$ is $L_f$, $F : \mathbb{R}^{n_V} \to \mathbb{R}$ in (6) is Lipschitz continuous and $\tau_n > 0$, let $K > 0$ be the constant in Assumption 3, $\hat{L}_f = \sqrt{d_{\max}^{in} d_{\max}^{out}} L_f$,*

$$\mathcal{F}_{0,n} = \mathcal{NN}^{\hat{L}_f, K}\left(W_{0,n}, \Theta\left(\log d_{\max}^{in}\right)\right), \mathcal{F}_{1,n} = \mathcal{NN}^{\infty,\infty}(\Theta(d_{\max}^U), L_{i,n}),$$
$$\mathcal{F}_{2,n} = \mathcal{NN}^{\infty, K}(\Theta((d_{\max}^U)^2), L_{i,n}),$$

*where $d_{\max}^U, d_{\max}^{in}$ and $d_{\max}^{out}$ are defined in Corollary 1, take the radius to be $\alpha_n = \epsilon_n + s_n, \epsilon_n = O(W_{0,n}^{-1/d_{\max}^{in}} + \sum_{i=1}^2 L_{i,n}^{-2/d_{\max}^U} + \tau_n \log \tau_n)$,*

$$s_n = O(m_n^{-1/(d_{\max}^U + 2)} \log m_n + \delta_n + \log(nm_n)\lambda_n), \quad \delta_n = O(n^{-1/\max\{2, d_{\max}^U\}} \log^2(n)).$$

*If $m_n = \Omega(n), L_{i,n} = \Theta(m_n^{d_{\max}^U/(2d_{\max}^U + 4)}), i = 1, 2, \lim_{n \to \infty} \min\{\tau_n^{-1}, W_{0,n}, (\log(nm_n)\lambda_n)^{-1}\} = \infty$, then with probability 1, $[\liminf_{n \to \infty} F_n, \limsup_{n \to \infty} F_n] \subset [\underline{F}^{\hat{L}_f}, F_*]$.*

Note that our theorem shows that the lower limit of the solution is large than the point $\underline{F}^{\hat{L}_f}$, where $\hat{L}_f$ is slightly larger than the original constraint $L_f$ we impose on the structural functions. Hence, this point can potentially be slightly smaller than the sharp bound $\underline{F}^{L_f}$. This worsening is due to the fact that we need to use NNs that satisfy a slightly worse Lipschitz property, to ensure that we have sufficient approximation power. Although Theorem 4 does not guarantee $\{F_n\}$ converges to a point, it states that $\{F_n\}$ may oscillate in the interval $[\underline{F}^{\hat{L}_f}, F_*]$, which will still give a useful lower bound to ground-truth value $F_*$. In particular, if the graph is identifiable, we have $F^* = \underline{F}^{\hat{L}_f}$ and $\{F_n\}$ converges to $F^*$. Also, note that in Theorem 4, we use wide NNs rather than deep NN for $\mathcal{F}_{0,n}$ because results in [53] show that wide NNs can approximate Lipschitz functions while controlling their Lipschitz constants (a property that is not yet established for deep NNs). Similar results can be obtained if wide neural nets are used for all components, invoking Theorem 2.

As a special case, we leverage Theorem 4 for a non-asymptotic rate for the ATE without confounding.

**Proposition 3** (Hölder continuity of ATE). *Given two causal models $\mathcal{M}, \hat{\mathcal{M}} \in \mathcal{M}(\mathcal{G}, \mathcal{F}_L, \boldsymbol{U})$ satisfying Assumption 1 and Assumption 3, let their observational distributions be $\nu, \mu$. Suppose the norms of all variables are bounded by $K > 0$. If (1) (Overlap) $\nu$ is absolutely continuous with respect to one probability measure $P$ and the density $p_\nu (t|Pa(T) = x) \geqslant \delta > 0$ for $x$ almost surely and $t \in [t_1, t_2]$ and (2) (No Confounding) there is no confounding in the causal graph $\mathcal{G}$, we have*

$$\int_{t_1}^{t_2} (\mathbb{E}_{\mathcal{M}}[Y(t)] - \mathbb{E}_{\hat{\mathcal{M}}}[\hat{Y}(t)])^2 P(dt) \leqslant \frac{2C_W}{\delta} W(\mu, \nu) + 2(L+1)^{n_V} W^2(\mu, \nu)(t_2 - t_1),$$

*where $C_W = 4(L+1)^{n_V} K + 2K \max \{(L+1)^{n_V}, 1\}$.*

**Corollary 2.** *Let $F, \mu_T$ in Theorem 4 to be $F(\boldsymbol{V}) = Y, \mu_T \sim Unif([t_1, t_2]), \epsilon > 0$. Suppose that the assumptions in Proposition 3 and Theorem 4 hold, with probability at least $1 - O(n^{-2})$, we have $|F_n - F_*| \leqslant O(\sqrt{\alpha_n})$, where $F_n, F_*, \alpha_n$ are defined in Theorem 4.*

### 4.1 Experiments

In this section, we examine the performance of our algorithm in two settings. We compare our algorithm with the Autobounds algorithm [14] in a binary IV example in [14] and in a continuous IV model [1]. Since Autobounds can only deal with discrete models, we discretize the continuous model for comparison purposes. The implementation details are provided in the Appendix D. The setting of the first experiment is taken from [14, Section D.1]. This is a binary IV problem and we can calculate the optimal bound using LP [4]. We find that our bound is close to the optimal bound. The second experiment is a general IV example where the treatment is binary but the rest of the variables are continuous. The program that Autobounds solves after discretization contains $\approx 2^{14}$ variables. Even with such a fine discretization, the bound obtained by Autobounds is not tighter than our NCM approach. The details of the structural equations and analysis can be found in Appendix D. We also provide an extra experiment on the counterexample of Proposition 5 in the appendix to show the effect of Lipschitz regularization.

| Setting | Algorithm | Average Bound | Optimal Bound | True Value |
|---------|-----------|---------------|---------------|------------|
| Binary IV | NCM (Ours) | [-0.49,0.05] | [-0.45, -0.04] | -0.31 |
| | Autobounds | [-0.45,-0.05] | [-0.45, -0.04] | |
| General IV | NCM (Ours) | [2.49,3.24] | – | 3 |
| | Autobounds | [1.40, 3.48] | – | |

Table 1: Experiment results of 2 IV settings. The sample sizes are taken to be 5000 in each experiment. STD is the standard derivation. The experiments are repeated 10 times for binary IV and 50 times for continuous IV. In all experiments, the bounds given by both algorithms all cover the true values.

**Conclusion**   In this paper, we provide theoretical justification for using NCMs for partial identification. We show that NCMs can be used to represent SCMs with complex unknown latent distributions under mild assumptions and prove the asymptotic consistency of the max/min estimator for partial identification of causal effects in general settings with both discrete and continuous variables. Our results also provide guidelines on the practical implementation of this method and on what hyperparameters are important, as well as recommendations on values that these hyperparameters should take for the consistency of the method. These practical guidelines were validated with a small set of targeted experiments, which also showcase superior performance of the neural-causal approach as compared to a prior main contender approach from econometrics and statistics, that involves discretization and polynomial programming.

An obvious next step in the theoretical foundation of neural-causal models is providing finite sample guarantees for this method, which requires substantial further theoretical developments in the understanding of the geometry of the optimization program that defines the bounds on the causal effect of interest. We take a first step in that direction for the special case, when there are no unobserved confounders and view the general case as an exciting avenue for future work.

**Acknowledgement**   Vasilis Syrgkanis is supported by NSF Award IIS-2337916 and a 2022 Amazon Research Award.

---

[1]The code can be found in https://github.com/Jiyuan-Tan/NeuralPartialID

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

# A   Illustration of Notions in Section 2

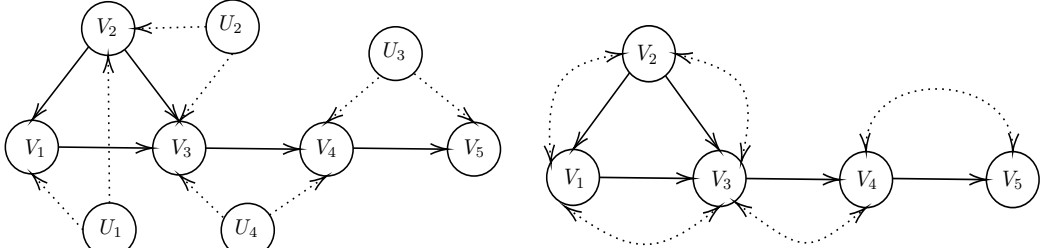

Figure 2: An SCM example.       Figure 3: The causal graph of this SCM.

To further explain the notions in Section 2, we consider the following example. Let $\mathcal{M}$ be an SCM with the following structure equations.

$$
\begin{aligned}
V_1 &= f_1(V_2, U_1), \\
V_2 &= f_2(U_1, U_2), \\
V_3 &= f_3(V_1, V_2, U_2, U_4), \\
V_4 &= f_4(V_3.U_3, U_4), \\
V_5 &= f_5(V_4, U_3),
\end{aligned}
\tag{8}
$$

where $U_1$ and $U_2$ are correlated and $U_3, U_4$ and $(U_1, U_2)$ are independent. The causal model is shown in Figure 2 and its causal graph is shown in Figure 3. In this example, $\boldsymbol{U}_{V_1} = \{U_1\}, \boldsymbol{U}_{V_2} = \{U_1, U_2\}, \boldsymbol{U}_{V_3} = \{U_2, U_4\}, \boldsymbol{U}_{V_4} = \{U_3, U_4\}, \boldsymbol{U}_{V_5} = \{U_3\}$. Since $U_1, U_2$ are correlated, $C_1 = \{V_1, V_2, V_3\}$ is one $C^2$ component because all nodes in $C_1$ are connected by bi-directed edges. Note that $\{V_1, V_2, V_3, V_4\}$ is not a $C^2$ component because $V_4$ and $V_2$ is not connected by any bi-directive edge. The rest of $C^2$ components are $C_2 = \{V_4, V_5\}, C_3 = \{V_3, V_4\}$.

Now, we consider the intervention $V_1 = t$. Under this intervention, the structure equations can be obtained by setting $V_1 = t$ while keeping all other equations unchanged, i.e.,

$$
\begin{aligned}
V_1(t) &= t, \\
V_2(t) &= f_2(U_1, U_2), \\
V_3 &= f_3(V_1, V_2, U_2, U_4), \\
V_4 &= f_4(V_3.U_3, U_4), \\
V_5(t) &= f_5(V_4, U_3).
\end{aligned}
$$

The figure of this model under intervention is shown in Figure 4.

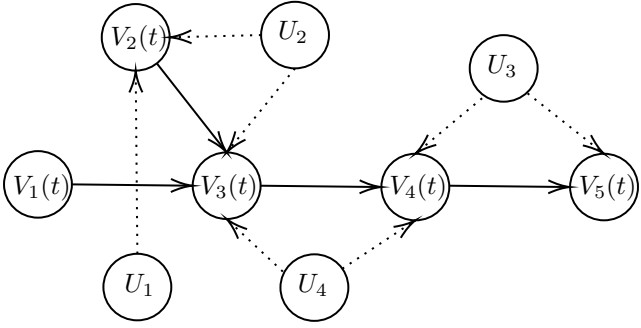

Figure 4: The SCM after intervention $V_1 = t$.

## B   Proof of approximation Theorems

### B.1   Illustration of how to construct canonical representations and neural architectures

In this section, we illustrate how to construct canonical representations and neural architectures given a causal graph via a simple example. We consider the example in the previous section. The causal graph is shown in Figure 3.

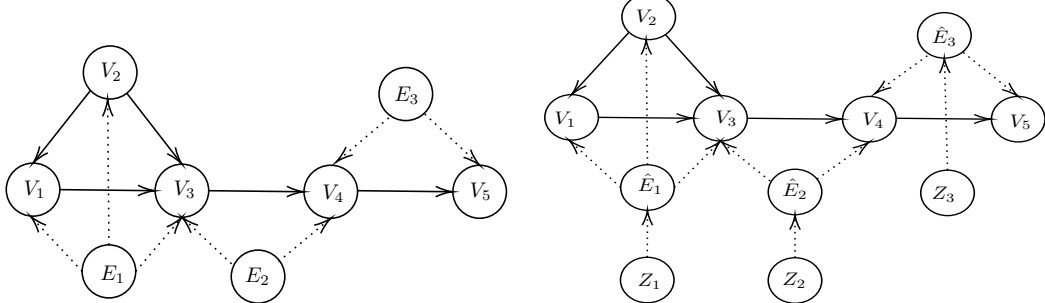

Figure 5: The canonical representation of Figure 3.

Figure 6: The neural network architecture.

Following the construction in Proposition 4, we use one latent variable for each $C^2$ component. As we explain in Appendix A, this causal model has three $C^2$ components, $\{V_1, V_2, V_3\}, \{V_3, V_4\}, \{V_4, V_5\}$. In the canonical representation, exactly the latent variables enter their corresponding $C^2$ component. The structure equation of this SCM is as follows.

$$
\begin{aligned}
V_1 &= f_1(V_2, E_1), \\
V_2 &= f_2(E_1), \\
V_3 &= f_3(V_1, V_2, E_1, E_2), \\
V_4 &= f_4(V_3.E_2), \\
V_5 &= f_5(V_4, E_3),
\end{aligned}
\tag{9}
$$

If we set $E_1 = (U_1, U_2), E_2 = U_4, E_3 = U_3$, (10) is equivalent to (8). Therefore, we can see from this example that the canonical representation does not lose any information about the SCM.

Now, we show how to construct the NCM architecture from a canonical representation. As we mentioned in Section 3, we approximate the latent distribution by pushing forward uniform and Gumbel variables. The structure equation of the NCM is

$$
\begin{aligned}
V_1 &= f_1^{\theta_1}(V_2, g_1^{\theta_1}(Z_1)), \\
V_2 &= f_2^{\theta_2}(g_1^{\theta_1}(Z_1)), \\
V_3 &= f_3^{\theta_2}(V_1, V_2, g_1^{\theta_1}(Z_1), g_2^{\theta_1}(Z_2)), \\
V_4 &= f_4^{\theta_4}(V_3.g_2^{\theta_1}(Z_2)), \\
V_5 &= f_5^{\theta_5}(V_4, g_2^{\theta_3}(Z_3)),
\end{aligned}
\tag{10}
$$

where $f_i^{\theta_i}, g_j^{\theta_j}$ are neural networks, $Z_i$ are join distribution of independent uniform and Gumbel variables, and $g_j^{\theta_j}$ has the special architecture described in Section 3.1 and Section 3.2. Figure 6 shows the architecture of the NCM. Each in-edge represents an input of a neural net.

### B.2   Proof of Theorem 1

The causal graph $\mathcal{G}$ does not specify how latent variables influence the observed variables. There could be many ways of recovering the latent variables from a graph $\mathcal{G}$. Figure 7 shows an example where two different causal models have the same causal graph. The next proposition gives a canonical representation of a causal model.

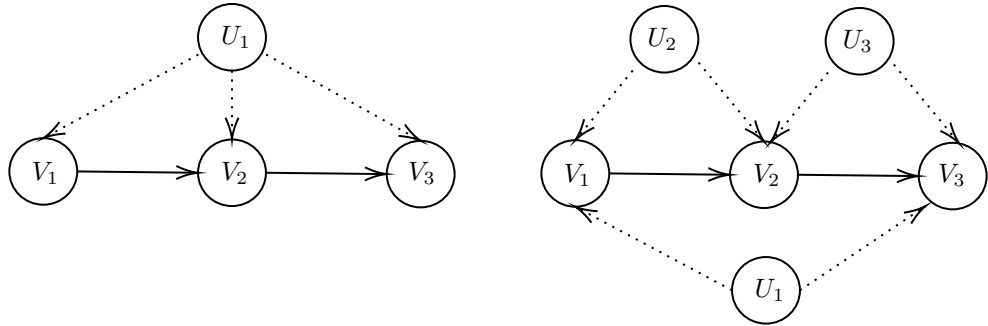

Figure 7: Example: two different SCMs with the same causal graph.

**Proposition 4.** *Suppose that Assumption 1 and Assumption 2 hold, given any SCM $\mathcal{M}$ with causal graph $\mathcal{G}$ and latent variables $\boldsymbol{U}$, we can construct a canonical SCM $\hat{\mathcal{M}}$ of the form (3) by merging the latent variables in $\boldsymbol{U}$ such that $\mathcal{M}$ and $\hat{\mathcal{M}}$ produce the same intervention results. Besides, functions in $\hat{\mathcal{M}}$ have the same smoothness as $\mathcal{M}$.*

*Proof of Proposition 4.* Let $\boldsymbol{G} = \{G_{V_i} : V_i \text{ is categorical}\}$ and the latent variables in the original model $\mathcal{M}$ be $\boldsymbol{U} = \{U_1, \cdots, U_{n_U}\} \cup \boldsymbol{G}$. By Assumption 1 and Assumption 2, $\boldsymbol{U}$ are independent and the structure equations of $\mathcal{M}$ have the form

$$V_i = \begin{cases} f_i\left(\mathrm{Pa}(V_i), \boldsymbol{U}_{V_i}\right), & V_i \text{ is continuous}, \\ \arg\max_{k \in [n_i]} \left\{g_k^{V_i} + \log\left(f_i\left(\mathrm{Pa}(V_i), \boldsymbol{U}_{V_i}\right)\right)_k\right\}, & V_i \text{ is categorical}, \end{cases}$$

where $g_k^{V_i}$ are i.i.d. standard Gumbel variables and $\boldsymbol{U}_{V_i} \subset \{U_1, \cdots, U_{n_U}\}$ contains the latent variables that affect $V_i$. We regroup and merge the variables $U_i$ to make the model have a canonical form while not changing the functions in the structure equations. Let $D_1, \cdots, D_{n_C} \subset \boldsymbol{V}$ be the $C^2$-component of $\mathcal{M}$. For each $U_i$, we define the vertices that are affected by $U_i$ as

$$I(U_i) = \{V_j : U_i \in \boldsymbol{U}_{V_j}\}.$$

We partition $\{U_1, \cdots, U_{n_U}\}$ into $n_C$ sets $\hat{\boldsymbol{U}}_1, \cdots, \hat{\boldsymbol{U}}_{n_C}$ in the following way. For each $U_k$, $U_k$ is in the set $\hat{\boldsymbol{U}}_i$ if $I(U_k) \subset D_i$. If there are two components $D_i, D_j$ satisfy the condition, we put $U_k$ into either of the sets $\hat{\boldsymbol{U}}_i, \hat{\boldsymbol{U}}_j$. Let $\hat{U}_i$ have the same distribution as the joint distribution of the random variables in set $\hat{\boldsymbol{U}}_i$. Let $\hat{\mathcal{M}} \in \mathcal{M}(\mathcal{G}, \mathcal{F}, \hat{\boldsymbol{U}})$ the SCM with structure equations

$$\hat{V}_i = \begin{cases} f_i\left(\mathrm{Pa}(\hat{V}_i), \hat{U}_{k_1}, \cdots, \hat{U}_{k_{n_i}}\right), & V_i \text{ is continuous}, \\ \arg\max_{k \in [n_i]} \left\{g_k^{V_i} + \log\left(f_i\left(\mathrm{Pa}(\hat{V}_i), \hat{U}_{k_1}, \cdots, \hat{U}_{k_{n_i}}\right)\right)_k\right\}, & V_i \text{ is categorical}, \end{cases} \hat{\boldsymbol{U}}_{k_i} \cap \boldsymbol{U}_{V_i} \neq \emptyset, \ i \in [n_V].$$

Here, we slightly abuse the notation $f_i$. $f_i$ ignores inputs from $\hat{\boldsymbol{U}}_{k_1}, \cdots, \hat{\boldsymbol{U}}_{k_{n_i}}$ that are not in $\boldsymbol{U}_{V_i}$. Note that $\hat{\boldsymbol{U}}_1, \cdots, \hat{\boldsymbol{U}}_{n_C}, \boldsymbol{G}$ has the same distribution as $\boldsymbol{U}$ because we only merge some latent variables. In addition, the functions in the new model $\hat{\mathcal{M}}$ has the same smoothness as the original model.

We first verify that the causal graph of $\hat{\mathcal{M}}$ is $\mathcal{G}$. If there is a bi-directed arrow between nodes $V_i, V_j$ in $\mathcal{G}$, by the independence assumption, there must be one latent variable $U_k \in \boldsymbol{U}_{V_i} \cap \boldsymbol{U}_{V_j}$. There exist one $D_l$ such that $I(U_k) \subset D_l$ and $\hat{\boldsymbol{U}}_l \cap \boldsymbol{U}_{V_i} \neq \emptyset, \hat{\boldsymbol{U}}_l \cap \boldsymbol{U}_{V_j} \neq \emptyset$. Therefore, $\hat{U}_l$ will affect both $V_i$ and $V_j$ and there is a bi-directed arrow between node $V_i, V_j$ in $\mathcal{G}_{\hat{\mathcal{M}}}$. Suppose that there is a bi-directed arrow between node $V_i, V_j$ in $\mathcal{G}_{\hat{\mathcal{M}}}$, it means there exist a $\hat{U}_k$ such that $\hat{\boldsymbol{U}}_k \cap \boldsymbol{U}_{V_i} \neq \emptyset, \hat{\boldsymbol{U}}_k \cap \boldsymbol{U}_{V_j} \neq \emptyset$. Let $U_{l_1} \in \hat{\boldsymbol{U}}_k \cap \boldsymbol{U}_{V_i}, U_{l_2} \in \hat{\boldsymbol{U}}_k \cap \boldsymbol{U}_{V_j}$, then

$$V_i \in I(U_{l_1}) \subset D_k, V_j \in I(U_{l_2}) \subset D_k.$$

Since $D_k$ is a $C^2$-component, there exist a bi-directed arrow between node $V_i, V_j$ in $\mathcal{G}$. Therefore, $\mathcal{G} = \mathcal{G}_{\hat{\mathcal{M}}}$.

Finally, we verify that $\mathcal{M}$ and $\hat{\mathcal{M}}$ produce the same intervention results. The intervention distribution can be viewed as the push-forward of the latent distribution, i.e., $\boldsymbol{V} = F(\boldsymbol{U})$, and the intervention operation only changes the function $F$. In our construction, we only merge the latent variables and leave the functions in structure equations being the same (except that they may ignore some coordinates in input). For any intervention $T = t$, suppose in $\mathcal{M}$ we have $\boldsymbol{V}(t) = F_t(\boldsymbol{U})$. Then, in $\hat{\mathcal{M}}$, we get $\hat{\boldsymbol{V}}(t) = F_t(\hat{\boldsymbol{U}})$. Since $\boldsymbol{U}$ and $\hat{\boldsymbol{U}}$ has the same distribution, the distribution of $\boldsymbol{V}(t)$, $\hat{\boldsymbol{V}}(t)$ are the same. $\qquad\square$

*Proof of Theorem 1.* The structure equations of $\mathcal{M}$ have the form

$$
V_i = \begin{cases} f_i\left(\mathrm{Pa}(V_i), \boldsymbol{U}_{V_i}\right), & V_i \text{ is continuous,} \\ \arg\max_{k\in[n_i]}\left\{g_k^{V_i} + \log\left(f_i\left(\mathrm{Pa}(V_i), \boldsymbol{U}_{V_i}\right)\right)_k\right\}, & V_i \text{ is categorical, } \|f_i\|_1 = 1, \end{cases}
$$

where $g_k^{V_i}$ are i.i.d. standard Gumbel variables and $\hat{\mathcal{M}} \in \mathcal{M}(\mathcal{G}, \hat{\mathcal{F}}, \hat{\boldsymbol{U}})$ has structure equations

$$
\hat{V}_i = \begin{cases} \hat{f}_i\left(\mathrm{Pa}(\hat{V}_i), \hat{\boldsymbol{U}}_{V_i}\right), & V_i \text{ is continuous,} \\ \arg\max_{k\in[n_i]}\left\{\hat{g}_k^{V_i} + \log\left(\hat{f}_i\left(\mathrm{Pa}(\hat{V}_i), \hat{\boldsymbol{U}}_{V_i}\right)\right)_k\right\}, & V_i \text{ is categorical, } \|\hat{f}_1\|_1 = 1, \end{cases}
$$

where $\hat{g}_k^{V_i}$ are i.i.d. standard Gumbel variables. Let the treatment variables set be $\boldsymbol{T} = \{T_1, \cdots, T_{n_T}\}$. We may give all vertices and $\boldsymbol{U}_0 = \{U_1, \cdots, U_{n_U}\}$, an topology ordering (rearrange the subscript if necessary)

$$
U_1, \cdots, U_{n_U}, T_1, \cdots, T_{n_T}, V_1(\boldsymbol{t}), \cdots, V_{n_V - n_T}(\boldsymbol{t})
$$

such that for each directed edge $(V_i(\boldsymbol{t}), V_j(\boldsymbol{t}))$, vertex $V_i(\boldsymbol{t})$ lies before vertex $V_j(\boldsymbol{t})$ in the ordering, ensuring that all edges start from vertices that appear early in the order and end at vertices that appear later in the order. We put $\boldsymbol{U}_0$ and $\boldsymbol{T}$ at the beginning because they are the root nodes of the intervened model. We denote $\mu_{\boldsymbol{U}_0, \boldsymbol{T}, 1:k}^{\boldsymbol{T}}$ (resp. $\hat{\mu}_{\boldsymbol{U}_0, \boldsymbol{T}, 1:k}^{\boldsymbol{T}}$) to be the distribution of $\boldsymbol{U}_0, \boldsymbol{T}, V_1(\boldsymbol{t}), \cdots, V_k(\boldsymbol{t})$ (resp. $\hat{\boldsymbol{U}}_0, \boldsymbol{T}, \hat{V}_1(\boldsymbol{t}), \cdots, \hat{V}_k(\boldsymbol{t})$), $\mu_{V_k|\boldsymbol{U}_0, \boldsymbol{T}, 1:k-1}^{\boldsymbol{T}}$ the distribution of $V_k$ given $\boldsymbol{U}_0, \boldsymbol{T}, V_1(\boldsymbol{t}), \cdots, V_{k-1}(\boldsymbol{t})$.

Let $S = \sum_{i=1}^{n_V} \|f_i - \hat{f}_i\|_\infty$. Next, we prove that

$$
W\left(\mu_{\boldsymbol{U}_0, \boldsymbol{T}, 1:k}^{\boldsymbol{T}}, \hat{\mu}_{\boldsymbol{U}_0, \boldsymbol{T}, 1:k}^{\boldsymbol{T}}\right) \leqslant (L+1)W\left(\mu_{\boldsymbol{U}_0, \boldsymbol{T}, 1:k-1}^{\boldsymbol{T}}, \hat{\mu}_{\boldsymbol{U}_0, \boldsymbol{T}, 1:k-1}^{\boldsymbol{T}}\right) + 2K^2 S. \tag{11}
$$

By definition of Wasserstein-1 distance,

$$
\begin{aligned}
W\left(\mu_{\boldsymbol{U}_0, \boldsymbol{T}, 1:k}^{\boldsymbol{T}}, \hat{\mu}_{\boldsymbol{U}_0, \boldsymbol{T}, 1:k}^{\boldsymbol{T}}\right) &= \sup_{g\in Lip(1)} \int g(\boldsymbol{u}, \boldsymbol{t}, v_1, \cdots, v_k) d\left(\mu_{\boldsymbol{U}_0, \boldsymbol{T}, 1:k}^{\boldsymbol{T}} - \hat{\mu}_{\boldsymbol{U}_0, \boldsymbol{T}, 1:k}^{\boldsymbol{T}}\right) \\
&= \sup_{g\in Lip(1)} \int g(\boldsymbol{u}, \boldsymbol{t}, v_1, \cdots, v_k)\, d\mu_{V_k|\boldsymbol{U}_0, \boldsymbol{T}, 1:k-1}^{\boldsymbol{T}}\, d\mu_{\boldsymbol{U}_0, \boldsymbol{T}, 1:k-1}^{\boldsymbol{T}} \\
&\qquad - \int g(\boldsymbol{u}, \boldsymbol{t}, v_1, \cdots, v_k)\, d\hat{\mu}_{\hat{V}_k|\hat{\boldsymbol{U}}_0, \boldsymbol{T}, 1:k-1}^{\boldsymbol{T}}\, d\hat{\mu}_{\hat{\boldsymbol{U}}_0, \boldsymbol{T}, 1:k-1}^{\boldsymbol{T}} \\
&= \sup_{g\in Lip(1)} \underbrace{\int g(\boldsymbol{u}, \boldsymbol{t}, v_1, \cdots, v_k)\, d\left(\mu_{V_k|\boldsymbol{U}_0, \boldsymbol{T}, 1:k-1}^{\boldsymbol{T}} - \hat{\mu}_{\hat{V}_k|\hat{\boldsymbol{U}}_0, \boldsymbol{T}, 1:k-1}^{\boldsymbol{T}}\right) d\hat{\mu}_{\hat{\boldsymbol{U}}_0, \boldsymbol{T}, 1:k-1}^{\boldsymbol{T}}}_{(1)} \\
&\qquad + \underbrace{\int g(\boldsymbol{u}, \boldsymbol{t}, v_1, \cdots, v_k)\, d\mu_{V_k|\boldsymbol{U}_0, \boldsymbol{T}, 1:k-1}^{\boldsymbol{T}}\, d\left(\mu_{\boldsymbol{U}_0, \boldsymbol{T}, 1:k-1}^{\boldsymbol{T}} - \hat{\mu}_{\hat{\boldsymbol{U}}_0, \boldsymbol{T}, 1:k-1}^{\boldsymbol{T}}\right)}_{(2)}.
\end{aligned}
$$

For (1), if $V_k$ is a continuous variable, we have

$$
\begin{aligned}
&\int g(\boldsymbol{u}, \boldsymbol{t}, v_1, \cdots, v_k)\, d\left(\mu_{V_k|\boldsymbol{U}_0, \boldsymbol{T}, 1:k-1}^{\boldsymbol{T}} - \hat{\mu}_{\hat{V}_k|\hat{\boldsymbol{U}}_0, \boldsymbol{T}, 1:k-1}^{\boldsymbol{T}}\right) \\
&= g\left(\boldsymbol{u}, \boldsymbol{t}, v_1, \cdots, v_{k-1}, f_k\left(\mathrm{pa}(v_k), \boldsymbol{u}_{V_k}\right)\right) - g\left(\boldsymbol{u}, \boldsymbol{t}, v_1, \cdots, v_{k-1}, \hat{f}_k\left(\mathrm{pa}(v_k), \boldsymbol{u}_{V_k}\right)\right) \\
&\leqslant \left\|f_k\left(\mathrm{pa}(v_k), \boldsymbol{u}_{V_k}\right) - \hat{f}_k\left(\mathrm{pa}(v_k), \boldsymbol{u}_{V_k}\right)\right\|_\infty \leqslant S, \tag{12}
\end{aligned}
$$

where we have used the Lipschitz property of $g$ in the first inequality. If $V_k$ is categorical, let $\hat{p}\left(\mathrm{pa}(v_k), \boldsymbol{u}_{V_k}\right) = (\hat{p}_1, \cdots, \hat{p}_{n_i}) = \hat{f}\left(\mathrm{pa}(v_k), \boldsymbol{u}_{V_k}\right)$ and $p\left(\mathrm{pa}(v_k), \boldsymbol{u}_{V_k}\right) = (p_1, \cdots, p_{n_i}) = f\left(\mathrm{pa}(v_k), \boldsymbol{u}_{V_k}\right)$, we get

$$\int g(\boldsymbol{u}, \boldsymbol{t}, v_1, \cdots, v_k) \, d\left(\mu^{\boldsymbol{T}}_{V_k | \boldsymbol{U}_0, \boldsymbol{T}, 1:k-1} - \hat{\mu}^{\boldsymbol{T}}_{\hat{V}_k | \hat{\boldsymbol{U}}_0, \boldsymbol{T}, 1:k-1}\right)$$

$$= \sum_{k=1}^{n_i - 1} (g(\boldsymbol{u}, \boldsymbol{t}, v_1, \cdots, k) - g(\boldsymbol{u}, \boldsymbol{t}, v_1, \cdots, n_i))(p_k - \hat{p}_k) \leqslant K \sum_{k=1}^{n_i-1} |p_k - \hat{p}_k| \leqslant K^2 S, \quad (13)$$

where we use $\|V_k\|_\infty \leqslant K, n_i \leqslant K$.

For (2), if $V_i$ is continuous,

$$\int g(\boldsymbol{u}, \boldsymbol{t}, v_1, \cdots, v_k) \, d\mu^{\boldsymbol{T}}_{V_k | \boldsymbol{U}_0, \boldsymbol{T}, 1:k-1} \, d\left(\mu^{\boldsymbol{T}}_{\boldsymbol{U}_0, \boldsymbol{T}, 1:k-1} - \hat{\mu}^{\boldsymbol{T}}_{\hat{\boldsymbol{U}}_0, \boldsymbol{T}, 1:k-1}\right)$$

$$= \int g\left(\boldsymbol{u}, \boldsymbol{t}, v_1, \cdots, v_{k-1}, f_k\left(\mathrm{pa}(v_k), \boldsymbol{u}_{V_k}\right)\right) \, d\left(\mu^{\boldsymbol{T}}_{\boldsymbol{U}_0, \boldsymbol{T}, 1:k-1} - \hat{\mu}^{\boldsymbol{T}}_{\hat{\boldsymbol{U}}_0, \boldsymbol{T}, 1:k-1}\right). \quad (14)$$

Since $g, f_k$ are Lipschitz continuous functions, $g\left(\boldsymbol{u}, \boldsymbol{t}, v_1, \cdots, v_{k-1}, f_k\left(\mathrm{pa}(v_k), \boldsymbol{u}_{V_k}\right)\right)$ is $(L+1)$-Lipschitz continuous with respect to $(\boldsymbol{u}, \boldsymbol{t}, v_1, \cdots, v_{k-1})$. We have

$$\int g\left(\boldsymbol{u}, \boldsymbol{t}, v_1, \cdots, v_{k-1}, f_k\left(\mathrm{pa}(v_k), \boldsymbol{u}_{V_k}\right)\right) d\left(\mu^{\boldsymbol{T}}_{\boldsymbol{U}_0, \boldsymbol{T}, 1:k-1} - \hat{\mu}^{\boldsymbol{T}}_{\boldsymbol{U}_0, \boldsymbol{T}, 1:k-1}\right)$$

$$\leqslant (L+1) W\left(\mu^{\boldsymbol{T}}_{\boldsymbol{U}_0, \boldsymbol{T}, 1:k-1}, \hat{\mu}^{\boldsymbol{T}}_{\hat{\boldsymbol{U}}_0, \boldsymbol{T}, 1:k-1}\right). \quad (15)$$

If $V_i$ is categorical,

$$\int g(\boldsymbol{u}, \boldsymbol{t}, x_1, \cdots, x_k) \, d\mu^{\boldsymbol{T}}_{X_k | \boldsymbol{U}, \boldsymbol{T}, 1:k-1} \, d\left(\mu^{\boldsymbol{T}}_{\boldsymbol{U}, \boldsymbol{T}, 1:k-1} - \hat{\mu}^{\boldsymbol{T}}_{\hat{\boldsymbol{U}}, \boldsymbol{T}, 1:k-1}\right)$$

$$= \sum_{k=1}^{n_i} p_k \int g(\boldsymbol{u}, \boldsymbol{t}, x_1, \cdots, k) \, d\left(\mu^{\boldsymbol{T}}_{\boldsymbol{U}, \boldsymbol{T}, 1:k-1} - \hat{\mu}^{\boldsymbol{T}}_{\hat{\boldsymbol{U}}, \boldsymbol{T}, 1:k-1}\right).$$

Since $g, f_k$ are $L$-Lipschitz continuous functions, $g(\boldsymbol{u}, \boldsymbol{t}, v_1, \cdots, v_{k-1}, i)$ is $L$-Lipschitz continuous with respect to $(\boldsymbol{u}, \boldsymbol{t}, v_1, \cdots, v_{k-1})$. We have

$$\sum_{k=1}^{n_i} \int p_k g(\boldsymbol{u}, \boldsymbol{t}, x_1, \cdots, k) \, d\left(\mu^{\boldsymbol{T}}_{\boldsymbol{U}_0, \boldsymbol{T}, 1:k-1} - \hat{\mu}^{\boldsymbol{T}}_{\boldsymbol{U}_0, \boldsymbol{T}, 1:k-1}\right) \leqslant L W\left(\mu^{\boldsymbol{T}}_{\boldsymbol{U}_0, \boldsymbol{T}, 1:k-1}, \hat{\mu}^{\boldsymbol{T}}_{\hat{\boldsymbol{U}}_0, \boldsymbol{T}, 1:k-1}\right).$$

$$(16)$$

Combine (12)-(16), we prove Equation (11). By induction, one can easily get

$$W\left(\mu^{\boldsymbol{T}}_{\boldsymbol{U}_0, \boldsymbol{T}, 1:n_V - n_T}, \hat{\mu}^{\boldsymbol{T}}_{\hat{\boldsymbol{U}}_0, \boldsymbol{T}, 1:n_V - n_T}\right) \leqslant (L+1)^{n_V - n_T} \left(W\left(\mu^{\boldsymbol{T}}_{\boldsymbol{U}_0, \boldsymbol{T}}, \hat{\mu}^{\boldsymbol{T}}_{\hat{\boldsymbol{U}}_0, \boldsymbol{T}}\right) + 2K^2 S/L\right) - 2K^2 S/L$$

$$= \frac{(L+1)^{n_V - n_T} - 1}{L} 2K^2 S + (L+1)^{n_V - n_T} W\left(\mu^{\boldsymbol{T}}_{\boldsymbol{U}_0, \boldsymbol{T}}, \hat{\mu}^{\boldsymbol{T}}_{\hat{\boldsymbol{U}}_0, \boldsymbol{T}}\right)$$

Let $C_{\mathcal{G}}(L, K) = 2K^2 \cdot \max\{\frac{(L+1)^{n_V - n_T} - 1}{L}, (L+1)^{n_V - n_T}\}$ and notice that $W\left(\mu^{\boldsymbol{T}}_{\boldsymbol{U}_0, \boldsymbol{T}}, \hat{\mu}^{\boldsymbol{T}}_{\hat{\boldsymbol{U}}_0, \boldsymbol{T}}\right) = W\left(P^{\mathcal{M}}(\boldsymbol{U}), P^{\hat{\mathcal{M}}}(\hat{\boldsymbol{U}})\right)$ because interventions $\boldsymbol{T} = \boldsymbol{t}$ are the same, we get

$$W\left(P^{\mathcal{M}}(\boldsymbol{V}(\boldsymbol{t})), P^{\hat{\mathcal{M}}}(\boldsymbol{V}(\boldsymbol{t}))\right) \leqslant W\left(\mu^{\boldsymbol{T}}_{\boldsymbol{U}_0, \boldsymbol{T}, 1:n_V - n_T}, \hat{\mu}^{\boldsymbol{T}}_{\hat{\boldsymbol{U}}_0, \boldsymbol{T}, 1:n_V - n_T}\right)$$

$$\leqslant C_{\mathcal{G}}(L, K)\left(S + W\left(P^{\mathcal{M}}(\boldsymbol{U}), P^{\hat{\mathcal{M}}}(\hat{\boldsymbol{U}})\right)\right). \quad (17)$$

$\square$

## B.3 Proof of results in Section 3.1

*Proof of Theorem 2.* Recall that the neural network consists of three parts $\hat{g} = \hat{g}_3^\tau \circ \hat{g}_2 \circ \hat{g}_1$ and $\hat{g}_2, \hat{g}_1$ have a separable form, dealing with each coordinate of the input individually. As mentioned before, each coordinate approximates the distribution of one connected component of the support. We first construct $\hat{g}_1$ and $\hat{g}_2$, then use Gumbel-Softmax layer to combine each component together.

To construct the first two parts, we only need to consider each coordinate individually. For the $i$-th component $C_i$, let $\mu_i = \mathbb{P}_{C_i}/\mathbb{P}(C_i)$, where $\mathbb{P}_{C_i}$ is measure $\mathbb{P}$ restricted to component $C_i$. Under Assumption 4, there exists a Lipschitz map $g_2^i \in \mathcal{H}([0,1]^{d_i^C}, C_i) \cap \mathcal{F}_L([0,1]^{d_i^C}, C_i)$. By [43, Theorem VIII.1], there exists quantized ReLu network $\hat{g}_1^i$ of width $W_1$ and depth $\Theta(d_i^C)$ (let $s = \frac{1}{\lceil n(n-1) \rceil}$ in [43, Theorem VIII.1]), such that

$$W\left(\left((g_2^i)^{-1}\right)_\# \mu_i, P(\hat{g}_1^i(U_i))\right) \leqslant O\left(W_1^{-1/d_i^C}\right),$$

where $U_i$ are i.i.d. uniform random variables on $[0,1]$. By [53, Theorem 2], there exist a deep ReLu network $\hat{g}_2^{i,j}$ of width $\Theta\left(d_i^C\right)$ and depth $L_2$ such that

$$\| (g_2^i)_j - \hat{g}_2^{i,j} \|_\infty \leqslant O\left(L_2^{-2/d_i^C}\right),$$

Let $\hat{g}_2^i(x) = \left(\hat{g}_2^{i,1}(x), \cdots, \hat{g}_2^{i,d}(x)\right)$, we get

$$\|g_2^i - \hat{g}_2^i\|_\infty \leqslant O\left(L_2^{-2/d_i^C}\right).$$

Thus, the width of $\hat{g}_2^i$ is $\Theta\left(d \cdot d_i^C\right)$. Let

$$\hat{g}_i(x) = \left(\hat{g}_i^1(x_1), \cdots, \hat{g}_i^{N_C}(x_{N_C})\right), i = 1, 2, x_k \in \mathbb{R}^{d_i}, d_1 = 1, d_2 = d, k = 1, \cdots, N_C,$$

where $N_C$ is the number of connected components. By Lemma 3, we get

$$\begin{aligned}
W\left(\mu_i, P(\hat{g}_2^i \circ \hat{g}_1^i(U_i))\right) &= W\left((g_2^i)_\# (g_2^i)_\#^{-1} \mu_i, (\hat{g}_2^i)_\# P(\hat{g}_1^i(U_i))\right) \\
&\leqslant LW\left((g_2^i)_\#^{-1} \mu_i, P(\hat{g}_1^i(U_i))\right) + \|g_2^i - \hat{g}_2^i\|_\infty \\
&= O\left(W_1^{-1/d_i^C} + L_2^{-2/d_i^C}\right).
\end{aligned}$$

Next, let $p_i = \mathbb{P}(C_i)$ and the distribution of $\hat{g}_2^k \circ \hat{g}_1^k(U_k)$ be $\hat{\mu}_k$, then we have $\mathbb{P} = \sum_{k=1}^{N_C} p_k \mu_k$. By Lemma 4, we have

$$W\left(\mathbb{P}, \sum_{k=1}^{N_C} p_k \hat{\mu}_k\right) \leqslant \sum_{k=1}^{N_C} p_k W\left(\mu_i, \hat{\mu}_k\right) = O\left(W_1^{-1/\max\{d_i^C\}} + L_2^{-2/\max\{d_i^C\}}\right).$$

Finally, we analyze the error caused by the Gumbel-Softmax layer. Note that as temperature parameter $\tau \to 0$,

$$w^\tau = \left(\frac{\exp((\log p_i + G_i)/\tau)}{\sum_{k=1}^{N_C} \exp((\log p_k + G_k)/\tau)}\right)_{i=1,\cdots,N_C} \xrightarrow{a.s.} \text{One-hot}(\arg\max_i\{G_i + \log p_i\})$$

where One-hot$(k)$ is $N_C$-dimensional vector with the $k$-th coordinate equals to 1 and the remaining coordinates are 0 and $G_i \sim \text{Gumbel}(0,1)$ are i.i.d. Gumbel distribution. Let $\nu^\tau$ be the distribution of $w^\tau$. By Lemma 6, for $0 < \tau < 1$, we have

$$W\left(\nu^\tau, \nu^0\right) \leqslant O(\tau - \tau \log \tau).$$

By Lemma 5,

$$W\left(\left(\nu^\tau, P\left(\left(\hat{g}_2^i \circ \hat{g}_1^i\right)(U_i)\right)\right), \left(\nu^0, P\left(\left(\hat{g}_2^i \circ \hat{g}_1^i\right)(U_i)\right)\right)\right) \leqslant O\left(\tau - \tau \log \tau\right). \tag{18}$$

Let

$$\hat{g}_3^\tau(x_1, \cdots, x_{N_C}) = \sum_{k=1}^{N_C} w_k^\tau \cdot x_k, \quad x_k \in \mathbb{R}^d, w^\tau = \left(w_1^\tau, \cdots, w_{N_C}^\tau\right).$$

We denote $g_3^0 = \lim_{\tau \to 0} g_3^\tau$. By Assumption 4, the support is bounded. Thus, $g_2^i$ and $\hat{g}_2^i$ are bounded, i.e., $\|\hat{g}_2^i(x)\|_\infty \leqslant K$. Since functions $h(w, X) = w^\mathsf{T} X$ is Lipschitz continuous in the region $\|w\|_1 = 1, \|X\|_\infty \leqslant K$, we get

$$W\left(P(\hat{g}_3^0 \circ \hat{g}_2 \circ \hat{g}_1(U_k)), P(\hat{g}_3^\tau \circ \hat{g}_2 \circ \hat{g}_1(U_k))\right) \leqslant O\left(\tau - \tau \log \tau\right) \tag{19}$$

from (18). Putting things together, we get

$$W\left(\mathbb{P}, P(\hat{g}_3^\tau \circ \hat{g}_2 \circ \hat{g}_1(U_k))\right) \leqslant W\left(\mathbb{P}, \sum_{k=1}^{N_C} p_k \hat{\mu}_k\right) + W\left(\sum_{k=1}^{N_C} p_k \hat{\mu}_k, P(\hat{g}_3^\tau \circ \hat{g}_2 \circ \hat{g}_1(U_k))\right)$$

$$= W\left(\mathbb{P}, \sum_{k=1}^{N_C} p_k \hat{\mu}_k\right) + W\left(P(\hat{g}_3^0 \circ \hat{g}_2 \circ \hat{g}_1(U_k)), P(\hat{g}_3^\tau \circ \hat{g}_2 \circ \hat{g}_1(U_k))\right)$$

$$\leqslant O\left(\tau - \tau \log \tau + W_1^{-1/\max\{d_i^C\}} + L_2^{-2/\max\{d_i^C\}}\right),$$

where we use (19). $\qquad\qquad\square$

### B.4  Proof of results in Section 3.2

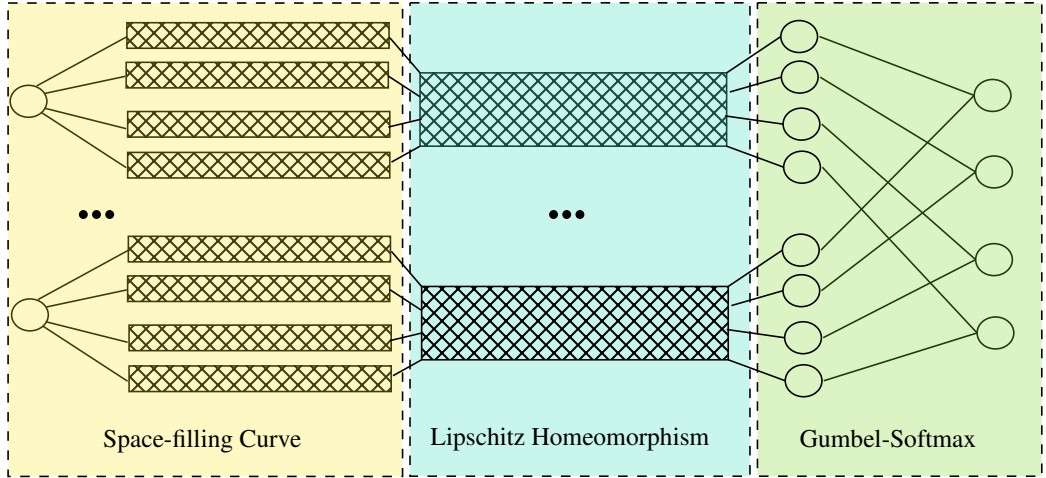

Figure 8: Architecture of the deep neural network for $4-$dimensional output. The first (yellow) part approximates the distribution on different connected components using deep ReLu Networks. The remaining two parts are similar to the wide neural network in Figure 1a.

Before we prove Proposition 1, we need to introduce some important notions. Let $\{\mathcal{C}_n\}$ be a sequence of partitions of the unit cube $[0,1]^d$. We say $\{\mathcal{C}_n\}$ has property (*) if the following properties are satisfied.

1. $\mathcal{C}_0 = \{C_{-1}\}, C_{-1} = [0,1]^d$
2. $\mathcal{C}_{n+1} = \{C_{i_1,i_2\cdots,i_{n+1}}\}_{i_j=0,\cdots,2^d-1}$ is a refinement of $\mathcal{C}_n = \{C_{i_1,i_2\cdots,i_n}\}_{i_j=0,\cdots,2^d-1}$, i.e., $C_{i_1,i_2\cdots,i_n} = \bigcup_{i_{n+1}=0}^{2^d-1} C_{i_1,i_2\cdots,i_{n+1}}$. Besides, $\mathcal{C}_n$ is obtained by cutting the unit cube $[0,1]^d$ by planes that are paralleled to the coordinate planes.

The following lemma is important in the construction of Hilbert curve.

**Lemma 1** (Algorithm 3 in [25]). *Given a sequence of partitions of the unit cube $\{\mathcal{C}_n\}$ that has property (*), there exist a ordering for each partition $\mathcal{C}_n$, i.e., $O_n : \{1, \cdots, 2^{nd}\} \to \{\overline{i_1 i_2 \cdots i_n} : i_j = 0, \cdots, 2^d - 1\}$, such that the following two conclusions hold.*

1. *If $(j-1)\cdot 2^d < i \leqslant j\cdot 2^d$, the first $n$ digits of $O_{n+1}(i)$ are the same as $O_n(j)$.*

2. *Adjacent cubes in the ordering intersect, i.e., $C_{O_n(i)} \cap C_{O_n(i+1)} \neq \emptyset$. Furthermore, $C_{O_n(i)} \cap C_{O_n(i+1)}$ is a $d-1$ dimensional cube.*

*Proof of Proposition 1.* Without loss of generality, we can assume that $\mathbb{P}$ is supported on $[0,1]^d$, vanishes at the boundary and $\mathbb{P}(B) \geqslant C_1 \lambda(B)$ for some constant $C_1 > 0$ and all measurable sets $B \in [0,1]^d$. Suppose we have proven the conclusion for this case. By Assumption 5, there exists $f \in \mathcal{H}([0,1]^d, K) \cap \mathcal{F}_L([0,1]^d, K)$ such that $f_{\#}^{-1}\mathbb{P}$ satisfies these conditions. There exists continuous function $\gamma$ such that $f_{\#}^{-1}\mathbb{P} = \gamma_{\#}\lambda$, which implies $\mathbb{P} = (f \circ \gamma)_{\#}\lambda$. In particular, if $\gamma$ is $1/d$-Holder continuous, $f \circ \gamma$ is also $1/d$-Holder continuous because $f$ is Lipschitz continuous.

When $d = 0$, the constant map satisfies the requirement. We focus on the case $d \geqslant 1$ in the following. The proof is to modify the construction of the Hilbert space-filling curve. By changing the velocity of the curve, the push-forward measure of the Hilbert space-filling curve can simulate a great number of distributions. We use $\lambda$ to represent the Lebesgue measure $[0,1]$.

**Proof Sketch:** The construction of $f$ is inspired by the famous Hilbert space-filling curve [28]. To illustrate the idea, let us first assume that $\mathbb{P}$ is absolutely continuous with respect to the Lebesgue measure $\lambda$ and consider $d = 2$. In the $k$-th step, we divide the unit cube into $2^{2k}$ evenly closed cubes $\mathcal{C}_n = \{C_{1,n}, \cdots, C_{2^{2k},n}\}$ such that $\mathcal{C}_n$ is a refinement of $\mathcal{C}_{n-1}$.

The construction of a standard Hilbert curve is to find a sequence of curves $\gamma_n$ that go through all the cubes in $\mathcal{C}_n$. The curve $\gamma_n$ has one special property. If $\gamma_{n-1}(t) \in C_{k,n-1}$, then $\gamma_m(t) \in C_{k,n-1}, \forall m \geqslant n$. For example, in Figure 9, the points on the curve in the lower-left cubes will stay in the lower-left cubes. Note that $(\gamma_n)_{\#}\lambda(C_{k,n}) = \lambda\left(\gamma_n^{-1}(C_{k,n})\right)$ is the time curve $\gamma_n$ stays in cubes $C_{k,n}$. The idea is to change the speed of the curve so that $\mathbb{P}$ and $(\gamma_n)_{\#}\lambda$ agree on cubes in $\mathcal{C}_n$, i.e., $\mathbb{P}(C_{k,n}) = \lambda\left(\gamma_n^{-1}(C_{k,n})\right)$. Since we assume that $\mathbb{P}$ is absolutely continuous, $\mathbb{P}(\partial C_{k,n}) = 0$ and we don't need to worry about how to divide the mass on the boundary. For example, let the green cubes in the Figure 9 to be $C_0$ and suppose that $\gamma_1$ starts from $C_0$. We will change the speed so that $\gamma_1$ spends $\mathbb{P}(C_0)$ time in this region. In the next step, we divide $C_0$ into four colored cubes $C_1, \cdots, C_4$ on the right. We change the speed again to let the time spent in each cube equal to $\mathbb{P}(C_i)$. Note that this construction preserves the aforementioned property, i.e., for $t \in [0, \mathbb{P}(C_0)]$, $\gamma_n(t) \in C_0, \forall n \geqslant 1$. As $n \to \infty$, it can be proven that $\gamma_n$ converges uniformly to a curve $\gamma$. We can also prove that $\mathbb{P}$ and $\gamma_{\#}\lambda$ agree on $\cup_{n=1}^{\infty}\mathcal{C}_n$. Given that $\cup_{n=1}^{\infty}\mathcal{C}_n$ generate the standard Borel algebra on $[0,1]^2$, we can conclude that $\mathbb{P} = \gamma_{\#}\lambda$.

In the general case, $\mathbb{P}$ may not be absolutely continuous. As a result, the boundary may not be $\mathbb{P}$-measure zero. We will need to perturb the cutting planes to ensure their boundaries are measured zero.

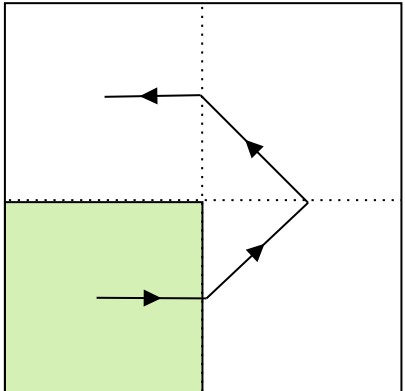 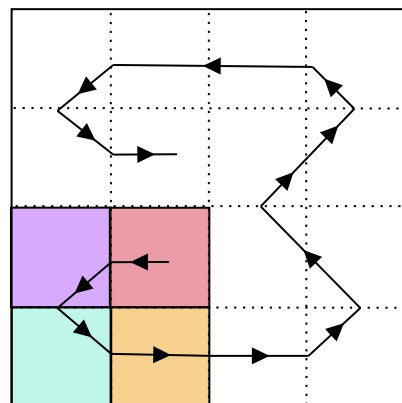

Figure 9: The construction of Hilbert curve.

**Preparation.** For a $n$-dimensional cube $C = \prod_{i=1}^{n}[a_i, b_i]$, the diameter of $C$ is $\mathrm{diam}(C) = \sqrt{\sum_{i=1}^{n}(b_i - a_i)^2}$. We also define the function $R$ that measures the ratio between maximum edge and minimum edge. For a collection of cubes $\mathcal{C} = \{C_1, \cdots, C_m\}, C_k = \prod_{i=1}^{n}[a_i^k, b_i^k]$, $R(\mathcal{C})$ is defined to be

$$R(\mathcal{C}) = \frac{\max_{i,k}|a_i^k - b_i^k|}{\min_{i,k}|a_i^k - b_i^k|}.$$

$R(\mathcal{C})$ measures the shape of the cubes in a collection. We need to control $R(\mathcal{C})$ in the construction to obtain Holder continuity in **Step 2**.

**Step 1: Define partition.** First, we construct a sequence of partitions that has the property (*) recursively. Let $\mathcal{C}_0 = \{[0,1]^d\}$. Suppose we have defined close cubes collection $\mathcal{C}_n = \{C_{i_1,i_2\cdots,i_n}\}_{i_j=0,\cdots,2^d-1}$, , such that

1. $\mathcal{C}_n$ is obtained by cutting the unit cube $[0,1]^d$ by planes that are paralleled to the coordinate planes.

2. $C_{i_1,i_2\cdots,i_{n-1}} = \bigcup_{i_n} C_{i_1,i_2\cdots,i_n}$ and $\mathrm{diam}(C_{i_1,i_2\cdots,i_{n-1}}) \leqslant 2/3 \cdot \mathrm{diam}(C_{i_1,i_2\cdots,i_n})$.

3. $\mathbb{P}(\partial C_{i_1,i_2\cdots,i_n}) = 0$ and $R(\mathcal{C}_{n+1}) \leqslant (1+2^{-n}) R(\mathcal{C}_n)$.

Next, we construct $\mathcal{C}_{n+1}$ from $\mathcal{C}_n$ that preserves these properties. We refine the division $\mathcal{C}_n = \{C_{i_1,i_2\cdots,i_n}\}_{i_j=0,\cdots,2^d-1}$ by union of hyperplane $P_{i,r} = \{y : y_i = r\}$ to get $\{C_{i_1,i_2\cdots,i_{n+1}}\}$ such that $C_{i_1,i_2\cdots,i_n} = \bigcup_{i_{n+1}} C_{i_1,i_2\cdots,i_{n+1}}$ and $\mathrm{diam}(C_{i_1,i_2\cdots,i_{n-1}}) \leqslant 2/3 \cdot \mathrm{diam}(C_{i_1,i_2\cdots,i_n})$. By property 1, $\mathcal{C}_n$ is obtained by dividing $[0,1]^d$ using

$$P_{1,r_0^1}, \cdots, P_{1,r_{2^n-1}^1}, \cdots, P_{i,r_j^i}, \cdots, P_{d,r_{2^n}^d}, r_0^i = 0, r_{2^n}^i = 1,$$

we can further cut the unit cube using hyperplane $\{P_{i,(r_j^i+r_{j+1}^i)/2)}\}_{i=1,\cdots d,j=1,\cdots 2^n-1}$. However, this construction may not satisfy property 3, $\mathbb{P}(\partial C_{i_1,i_2\cdots,i_{n+1}}) = 0$. We need to perturb each hyperplane to ensure a measure-zero boundary. Since

$$Z_i = \left\{ r \in \mathbb{R} : \mathbb{P}\left(\{x \in [0,1]^d, [x]_i = r\}\right) \neq 0 \right\}$$

is at most countable, we can perturb the hyperplanes a little bit, i.e., $\{P_{i,(r_j^i+r_{j+1}^i)/2+\epsilon_{i,j}}\}_{i=1,\cdots d,j=1,\cdots 2^n-1}$ to ensure property 3. In this way, we can choose $|\epsilon_{i,j}|$ to be sufficiently small such that

$$\mathrm{diam}(C_{i_1,i_2\cdots,i_{n+1}}) \leqslant 2/3 \cdot \mathrm{diam}(C_{i_1,i_2\cdots,i_n}),$$

and

$$\frac{\max_{i,j}\left(\left(r_{j+1}^i - r_j^i\right)/2 + |\epsilon_{i,j}|\right)}{\min_{i,j}\left(\left(r_{j+1}^i - r_j^i\right)/2 - |\epsilon_{i,j}|\right)} \leqslant \frac{\max_{i,j}\left(r_{j+1}^i - r_j^i\right)}{\min_{i,j}\left(r_{j+1}^i - r_j^i\right)} \cdot \left(1 + 2^{-n}\right) \quad .$$

Therefore, $R(\mathcal{C}_{n+1}) \leqslant (1+2^{-n}) R(\mathcal{C}_n)$ and it is easy to see that properties 1,2,3 are satisfied.

**Step 2: Construct Hilbert Curve.** By Lemma 1, there exist an sequence of ordering $\{O_n\}$ of $\{\mathcal{C}_n\}$ that satisfies

- The first $n$ digits of $O_{n+1}(i)$ with $O_n(j)$, $(j-1)\cdot 2^d < i \leqslant j\cdot 2^d$.

- $C_{O_{n+1}(i)} \cap C_{O_{n+1}(i+1)}$ are $d-1$ dimensional cubes.

Let $\mathcal{I}_0 = \{[0,1]\}$, we define $\mathcal{I}_n$ recursively. Suppose we have define interval collection $\mathcal{I}_n = \{\{I_{i_1,i_2\cdots,i_n}\}_{i_j=0,\cdots,2^d-1}\}$ such that $I_{i_1,i_2\cdots,i_{n-1}} = \bigcup_{i_n} I_{i_1,i_2\cdots,i_n}$ and $\mathbb{P}(C_{i_1,i_2\cdots,i_n}) = |I_{i_1,i_2\cdots,i_n}|$. Since $\mathbb{P}(\partial C_{i_1,i_2\cdots,i_{n+1}}) = 0$, we have $\mathbb{P}(C_{i_1,i_2\cdots,i_n}) = \sum_{i_{n+1}} \mathbb{P}(C_{i_1,i_2\cdots,i_{n+1}})$. With the ordering, we divide each $I_{O_n(j)}, 0 \leqslant j \leqslant 2^{nd}-1$ into $2^d$ closed sub-interval $I_{O_{n+1}(i)}, j2^d \leqslant i < (j+1)2^d$ such that $I_{O_{n+1}(i)} = [\sum_{k=1}^{i-1} p_k, \sum_{k=1}^{i} p_k]$, where $p_i = \mathbb{P}(C_{O_{n+1}(i)})$. This construction satisfies $|I_{O_{n+1}(i)}| = \mathbb{P}(C_{O_{n+1}(i)})$. Note that by condition 3 in step 1, $\mathbb{P}$ vanishes at the boundary of the cubes. Therefore,

$$|I_{O_n(i)}| = \mathbb{P}(C_{O_n(i)}) = \sum_{j=(i-1)2^d+1}^{i2^d} \mathbb{P}(C_{O_{n+1}(j)}) = \sum_{j=(i-1)2^d+1}^{i2^d} |I_{O_{n+1}(j)}|. \tag{20}$$

Now, we can construct a piecewise linear function $\gamma_{n+1}(t)$ such that

$$(\gamma_{n+1})_\# \lambda(C_{O_{n+1}(i)}) = \mathbb{P}(C_{O_{n+1}(i)}), (\gamma_{n+1})_\# \lambda(\partial C_{O_{n+1}(i)}) = 0, \forall i. \tag{21}$$

The idea is to construct a piecewise linear curve going through all cubes in $\mathcal{C}_{n+1}$ exactly once and modify its speed. Take $v_0 \in \text{Center}(C_{O_{n+1}(1)})$, $v_i \in \text{Center}(C_{O_{n+1}(i)} \cap C_{O_{n+1}(i+1)})$, $v_{2^{d(n+1)}} \in \text{Center}(C_{O_{n+1}(2^{d(n+1)})})$, where $\text{Center}(\prod_{i=1}^{k}[a_i, b_i]) = ((a_i + b_i)/2)_{i=1,\cdots,k}$ is the center of a cube. $v_i$ is well-defined since $C_{O_{n+1}(i)} \cap C_{O_{n+1}(i+1)}$ are cubes of dimension $d-1$ by Lemma 1. One possible choice of $\gamma_{n+1}(t)$ is

$$\gamma_{n+1}(t) = v_0 + \sum_{i=1}^{2^{(n+1)d}} \frac{1}{p_i}(v_i - v_{i-1})\left(\left(t - \sum_{k=1}^{i-1}p_k\right)^+ - \left(t - \sum_{k=1}^{i}p_k\right)^+\right). \quad (22)$$

The first two curves $\gamma_1, \gamma_2$ are shown in Figure 9. It is straightforward to verify that $\gamma_{n+1}(\sum_{k=1}^{i}p_k) = v_i$ and

$$\gamma_{n+1}(I_{O_{n+1}(i)}) = \gamma_{n+1}([\sum_{k=1}^{i-1}p_k, \sum_{k=1}^{i}p_k]) \subset C_{O_{n+1}(i)}.$$

In fact, since $v_{i-1}, v_i$ are on different surfaces of the cube $C_{O_{n+1}(i)}$, the line segment between $v_{i-1}, v_i$ lies inside the interior of $C_{O_{n+1}(i)}$ and $\gamma_{n+1}$ goes through all the cubes $C_{O_{n+1}(i)}$ exactly once. We have $\gamma_{n+1}^{-1}(\text{Int}(C_{O_{n+1}(i)})) \subset [\sum_{k=1}^{i-1}p_k, \sum_{k=1}^{i}p_k]$ and $(\gamma_{n+1})_{\#}\lambda(C_{O_{n+1}(i)}) = \mathbb{P}(C_{O_{n+1}(i)})$, where $\text{Int}(\cdot)$ is the interior of a set. Besides, we also have $(\gamma_{n+1})_{\#}\lambda(\partial C_{O_{n+1}(i)}) \leqslant \lambda(\gamma_{n+1}^{-1}(\{v_j\}_{j=1\ldots 2^{d(n+1)}})) = 0$.

Finally, for any $k_1 \geqslant k_2 \geqslant n, t \in [0, 1]$, by property 2 in step 1, (20) and (22), $\gamma_{k_1}(t), \gamma_{k_2}(t)$ are in one cube $C_{i_1, i_2\cdots, i_n}$ and $\text{diam}(C_{i_1, i_2\cdots, i_n}) \leqslant \left(\frac{2}{3}\right)^n \sqrt{d}$. Thus, $|\gamma_{k_1}(t) - \gamma_{k_2}(t)| \leqslant \left(\frac{2}{3}\right)^n \sqrt{d}$ which implies $\{\gamma_n(t)\}$ converges uniformly to one continuous function $\gamma(t)$.

**Step 3: Verify Conclusion** $\mathbb{P} = \gamma_{\#}\lambda$. By $W((\gamma_k)_{\#}\lambda, \gamma_{\#}\lambda) \leqslant \|\gamma_k - \gamma\|_{\infty} \to 0$, $(\gamma_k)_{\#}\lambda$ converges weakly to $\gamma_{\#}\lambda$. By construction, $\mathbb{P}(\partial C_{i_1, i_2\cdots, i_n}) = 0$ and $(\gamma_n)_{\#}\lambda(C_{i_1, i_2\cdots, i_n}) = \mathbb{P}(C_{i_1, i_2\cdots, i_n})$. Therefore, by condition 2 in step 1, for any $k \geqslant n$,

$$\begin{aligned}
(\gamma_k)_{\#}\lambda(C_{i_1, i_2\cdots, i_n}) &= (\gamma_k)_{\#}\lambda\left(\bigcup_{i_{n+1},\cdots,i_k} C_{i_1, i_2\cdots, i_k}\right) \\
&= \sum_{i_{n+1},\cdots,i_k} (\gamma_k)_{\#}\lambda(C_{i_1, i_2\cdots, i_k}) \\
&= \sum_{i_{n+1},\cdots,i_k} \lambda\left(\gamma_k^{-1}(C_{i_1, i_2\cdots, i_k})\right) \\
&= \sum_{i_{n+1},\cdots,i_k} \mathbb{P}(C_{i_1, i_2\cdots, i_k}) = \mathbb{P}(C_{i_1, i_2\cdots, i_n}), \quad (23)
\end{aligned}$$

where we use $\mathbb{P}(\partial C_{i_1, i_2\cdots, i_n}) = 0$ and $(\gamma_k)_{\#}\lambda(\partial C_{i_1, i_2\cdots, i_n}) = 0$ in the second and the last equation. We claim that $\gamma_{\#}\lambda(\partial C_{i_1, i_2\cdots, i_n}) = 0$. For $k > n$, we define

$$B_k = \bigcup_{\partial C_{i_1, i_2\cdots, i_k} \cap \partial C_{i_1, i_2\cdots, i_n} \neq \emptyset} C_{i_1, i_2\cdots, i_k}.$$

Let $k > k_1 \geqslant n$ and $B_n^{\epsilon} = \{x : d(x, \partial C_{i_1, i_2\cdots, i_n}) < \epsilon\}$, we have

$$(\gamma_k)_{\#}\lambda\left(\text{Int}(B_{k_1})\right) \leqslant (\gamma_k)_{\#}\lambda(B_{k_1}) = \mathbb{P}(B_{k_1}) \leqslant \mathbb{P}\left(B_n^{(2/3)^{k_1}\sqrt{d}}\right), \quad \forall k > k_1,$$

where we use (23) and the fact that $(\gamma_k)_{\#}\lambda$ vanishes on $\partial C_{i_1, i_2\cdots, i_k}$ in the first equality and $\text{diam}(C_{i_1, i_2\cdots, i_{k_1}}) \leqslant \left(\frac{2}{3}\right)^{k_1} \sqrt{d}$ in the last inequality. Let $k \to \infty$, by Portmanteau Theorem, we get

$$\gamma_{\#}\lambda(\partial C_{i_1, i_2\cdots, i_n}) \leqslant \gamma_{\#}\lambda\left(\text{Int}(B_{k_1})\right) \leqslant \liminf_{k\to\infty}(\gamma_k)_{\#}\lambda(B_{k_1}) \leqslant \liminf_{k\to\infty}\mathbb{P}\left(B_n^{(2/3)^{k_1}\sqrt{d}}\right) = \mathbb{P}\left(B_n^{(2/3)^{k_1}\sqrt{d}}\right).$$

Since $k_1$ is arbitrary, let $k_1 \to \infty$,

$$\gamma_{\#}\lambda(\partial C_{i_1, i_2\cdots, i_n}) \leqslant \lim_{k_1\to\infty} \mathbb{P}\left(B_n^{(2/3)^{k_1}\sqrt{d}}\right) = \mathbb{P}(\partial C_{i_1, i_2\cdots, i_n}) = 0$$

and we conclude that $\gamma_{\#}\lambda(\partial C_{i_1,i_2\cdots,i_n}) = 0$. By Portmanteau Theorem, let $k \to \infty$ in (23), we obtain $\gamma_{\#}\lambda(C_{i_1,i_2\cdots,i_n}) = \mathbb{P}(C_{i_1,i_2\cdots,i_n})$. Let $\mathcal{C}^* = \bigcup_{i=1}^{\infty} \mathcal{C}_i$. Notice that for any open set $U \in [0,1]^d$, $U = \bigcup_{C \subset U, C \in \mathcal{C}^*} C$, which means the $\sigma$-algebra generated by $\mathcal{C}^*$ contains all Borel sets. Besides, $\mathcal{C}^*$ is a $\pi$-system. Hence, $\lambda\left(\gamma^{-1}(U)\right) = \mathbb{P}(U)$ for all Lebesgue measurable set $U$.

**Step 4: Holder Continuity of $\gamma$.** We verify the condition of [47, Theorem 1]. $\mathcal{I}_n$ and $\mathcal{C}_n$ in the above construction correspond to developments $\alpha_k, \beta_k$ in [47, Theorem 1]. Define $f_k(I_{O_k(i)}) = C_{O_k(i)}$, $i = 1, \cdots, 2^{kd}$ as a map from $\mathcal{I}_n$ to $\mathcal{C}_n$.

1. By construction, if $I_1 \in \mathcal{I}_k, I_2 \in \mathcal{I}_{k-1}, I_1 \subset I_2$, $f_k(I_1) \subset f_{k-1}(I_2)$.

2. If $I_1, I_2 \in \mathcal{I}_k$ and $I_1 \cap I_2 \neq \emptyset$, $I_1, I_2$ are adjacent. By Lemma 1, $f_k(I_1), f_k(I_2)$ share a $(d-1)$-dimensional boundary. Therefore, $f_k(I_1) \cap f_k(I_2) \neq \emptyset$.

3. We have $|\mathcal{C}_{k-1}| \leqslant 2^d |\mathcal{C}_k|$.

4. By property 3,
$$R(\mathcal{C}_n) \leqslant \prod_{k=1}^{n-1} \left(1 + 2^{-k}\right) < \prod_{k=1}^{\infty} \left(1 + 2^{-k}\right) < \infty.$$

   Let $A = \prod_{k=1}^{\infty} \left(1 + 2^{-k}\right)$. We have for any $C \in \mathcal{C}_n$, by Assumption 5
$$\text{diam}^d(C) \leqslant R(\mathcal{C}_n)\lambda(C) \leqslant AC_1^{-1}\mathbb{P}(C) \leqslant AC_1^{-1} \max_{I \in \mathcal{I}} \left(\text{diam}(I)\right),$$

   where $C_1$ is the constant in Assumption 5.

5. Since $\mathcal{I}_k$ consists of one-dimensional intervals, we have $\text{diam}(\mathcal{I}_k) = \text{gap}(\mathcal{I}_k)$.

Therefore, $\gamma$ is $1/d$-Holder continuous. $\qquad \square$

*Proof of Theorem 3.* The proof is similar to the proof of Theorem 2. The difference is that in the first part, we use a deep ReLU network to approximate the Holder continuous curve $\gamma$ instead of wide ReLU network.

Let $\mu_i = \mathbb{P}_{C_i}/\mathbb{P}(C_i)$. By Assumption 4 and Assumption 5, for each connected component $C_i$ of the support, there exists Lipschitz maps $g_2^i \in \mathcal{H}([0,1]^{d_i^C}, C_i) \cap \mathcal{F}_L([0,1]^{d_i^C}, C_i)$ such that
$$(g_2^i)_{\#}^{-1}\mu_i(B) \geqslant C_g\lambda(B)/\mathbb{P}(C_i),$$

for any measurable set $B \subset [0,1]^{d^C}$. By Proposition 1, there exist a $1/d_i^C$-Holder continuous curve $\gamma_i : [0,1] \to [0,1]^{d_i^C}$ such that $(\gamma_i)_{\#}\lambda = (g_2^i)_{\#}^{-1}\mu_i$.

By [53, Theorem 2] (take $\omega(x) = C\|x\|^{1/d}$ where $C$ is the Holder continuity constant factor), there exists deep ReLu network $\hat{g}_1^{i,j}$ of width 12 and depth $L_1$ such that
$$\|\hat{g}_1^{i,j} - (\gamma_i)_j\|_{\infty} \leqslant O\left(L_1^{-2/d_i^C}\right),$$

where $(\gamma_i)_j$ is the $j$-coordinate of $\gamma_i$. Let $\hat{g}_1^i(x) = \left(\hat{g}_1^{i,1}(x), \cdots, \hat{g}_1^{i,d}(x)\right)$, we get
$$\|\hat{g}_1^i - \gamma_i\|_{\infty} \leqslant O\left(L_1^{-2/d_i^C}\right).$$

Thus, the width of $\hat{g}_1^i$ is $\Theta\left(d_i^C\right)$. By Lemma 3
$$W\left((g_2^i)_{\#}^{-1}\mu_i, P(\hat{g}_1^i(U_i))\right) = W\left((\gamma_i)_{\#}U_i, P(\hat{g}_1^i(U_i))\right) \leqslant \|\hat{g}_1^i - \gamma_i\|_{\infty} \leqslant O\left(L_1^{-2/d_i^C}\right).$$

The rest are the same as proof of Theorem 2. $\qquad \square$

*Proof of Corollary 1.* Suppose that structure equations of $\mathcal{M}$ are

$$V_i = \begin{cases} f_i\left(\mathrm{Pa}(V_i), \boldsymbol{U}_{V_i}\right), & V_i \text{ is continuous,} \\ \arg\max_{k\in[n_i]} \left\{ g_k^{V_i} + \log\left(f_i\left(\mathrm{Pa}(V_i), \boldsymbol{U}_{V_i}\right)\right)_k \right\}, & V_i \text{ is categorical,} \|f_i\|_1 = 1 \end{cases}, \quad f_i \in \mathcal{F}_L.$$

Let $d_i^{\mathrm{in}}$ be the input dimension of $f_i$, $d_i^{\mathrm{out}}$ be the output dimension,

$$\hat{f}_i = \begin{cases} \arg\min_{\hat{f}_i \in \mathcal{NN}_{d_i^{\mathrm{in}}, d_i^{\mathrm{out}}}(d_i^{\mathrm{out}}(2d_i^{\mathrm{in}}+10), L_0)} \|\hat{f}_i - f_i\|_\infty, & V_i \text{ continuous,} \\ \max\{0, \arg\min_{\hat{f}_i \in \mathcal{NN}_{d_i^{\mathrm{in}}, d_i^{\mathrm{out}}}(d_i^{\mathrm{out}}(2d_i^{\mathrm{in}}+10), L_0)} \|\hat{f}_i - f_i\|_\infty\}, & V_i \text{ categorical.} \end{cases}$$

We truncate the neural network at 0 for categorical variables because the propensity functions are required to be non-negative. This truncate operation will not influence the approximation error since $f_i$ are non-negative. According to Assumption 3, $f_i$ are Lipschitz continuous. By [53, Theorem 2], we have, if $V_i$ is continuous, $\|\hat{f}_i - f_i\|_\infty \leqslant O(L_0^{-1/d_i^{\mathrm{in}}}) \leqslant O(L_0^{-1/d_{\max}^{\mathrm{in}}})$.

If $V_i$ is categorical, let $S = \|\hat{f}\left(\mathrm{pa}(v_k), \boldsymbol{u}_{V_k}\right) - f\left(\mathrm{pa}(v_k), \boldsymbol{u}_{V_k}\right)\|_\infty$, $\boldsymbol{p} = f\left(\mathrm{pa}(v_k), \boldsymbol{u}_{V_k}\right)$, $\hat{\boldsymbol{p}} = \hat{f}\left(\mathrm{pa}(v_k), \boldsymbol{u}_{V_k}\right)/\|\hat{f}\|_1$. If $S > 1/(2K)$, $\|\boldsymbol{p} - \hat{\boldsymbol{p}}\|_\infty \leqslant 2KS$. If $S \leqslant 1/(2K)$,

$$\|\boldsymbol{p} - \hat{\boldsymbol{p}}\|_\infty \leqslant \frac{1}{\|\hat{f}\|_1} \|\hat{f}\left(\mathrm{pa}(v_k), \boldsymbol{u}_{V_k}\right) - f\left(\mathrm{pa}(v_k), \boldsymbol{u}_{V_k}\right)\|_\infty + \frac{1}{\|\hat{f}\|_1} \|\|\hat{f}\|_1 - 1\|\|f\|_\infty$$

$$\leqslant \frac{(n_i+1)}{\|\hat{f}\|_1} S \leqslant \frac{(n_i+1)}{1 - n_i S} S \leqslant 2(K+1)S$$

where we use Assumption 3 that $n_i \leqslant K$ and $S \leqslant 1/(2K)$. Thus,

$$\|f_i - \hat{f}_i/\|\hat{f}_i\|_1\|_\infty \leqslant O(\|f_i - \hat{f}_i\|_\infty) \leqslant O(L_0^{-1/d_{\max}^{\mathrm{in}}}).$$

By Theorem 3, there exists a neural network $\hat{g}_j$ with architecture in Theorem 3 such that

$$W(P(U_j), P(\hat{g}_j(Z_j))) \leqslant O\left(L_1^{-2/d_j^U} + L_2^{-2/d_j^U} + (\tau - \tau\log\tau)\right),$$

where $Z_j \sim U([0,1]^{N_{C,j}})$ i.i.d., $N_{C,j}$ is the number of connected components of support of $U_j$, $d_j^U$ is the dimension of latent variables $U_j$ and $L_1, L_2, \tau$ are hyperparameters of the neural network defined in Theorem 3. Let $\hat{U}_j = \hat{g}_j(Z_j)$, By Lemma 5,

$$W(P(U_1, \cdots, U_{n_U}), P(\hat{U}_1, \cdots, \hat{U}_{n_U})) \leqslant \sum_{j=1}^{n_U} W(P(U_j), P(\hat{g}_j(Z_j))) \leqslant O\left(L_1^{-2/d_j^U} + L_2^{-2/d_j^U} + (\tau - \tau\log\tau)\right).$$

Let $\hat{\mathcal{M}}$ be the casual model with the following structure equations

$$\hat{V}_i = \begin{cases} \hat{f}_i\left(\mathrm{Pa}(\hat{V}_i), (\hat{g}_j(Z_{C_j}))_{U_{C_j} \in \boldsymbol{U}_{V_i}}\right), & V_i \text{ is continuous,} \\ \arg\max_{k\in[n_i]} \left\{ g_k + \log\left(\hat{f}_i\left(\mathrm{Pa}(\hat{V}_i), (\hat{g}_j(Z_{C_j}))_{U_{C_j} \in \boldsymbol{U}_{V_i}}\right)\right)_k \right\}, & V_i \text{ is categorical.} \end{cases}$$

By Theorem 1, we get

$$W\left(P^{\mathcal{M}^*}(\boldsymbol{V}(t)), P^{\hat{\mathcal{M}}}(\boldsymbol{V}(t))\right) \leqslant O(\sum_{V_i \text{ continuous}} \|f_i - \hat{f}_i\|_\infty + \sum_{V_i \text{ categorical}} \|f_i - \hat{f}_i/\|\hat{f}_i\|_1\|_\infty$$

$$+ W(P(U_1, \cdots, U_{n_U}), P(\hat{U}_1, \cdots, \hat{U}_{n_U})))$$

$$\leqslant O(L_0^{-2/d_{\max}^{\mathrm{in}}} + L_1^{-2/d_{\max}^U} + L_2^{-2/d_{\max}^U} + (\tau - \tau\log\tau)),$$

for any intervention $\boldsymbol{T} = \boldsymbol{t}$. $\qquad\square$

## C  Proof of Consistency

### C.1  Inconsistent Counterexample (Proposition 2)

**Proposition 5.** *There exists a constant $c > 0$ and an SCM $\mathcal{M}^*$ satisfying Assumptions 1-5 with a backdoor causal graph such that there is no unobserved confounding in $\mathcal{M}^*$. For any $\epsilon > 0$, there exists an SCM $\mathcal{M}_\epsilon$ with the same causal graph such that $W(P^{\mathcal{M}^*}(\boldsymbol{V}), P^{\mathcal{M}_\epsilon}(\boldsymbol{V})) \leq \epsilon$ and $|ATE_{\mathcal{M}^*} - ATE_{\mathcal{M}_\epsilon}| > c$.*

*Proof.* Let $\boldsymbol{V} = \{X, T, Y\}$ be the covariate, treatment and outcome respectively. $T$ is a binary variable. Let structure equations of causal model $\mathcal{M}_\delta$ be

$$X = \begin{cases} -1 & ,\text{w.p. } 1/2 \\ 0 & ,\text{w.p. } 1/4 \\ \delta & ,\text{w.p. } 1/4, \end{cases}$$

$$P(T = 1 | X = x) = \begin{cases} 1/2 & , x = -1 \text{ or } 0, \\ 3/4 & , x = \delta, \end{cases}$$

$$Y = \begin{cases} T + U_y & , X < 0 \\ X/\delta + U_y & , X \geqslant 0 \end{cases}$$

where all latent variables are independent and $U_y$ is mean-zero noise. The distribution of $\mathcal{M}_\delta$ is

$$P_{X,T,Y}(0, 1, y) = p_{U_y}(y)/8, \, P_{X,T,Y}(0, 0, y) = p_{U_y}(y)/8$$
$$P_{X,T,Y}(\delta, 1, y) = 3p_{U_y}(y - 1)/16, \, P_{X,T,Y}(\delta, 0, y) = p_{U_y}(y - 1)/16.$$
$$P_{X,T,Y}(-1, 1, y) = p_{U_y}(y - 1)/4, P_{X,T,Y}(-1, 0, y) = p_{U_y}(y)/4.$$

As $\delta \to 0$, distribution of $\mathcal{M}_\delta \xrightarrow{d} \mathcal{M}^*$, where structure equations of $\mathcal{M}^*$ are

$$P(U_1 = 1) = 3/5, P(U_1 = 0) = 2/5, P(U_2 = 1) = 1/3, P(U_1 = 0) = 2/3,$$

$$X = \begin{cases} -1 & ,\text{w.p. } 1/2, \\ 0 & ,\text{w.p. } 1/2, \end{cases}$$

$$P(T = 1 | X = x) = \begin{cases} 1/2 & , x = -1, \\ 5/8 & , x = 0, \end{cases}$$

$$Y = \begin{cases} T + U_y & , X = -1, \\ U_1 + U_y & , X = 0, T = 1, \\ U_2 + U_y & , X = 0, T = 0, \end{cases}$$

where $U_1, U_2, U_y$ are independent. The distribution of $\mathcal{M}^*$ is

$$P_{X,T,Y}(0, 1, y) = p_{U_y}(y)/8 + 3p_{U_y}(y - 1)/16,$$
$$P_{X,T,Y}(0, 0, y) = p_{U_y}(y)/8 + p_{U_y}(y - 1)/16,$$
$$P_{X,T,Y}(-1, 1, y) = p_{U_y}(y - 1)/4,$$
$$P_{X,T,Y}(-1, 0, y) = p_{U_y}(y)/4.$$

It is easy to see that $\mathcal{M}^*$ satisfies Assumption 1-5. Some calculation gives $W(P^{\mathcal{M}^*}(\boldsymbol{V}), P^{\mathcal{M}_\delta}(\boldsymbol{V})) \leqslant \delta/2$. It is easy to calculate the ATE of the two models.

$$\text{ATE}_{\mathcal{M}^*} = 19/30, \text{ATE}_{\mathcal{M}_\delta} = 1/2.$$

$\square$

This example implies that even as the Wasserstein distance between $P^{\mathcal{M}^*}(\boldsymbol{V}), P^{\mathcal{M}_\delta}(\boldsymbol{V})$ converges to zero, their ATEs do not change. As mentioned in the main body, this problem is caused by the violation of Lipschitz continuity assumption for $\mathcal{M}_\delta$. Note that in $\mathcal{M}_\delta$, $\mathbb{E}[Y | X, T] = X/\delta$. As $\delta \to 0$, the Lipschitz constant explodes.

This problem may also arise in (6). If the distribution ball $B_n = \{\hat{\mathcal{M}} : W(P^{\hat{\mathcal{M}}}, P_n^{\mathcal{M}^*}) \leqslant \alpha_n\}$ includes a small ball around the true distribution $S_\epsilon = \{\hat{\mathcal{M}} : W(P^{\hat{\mathcal{M}}}, P^{\mathcal{M}^*}) \leqslant \epsilon\} \subset B_n$ and the NCM is expressive enough to approximate all the SCMs in the $S_\epsilon$, this example tells us the confidence interval may not shrink to one point as sample size increases to infinity even if the model is identifiable.

## C.2 Proof of Theorem 4

To prove Theorem 4, we will need the following proposition.

**Proposition 6.** *Given a metric space $(M, d)$ and sets $\Theta_1 \subset \Theta_2 \cdots \subset \Theta_n \subset \cdots \subset \Theta_\infty \subset M$, where $\Theta_\infty = \overline{\cup_{n=1}^\infty \Theta_n}$, positive sequences $\{\epsilon_n\}_{n\in\mathbb{N}}, \{\delta_n\}_{n\in\mathbb{N}}$ such that $\lim_{n\to\infty} \epsilon_n = \lim_{n\to\infty} \tau_n = \lim_{n\to\infty} \delta_n = 0$ and continuous functions $f, g_n : \Theta_\infty \to \mathbb{R}$, suppose that*

1. *$\Theta_\infty$ is compact.*

2. *$g_n$ satisfies*
$$\|g_n(\theta_1) - g_n(\theta_2)\| \leqslant L_g d(\theta_1, \theta_2) + \tau_n, \quad \forall \theta_1, \theta_2 \in \Theta \tag{24}$$
   *and $\sup_{\theta \in \Theta_\infty} \|g_n(\theta) - g(\theta)\| \leqslant \delta_n, g_n \geqslant 0$.*

3. *There exists a compact subset $\tilde{\Theta}_\infty \subset \Theta_\infty$ such that for any $\theta_0 \in \tilde{\Theta}_\infty$, there exists $\theta \in \Theta_n$ such that $d(\theta, \theta_0) \leqslant \epsilon_n$.*

*Consider the following optimization problems:*

$$\min_{\theta \in \Theta_n} f(\theta),$$
$$s.t. \ g_n(\theta) \leqslant L_g \epsilon_n + \delta_n, \tag{25}$$

$$\min_{\theta \in \Theta_\infty} f(\theta),$$
$$s.t. \ g(\theta) = 0. \tag{26}$$

*and*

$$\min_{\theta \in \tilde{\Theta}_\infty} f(\theta),$$
$$s.t. \ g(\theta) = 0. \tag{27}$$

*We assume that the feasible region of (26) and (27) are nonempty. Let $f_n^*, f^*, \tilde{f}^*$ be the minimal of (25), (26) and (27) respectively. Then, $[\liminf_{k\to\infty} f_n^*, \limsup_{k\to\infty} f_n^*] \subset [f^*, \tilde{f}^*]$.*

*Proof of Proposition 6.* By compactness of $\Theta_\infty$ and $\tilde{\Theta}_\infty$, the minimizers of (26) and (27) are achievable insider these two sets. Let $\theta^* \in \Theta_\infty, \tilde{\theta}^* \in \tilde{\Theta}_\infty$ be the minimizer of (26) and (27). We first prove that $\limsup_{n\to\infty} f_n^* \leqslant \tilde{f}^*$. Note that by (24), the limiting function $g$ is $L_g$-Lipschitz. By condition 3, there exist $\theta_n \in \Theta_n$ such that $d\left(\theta_n, \tilde{\theta}^*\right) \leqslant \epsilon_n$. Note that

$$g_n(\theta_n) \leqslant |g(\theta_n) - g_n(\theta_n)| + |g(\theta_n) - g(\theta^*)| + |g(\theta^*)|$$
$$\leqslant \delta_n + L_g d(\theta_n, \theta^*) \leqslant \delta_n + L_g \epsilon_n.$$

Therefore, $\theta_n$ is a feasible point of (25). We have $\limsup_{n\to\infty} f_n^* \leqslant \limsup_{n\to\infty} f(\theta_n) = f\left(\tilde{\theta}^*\right) = \tilde{f}^*$.

Next, we argue that $\liminf_{n\to\infty} f_n^* \geqslant f^*$. If this equation does not hold, there exists $\epsilon > 0$ and subsequence $\{f_{n_k}^*\}_{k\in\mathbb{N}}$ such that $f_{n_k}^* < f^* - \epsilon, \forall k \in \mathbb{N}$. By compactness of $\Theta_\infty$, for each $k$, there exists a subsequence of $\theta_{n_k}^* \in \Theta_\infty$ such that $\theta_{n_k}^*$ satisfies constraint of (25) and $f_{n_k}^* = f\left(\theta_{n_k}^*\right)$. By compactness, $\{\theta_{n_k}^*\}_{k\in\mathbb{N}}$ has a converging subseqence. Without loss of generality, we may assume that $\{\theta_{n_k}^*\}_{k\in\mathbb{N}}$ converges to $\hat{\theta}^* \in \Theta_\infty$. Since $g_k$ converge uniformly to $g$,

$$\lim_{k\to\infty} g_k\left(\theta_{n_k}^*\right) \leqslant \lim_{k\to\infty} |g_k\left(\theta_{n_k}^*\right) - g\left(\theta_{n_k}^*\right)| + g\left(\theta_{n_k}^*\right) = g\left(\hat{\theta}^*\right) \leqslant \lim_{k\to\infty} \alpha_k = 0.$$

$\hat{\theta}^*$ is a feasible point of Equation (26). Since $f$ is continuous on $\Theta$,

$$f\left(\hat{\theta}^*\right) = \lim_{k\to\infty} f\left(\theta_{n_k}^*\right) \leqslant f^* - \epsilon.$$

which leads to contradiction. $\square$

*Proof of Theorem 4.* We begin by defining a proper metric space. By Assumption 3, all random variables are bounded. Suppose that $\max_{i,j}\{\|V_i\|_\infty, \|U_j\|_\infty\} \leqslant K$. A canonical causal model $\mathcal{M} \in \mathcal{M}(\mathcal{G}, \mathcal{F}, \boldsymbol{U})$ is decided by

$$\theta_\mathcal{M} = (f_1, \cdots, f_{n_V}, P(U_1), \cdots, P(U_{n_U})),$$

where $f_i$ are functions in the structure equations (3) and $U_j$ are uniform parts of the latent variables. We denote $\mathcal{M}^\theta$ to be the SCM represented by $\theta$, the underlying SCM be $\mathcal{M}^{\theta^*}$ and $P^\theta$ to be the distribution of $\mathcal{M}^\theta$. We consider the space

$$M = \mathcal{F}_{V_1} \times \cdots \times \mathcal{F}_{V_{n_V}} \times \mathcal{P}\left([-K,K]^{d_1^U}\right) \cdots \times \mathcal{P}\left([-K,K]^{d_{n_U}^U}\right),$$

where

$$\mathcal{F}_{V_i} = \begin{cases} \mathcal{F}_L^K\left([-K,K]^{d_{i,\text{in}}^V}, [-K,K]^{d_{i,\text{out}}^V}\right), & V_i \text{ continuous}, \\ \{f : \|f\|_1 = 1, f \in \mathcal{F}_L^K\left([-K,K]^{d_{i,\text{in}}^V}, [-K,K]^{d_{i,\text{out}}^V}\right)\} & V_i \text{ categorical}, \end{cases}$$

$d_{i,\text{in}}^V, d_{i,\text{out}}^V$ are the input and output dimensions of $f_i$,

$$\mathcal{F}_L^K = \{f : \|f\|_\infty \leqslant K, f \text{ Lipschitz continuous}\}$$

and $\mathcal{P}(K)$ is the probability space on $K$. For $\theta = (f_1, \cdots, f_{n_V}, P(U_1), \cdots, P(U_{n_U})), \theta' = (f'_1, \cdots, f'_{n_V}, P(U'_1), \cdots, P(U'_{n_U}))$, we define a metric on $M$

$$d(\theta, \theta') = \sum_{k=1}^{n_V} \|f_k - f'_k\|_\infty + \sum_{k=1}^{n_U} W(P(U_k), P(U'_k)).$$

Theorem 1 states that the Wasserstein distance between two causal models is Lipschitz with respect to metric $d$. Now, we define $\Theta_n$. Let

$$\mathcal{P}_n = \left\{ P(\boldsymbol{U}) : \boldsymbol{U} \text{ is the latent distribution of } \hat{\mathcal{M}} \in \text{NCM}_\mathcal{G}(\mathcal{F}_{0,n}, \mathcal{F}_{1,n}, \mathcal{F}_{2,n}, \tau_n) \right\}.$$

In other words, $\mathcal{P}_n$ contains all the push-forward measures of the uniform distribution by neural networks. We denote $\Theta_n = \hat{\mathcal{F}}_{V_1,n} \times \cdots \times \hat{\mathcal{F}}_{V_{n_V},n} \times \times \mathcal{P}_n$, where

$$\hat{\mathcal{F}}_{V_i,n} = \begin{cases} \mathcal{F}_{0,n}, & V_i \text{ continuous}, \\ \{f/\|f\|_1 : f \in \mathcal{F}_{0,n}\} & V_i \text{ categorical}, \end{cases}$$

and $\mathcal{F}_{0,n}$ is defined in Theorem 4. Note that by construction, latent variables are independent and $\mathcal{P}_n$ can be decomposed into direct produce $\mathcal{P}_{n,1} \times \cdots \times \mathcal{P}_{n,n_U}$. Let

$$\Theta_\infty = \overline{\cup_{n=1}^\infty \Theta_n}, g_n(\theta) = S_{\lambda_n}\left(P_n^{\theta^*}(\boldsymbol{V}), P_{m_n}^\theta(\boldsymbol{V})\right), f(\theta) = \mathbb{E}_{t \sim \mu_T}\mathbb{E}_{\mathcal{M}^\theta}[F(V_1(t), \cdots, V_{n_V}(t))]$$

and $\tilde{\Theta}_\infty = \{\theta^*\}$. Note that $f(\theta)$ is a continuous function since by Theorem 1 and the fact that $F$ is Lispchitz continuous,

$$|f(\theta) - f(\theta')| \leqslant \mathbb{E}_{t \sim \mu_T}|W(P^\theta(\boldsymbol{V}(t)), P^{\theta'}(\boldsymbol{V}(t))|$$
$$\leqslant \mathbb{E}_{t \sim \mu_T}[O(d(\theta, \theta'))] = O(d(\theta, \theta')).$$

Now, we verify the conditions in Proposition 6.

1. By Arzelà–Ascoli theorem, $\mathcal{F}_L^K\left([-K,K]^{d_{i,\text{in}}^V}, [-K,K]^{d_{i,\text{out}}^V}\right)$ are precompact set with respect to the infinity norm in space of continuous functions. And thus $\mathcal{F}_{V_i}$ are compact sets.

   Since measures in $\mathcal{P}\left([-K,K]^{d_j^U}\right)$ are tight , $\mathcal{P}\left([-K,K]^{d_j^U}\right)$ are compact with respect to weak topology by Prokhorov's theorem. And the Wasserstein distance metricizes the weak topology. Thus, $\mathcal{P}\left([-K,K]^{d_j^U}\right)$ are compact and the space $(M, d)$ is compact space. Closed set $\Theta_\infty \subset M$ is also compact.

2. By definition of $g_n$,

$$|g_n(\theta) - g_n(\theta')| = |S_{\lambda_n}\left(P_n^{\theta^*}(\boldsymbol{V}), P_{m_n}^{\theta}(\boldsymbol{V})\right) - S_{\lambda_n}\left(P_n^{\theta^*}(\boldsymbol{V}), P_{m_n}^{\theta'}(\boldsymbol{V})\right)|$$
$$\leqslant |W\left(P_n^{\theta^*}(\boldsymbol{V}), P_{m_n}^{\theta}(\boldsymbol{V})\right) - W\left(P_n^{\theta^*}(\boldsymbol{V}), P_{m_n}^{\theta'}(\boldsymbol{V})\right)| + 4(\log(mn)+1)\lambda_n$$
$$\leqslant W\left(P_{m_n}^{\theta}(\boldsymbol{V}), P_{m_n}^{\theta'}(\boldsymbol{V})\right) + 4(\log(mn)+1)\lambda_n$$
$$\leqslant O(d(\theta,\theta')) + 4(1+\log(nm_n))\lambda_n,$$

where we use Lemma 7 in the second inequality and triangle inequality and Theorem 1 in the third inequality. By the condition in Theorem 4, the second term $(1+\log(nm_n))\lambda_n \to 0$ as $n \to \infty$. Therefore, (24) is verified with $\tau_n = 4(1+\log(nm_n))\lambda_n$.

We then verify the uniform convergence of $g_n$. By triangle inequality,

$$|g_n(\theta) - g(\theta)| = |S_{\lambda_n}\left(P_n^{\theta^*}(\boldsymbol{V}), P_{m_n}^{\theta}(\boldsymbol{V})\right) - W\left(P^{\theta^*}(\boldsymbol{V}), P^{\theta}(\boldsymbol{V})\right)|$$
$$\leqslant |S_{\lambda_n}\left(P_n^{\theta^*}(\boldsymbol{V}), P_{m_n}^{\theta}(\boldsymbol{V})\right) - W\left(P_n^{\theta^*}(\boldsymbol{V}), P_{m_n}^{\theta}(\boldsymbol{V})\right)|$$
$$+ |W\left(P_n^{\theta^*}(\boldsymbol{V}), P_{m_n}^{\theta}(\boldsymbol{V})\right) - W\left(P^{\theta^*}(\boldsymbol{V}), P^{\theta}(\boldsymbol{V})\right)|.$$

By Lemma 7, $|S_{\lambda_n}\left(P_n^{\theta^*}(\boldsymbol{V}), P_{m_n}^{\theta}(\boldsymbol{V})\right) - W\left(P_n^{\theta^*}(\boldsymbol{V}), P_{m_n}^{\theta}(\boldsymbol{V})\right)| \leqslant 2(\log(mn)+1)\lambda_n$ and we get

$$|g_n(\theta) - g(\theta)| \leqslant 2(\log(mn)+1)\lambda_n + |W\left(P_n^{\theta^*}(\boldsymbol{V}), P_{m_n}^{\theta}(\boldsymbol{V})\right) - W\left(P_n^{\theta^*}(\boldsymbol{V}), P^{\theta}(\boldsymbol{V})\right)|$$
$$+ |W\left(P_n^{\theta^*}(\boldsymbol{V}), P^{\theta}(\boldsymbol{V})\right) - W\left(P^{\theta^*}(\boldsymbol{V}), P^{\theta}(\boldsymbol{V})\right)|$$
$$\leqslant 2(\log(mn)+1)\lambda_n + W\left(P_{m_n}^{\theta}(\boldsymbol{V}), P^{\theta}(\boldsymbol{V})\right) + W(P_n^{\theta^*}(\boldsymbol{V}), P^{\theta^*}(\boldsymbol{V}))$$
$$\leqslant 2(\log(mn)+1)\lambda_n + O\left(W(P_n^{\theta^*}(\boldsymbol{U}), P^{\theta^*}(\boldsymbol{U})) + W\left(P^{\theta}(\boldsymbol{U}), P_{m_n}^{\theta}(\boldsymbol{U})\right)\right),$$
(28)

where we use Theorem 1 in the last inequality.

We first bound $\sup_\theta W\left(P^{\theta}(\boldsymbol{U}), P_{m_n}^{\theta}(\boldsymbol{U})\right)$ using standard VC dimension argument. Note that $\{U_1, \cdots, U_{n_U}\}$ are independent. By Lemma 5,

$$W\left(P^{\theta}(\boldsymbol{U}), P_{m_n}^{\theta}(\boldsymbol{U})\right) \leqslant \sum_{i=1}^{n_U} W\left(P^{\theta}(U_i), P_{m_n}^{\theta}(U_i)\right).$$
(29)

By the construction in Theorem 3, $P^{\theta}(U_i) \sim \sum_{j\in[N_{C,i}]} p_j P(f_{\theta_{i,j}}(Z_{i,j})), Z_{i,j} \sim$ Unif$(0,1)$, where $f_{\theta_{i,j}}$ are neural networks with constant width and depth $L_{1,n}+L_{2,n}$, $N_{C,i}$ is the number of connected components of supp$(P(U_i))$ and $\theta_{i,j}$ are the parameters of the neural networks. By Lemma 4,

$$\sup_\theta W\left(P^{\theta}(U_i), P_{m_n}^{\theta}(U_i)\right) \leqslant O(\max_{j\in[N_{C,j}]} \sup_{\theta_{i,j}} W(P(f_{\theta_{i,j}}(Z_{i,j})), P_{m_n}(f_{\theta_{i,j}}(Z_{i,j}))).$$

By [6], the pseudo-dimension of $\mathcal{NN}(\Theta(1), L)$ is $O\left(L^2 \log L\right)$. By boundness of all the neural networks and Lemma 8 [2], with probability at least

$$1 - \exp\left(O\left((\epsilon_n')^{-d_{\max}^U}\log(\epsilon_n')^{-1} + (L_{n,1}+L_{n,2})^2\log(L_{n,1}+L_{n,2})\log\epsilon_n^{-1} - m_n\epsilon_n^2\right)\right),$$

the following event happens.

$$\sup_{\theta_{i,j}} W(P(f_{\theta_{i,j}}(Z_{i,j})), P_{m_n}(f_{\theta_{i,j}}(Z_{i,j})) \leqslant \epsilon_n + \epsilon_n'.$$

---

[2]Note that we truncate the neural network to ensure boundness in Theorem 4. This extra truncate operation $\tilde{f} = \max\{-K, \min\{K, f\}\}$ can be viewed as two extra ReLu layers, so Lemma 8 is still applicable.

Let

$$\epsilon'_n = m_n^{-1/\left(d_{\max}^U+2\right)} \log m_n, \quad L_{n,i} = m_n^{d_{\max}^U/\left(2d_{\max}^U+4\right)} \log m_n, \quad \epsilon_n = Cm_n^{-1/\left(d_{\max}^U+2\right)} \log m_n$$

with constant $C > 0$ sufficiently large, we get

$$P\left(\sup_{\theta_{i,j}} W(P(f_{\theta_{i,j}}(Z_{i,j})), P_{m_n}(f_{\theta_{i,j}}(Z_{i,j}))) \leqslant O\left(m_n^{-1/\left(d_{\max}^U+2\right)} \log m_n\right)\right) \geqslant 1 - O\left(m_n^{-2}\right).$$

As long as $m_n = \Omega(n)$, the Borel-Cantelli lemma implies that almost surely, there exists $N > 0$ such that when $n > N$, $\sup_{\theta_{i,j}} W(P(f_{\theta_{i,j}}(Z_{i,j})), P_{m_n}(f_{\theta_{i,j}}(Z_{i,j}))) \leqslant O\left(m_n^{-1/\left(d_{\max}^U+2\right)} \log m_n\right)$ for all $i, j$. Therefore, almost surely, for sufficiently large $n$,

$$\sup_{\theta} W\left(P^{\theta}(U_i), P_{m_n}^{\theta}(U_i)\right) \leqslant O\left(m_n^{-1/\left(d_{\max}^U+2\right)} \log m_n\right), \tag{30}$$

[17, Theorem 2] shows that for sufficiently large $n$, if $\delta_n = C_1 n^{-1/\max\{d_{\max}^U, 2\}} \log^2(n)$, where $C_1 > 0$ is a constant,

$$P(W(P^{\theta^*}(U_i), P_n^{\theta^*}(U_i)) > \delta_n) \leqslant \begin{cases} \exp(-CC_1 \log^2(n)) & d_{\max}^U = 1, \\ \exp\left(\dfrac{-CC_1 \log^4(n)}{\log^2\left(2+C_1^{-1}n^{1/2}\log^{-2}(n)\right)}\right) & d_{\max}^U = 2, \\ \exp(-CC_1 \log^{2/d_{\max}^U}(n)) & d_{\max}^U > 2, \end{cases}$$

where $C > 0$ is a constant. Let $C_1$ sufficiently large such that

$$P(W(P^{\theta^*}(U_i), P_n^{\theta^*}(U_i)) > \delta_n) \leqslant O\left(n^{-2}\right).$$

Therefore, $\sum_{n=1}^{\infty} P(W(P^{\theta^*}(U_i), P_n^{\theta^*}(U_i)) > \delta_n) < \infty$. By Borel-Cantelli lemma, with probability 1, there exist $N > 0$ such that $W(P^{\theta^*}(U_i), P_n^{\theta^*}(U_i)) \leqslant \delta_n, \forall n > N$. By Equation (29), with probability 1, there exist $N > 0$ such that

$$W(P^{\theta^*}(\boldsymbol{U}), P_n^{\theta^*}(\boldsymbol{U})) \leqslant O(\delta_n), \forall n > N. \tag{31}$$

Therefore, by Equations (28), (30) and (31), almost surely, the following inequality holds eventually,

$$\sup_{\theta \in \Theta_{\infty}} |g_n(\theta) - g(\theta)| \leqslant O\left(\log(nm_n)\lambda_n + \delta_n + m_n^{-1/\left(d_{\max}^U+2\right)} \log m_n\right) = s_n,$$

where $s_n$ is defined in Theorem 4.

3. [53, Proposition 1] shows that given a $L$-Lipschitz continuous function $f : [0,1]^m \to \mathbb{R}$, there exist $\hat{f} \in \mathcal{NN}_{m,1}(W, \Theta(\log(m))$ such that $\|f - \hat{f}\|_{\infty} \leqslant O\left(W^{-1/d}\right)$ and $\hat{f}$ is $\sqrt{m}L$-Lipschitz continuous. [3] For a multivariate function with output dimension $m'$, we can approximate each coordinate individually and get a neural network approximation $\hat{f}$ that is $\sqrt{mm'}L$-Lipschitz continuous and $\|f - \hat{f}\|_{\infty} \leqslant O\left(W^{-1/m}\right)$.

Next, we define the truncate operator $T_K f = \min\{K, \max\{-K, f\}\}$, which is a contraction mapping, and prove that applying the truncate operator will not increase approximation error. If $|f| \leqslant K$, we have

$$\|f - T_K \hat{f}\|_{\infty} = \|T_K f - T_K \hat{f}\|_{\infty} \leqslant \|f - \hat{f}\|_{\infty} \leqslant O\left(W^{-1/m}\right).$$

---

[3] The original proof does not specify the depth of the neural network. The authors show that the network architectures can be chosen as consisting of $O(W)$ parallel blocks each having the same architecture. Each block is used to realize the function

$$\phi(x) = (\min(\min_{k \neq s}(x_k - x_s + 1), \min_k(1 + x_k), \min_k(1 - x_k)))_+, \quad x \in \mathbb{R}^m.$$

This function can be realized by feed-forward network with width $O\left(m^2\right)$ and depth $O(\log m)$.

Therefore, we get that

$$\sup_{f\in\mathcal{F}_L^K([-K,K]^m,[-K,K]^{m'})}\inf_{\hat{f}\in\mathcal{NN}_{m,m'}^{\sqrt{mm'}L,K}(W_{0,n},\Theta(\log m))}\|f-\hat{f}\|_\infty\leqslant O\left(W_{0,n}^{-1/m}\right).\quad(32)$$

Theorem 3 implies that for each latent variable $U_i$, there exist a neural network $g_i$ with architecture in Corollary 1 such that

$$W(P(U_i),P(g_i(Z_i)))\leqslant O(\sum_{i=1}^{n}L_{i,n}^{-2/d_{\max}}+\tau_n(1-\log\tau_n)).$$

By Assumption 3, we know $\|U_i\|_\infty\leqslant K$, which implies

$$W(P(U_i),P((T_Kg_i)(Z_i)))\leqslant W(P(U_i),P(g_i(Z_i)))\leqslant O(\sum_{i=1}^{n}L_{i,n}^{-2/d_{\max}^U}+\tau_n(1-\log\tau_n)).$$
$$(33)$$

Combining Equations (32) and (33) with the same proof as Corollary 1, it can be proven in the same way as Corollary 1 that there exists a $\theta_n\in\Theta_n$ satisfying

$$d(\theta^*,\theta_n)\leqslant\epsilon_n=O\left(W_{0,n}^{-1/d_{\max}^{in}}+L_{1,n}^{-2/d_{\max}^U}+L_{2,n}^{-2/d_{\max}^U}+\tau_n(1-\log\tau_n)\right).$$

Let $\tilde{\Theta}_\infty=\{\theta^*\}$, we have verified the third assumption in Proposition 6.

Take the Wasserstein radius to be $\alpha_n=O(s_n+\epsilon_n)$, Proposition 6 implies the conclusion. □

### C.3 Proof of Proposition 3

**Lemma 2.** *Let $(\hat{T},\hat{Y})\sim\mu,(T,Y)\sim\nu$ and suppose that $f(t)=\mathbb{E}_\nu[Y|T],\hat{f}(t)=\mathbb{E}_\mu[\hat{Y}|\hat{T}]$ are $L$-Lipschitz continuous and $|f(t)|\leqslant K,|\hat{f}(t)|\leqslant K$, then we have*

$$\int(f(t)-\hat{f}(t))^2d\nu(dt)\leqslant C_WW(\mu,\nu),$$

*where $C_W=4LK+2K\max\{L,1\}$.*

*Proof of Lemma 2.* By the duality formulation of Wasserstein-1 distance, we have

$$W(\mu,\nu)=\sup_{g\in\mathrm{Lip}(1)}\mathbb{E}_\mu[g(\hat{T},\hat{Y})]-\mathbb{E}_\nu[g(T,Y)].$$

Let $g_0(t,y)=(\hat{f}(t)-f(t))y$, we verify $g_0$ is a Lipschitz continuous function in $\{(t,y):\|(t,y)\|_\infty\leqslant K\}$.

$$\begin{aligned}|g_0(t_1,y_1)-g_0(t_2,y_2)|&\leqslant|g_0(t_1,y_1)-g_0(t_1,y_2)|+|g_0(t_1,y_2)-g_0(t_2,y_2)|\\&=|(\hat{f}(t_1)-f(t_1))(y_1-y_2)|+|y_2(\hat{f}(t_1)-f(t_1)-(\hat{f}(t_2)-f(t_2)))|\\&\leqslant2K|y_1-y_2|+K(|\hat{f}(t_1)-\hat{f}(t_2)|+|f(t_1)-f(t_2)|)\\&\leqslant2K|y_1-y_2|+2KL|t_1-t_2|.\end{aligned}$$

Let $L_g=2K\max\{L,1\}$, we have proven that $g_0$ is $L_g$-Lipschitz continuous in $\{(t,y):\|(t,y)\|_\infty\leqslant K\}$. Thus,

$$\begin{aligned}W(\mu,\nu)&\geqslant\frac{1}{L_g}(\mathbb{E}_\mu[g_0(\hat{T},\hat{Y})]-\mathbb{E}_\nu[g_0(T,Y)])\\&=\frac{1}{L_g}(\mathbb{E}_\mu[(\hat{f}(\hat{T})-f(\hat{T}))\mathbb{E}[\hat{Y}|\hat{T}=t]]-\mathbb{E}_\nu[(\hat{f}(T)-f(T))\mathbb{E}[Y|T=t]])\\&=\frac{1}{L_g}(\mathbb{E}_\mu[(\hat{f}(\hat{T})-f(\hat{T}))\hat{f}(\hat{T})]-\mathbb{E}_\nu[(\hat{f}(T)-f(T))f(T)]).\end{aligned}\quad(34)$$

Now, let $h(t) = (\hat{f}(t) - f(t))\hat{f}(t)$. Following the same argument, it can be proven that $h(t)$ is Lipschitz continuous with Lipschitz constant being $L_h = 4LK$. Hence,

$$\mathbb{E}_\mu[h(\hat{T})] \geqslant \mathbb{E}_\nu[h(T)] - L_h W(\mu, \nu).$$

Plug into (34), and we get

$$(L_g + L_h)W(\mu, \nu) \geqslant \mathbb{E}_\nu[(\hat{f}(T) - f(T))\hat{f}(T) - (\hat{f}(T) - f(T))f(T)]$$
$$= \mathbb{E}_\nu\left[(\hat{f}(T) - f(T))^2\right].$$

$\square$

*Proof of Proposition 3.* If $Y$ is not descendant of $T$, $\mathbb{E}[Y(t)] = \mathbb{E}[Y]$ and we have

$$\int_{t_1}^{t_2} (\mathbb{E}_\mathcal{M}[Y(t)] - \mathbb{E}_{\hat{\mathcal{M}}}[\hat{Y}(t)])^2 dt = \int_{t_1}^{t_2} (\mathbb{E}_\mathcal{M}[Y] - \mathbb{E}_{\hat{\mathcal{M}}}[\hat{Y}])^2 dt$$
$$= (t_2 - t_1)(\mathbb{E}_\mathcal{M}[Y] - \mathbb{E}_{\hat{\mathcal{M}}}[\hat{Y}])^2 \leqslant (t_2 - t_1)W(P^\mathcal{M}(\boldsymbol{V}), P^{\hat{\mathcal{M}}}(\hat{\boldsymbol{V}}))^2.$$

Now, suppose that $Y$ is a descendant of $T$. Let $X = \text{Pa}(T)$. Note that $f_y(x, t) = \mathbb{E}_\mathcal{M}[Y|X = x, T = t]$ is $L_y$-Lipschitz continuous with Lipschitz constant $L_y \leqslant (L + 1)^{n_V}$. This is because $Y$ can be written as $Y = F_0(X, T, \boldsymbol{U}_y)$ where $\boldsymbol{U}_y$ are latent variables independent of $T, X$ and $F_0$ is composition of $f_i$ in structure equations. The composition of Lipschitz functions is still Lipschitz and $F_0$ is a composition of at most $n_V$ $L$-Lipschitz functions. $F_0$ is $(L + 1)^{n_V}$-Lipschitz and so is $f_y(x, t) = \mathbb{E}_{\boldsymbol{U}_y}[F_0(X, T, \boldsymbol{U}_y)]$. Recall that $\nu, \mu$ are the observation distributions of $\mathcal{M}$ and $\hat{\mathcal{M}}$ respectively in Proposition 3. Similarly, $\hat{f}_y(x, t) = \mathbb{E}_{\hat{\mathcal{M}}}[\hat{Y}|X = x, T = t]$ is $L_y$-Lipschitz continuous. By Lemma 2 and the overlap assumption, we have

$$C_W W(\mu, \nu) \geqslant \mathbb{E}_\nu\left[(f_y(X, T) - \hat{f}_y(X, T))^2\right]$$
$$= \int (f_y(x, t) - \hat{f}_y(x, t))^2 \nu(dt|x)\nu(dx)$$
$$\geqslant \delta \int_{t_1}^{t_2} \int (f_y(x, t) - \hat{f}_y(x, t))^2 \nu(dx)P(dt).$$
$$\geqslant \delta \int_{t_1}^{t_2} \left(\int f_y(x, t)\nu(dx) - \int \hat{f}_y(x, t)\nu(dx)\right)^2 P(dt).$$

Note that $\hat{f}_y(x, t)$ is Lipschitz continuous, we have

$$\left|\int \hat{f}_y(x, t)\nu(dx) - \int \hat{f}_y(x, t)\mu(dx)\right| \leqslant L_y W(\mu, \nu).$$

Therefore,

$$\left(\int f_y(x, t)\nu(dx) - \int \hat{f}_y(x, t)\nu(dx)\right)^2 \geqslant \frac{1}{2}\left(\int f_y(x, t)\nu(dx) - \int \hat{f}_y(x, t)\mu(dx)\right)^2$$
$$- \left(\int \hat{f}_y(x, t)\mu(dx) - \int \hat{f}_y(x, t)\nu(dx)\right)^2$$
$$\geqslant \frac{1}{2}\left(\int f_y(x, t)\nu(dx) - \int \hat{f}_y(x, t)\mu(dx)\right)^2 - L_y^2 W^2(\mu, \nu)$$
$$= \frac{1}{2}(\mathbb{E}_\mathcal{M}[Y(t)] - \mathbb{E}_{\hat{\mathcal{M}}}[\hat{Y}(t)])^2 - L_y^2 W^2(\mu, \nu).$$

Combine all the equations, we get

$$\int_{t_1}^{t_2} (\mathbb{E}_\mathcal{M}[Y(t)] - \mathbb{E}_{\hat{\mathcal{M}}}[\hat{Y}(t)])^2 P(dt) \leqslant \frac{2C_W}{\delta}W(\mu, \nu) + 2L_y^2 W^2(\mu, \nu)(t_2 - t_1).$$

$\square$

*Proof of Corollary 2.* In the proof of Theorem 4, we know with probability at least $1 - O(n^{-2})$, (6) has feasible solutions and

$$W(P_n^{\mathcal{M}^*}(\boldsymbol{V}), P^{\mathcal{M}^*}(\boldsymbol{V})) \leqslant O(\alpha_n), \quad W(P_{m_n}^{\theta_n}(\boldsymbol{V}), P^{\theta_n}(\boldsymbol{V})) \leqslant O(\alpha_n),$$

where notations $\theta$ and $P^\theta$ are defined in the proof of Theorem 4 and $\theta_n$ is one of the minimizers of (6). By Lemma 7, we know that

$$W(P_n^{\mathcal{M}^*}(\boldsymbol{V}), P_{m_n}^{\theta_n}(\boldsymbol{V})) \leqslant S_{\lambda_n}(P_n^{\mathcal{M}^*}(\boldsymbol{V}), P_{m_n}^{\theta_n}(\boldsymbol{V})) + 2(\log(m_n n) + 1)\lambda_n \leqslant O(\alpha_n).$$

We get that

$$W(P^{\mathcal{M}^*}(\boldsymbol{V}), P^{\theta_n}(\boldsymbol{V})) \leqslant W(P_n^{\mathcal{M}^*}(\boldsymbol{V}), P^{\mathcal{M}^*}(\boldsymbol{V})) + W(P^{\mathcal{M}^*}(\boldsymbol{V}), P^{\theta_n}(\boldsymbol{V})) + W(P_{m_n}^{\theta_n}(\boldsymbol{V}), P^{\theta_n}(\boldsymbol{V})) \leqslant O(\alpha_n).$$

By Proposition 3, we get $|F_n - F_*| \leqslant O(\sqrt{\alpha_n})$. $\qquad\square$

## D  Experiments

The structure equations of the generative models are (5). We use three-layer feed-forward neural networks with width 128 for each $\hat{f}_i$ and six-layer neural networks with width 128 for each $\hat{g}_j$. We use the Augmented Lagrangian Multiplier (ALM) method to solve the optimization problems as in [3]. We run 600 epochs and use a batch size of 2048 in each epoch. $m_n$ is set to be $m_n = n$. The "geomloss" package [16] is used to calculate the Sinkhorn distance. To impose Lipschitz regularization, we use the technique from [23] to do layer-wise normalization to the weight matrices with respect to infinity norm. The upper bound of the Lipschitz constant in each layer is set to be 8. The $\tau$ in the Gumbel-softmax layer is set to be 0.

For the choice of Wasserstein ball radius $\alpha_n$, we use the subsampling technique from [11, Section 3.4]. We can take the criterion function in [11] to be $Q(\theta) = W(P^{\mathcal{M}^*}(\boldsymbol{V}), P^{\mathcal{M}^\theta}(\boldsymbol{V}))$ and its empirical estimation to be $Q_n(\theta) = W(P_n^{\mathcal{M}^*}(\boldsymbol{V}), P_n^{\mathcal{M}^\theta}(\boldsymbol{V}))$. In the first step, we minimize the Wasserstein distance $\hat{\theta}_n^* = \arg\min_\theta W(P_n^{\mathcal{M}^*}(\boldsymbol{V}), P_n^{\mathcal{M}^\theta}(\boldsymbol{V}))$. Then, [11] propose to refine the radius $Q_n(\hat{\theta}_n^*)$ by taking the quantile of subsample $\{\sup_{\theta:Q_n(\hat{\theta}) \leqslant \beta Q_n(\hat{\theta}_n^*)} Q_{j,b}(\theta)\}$, where $Q_{j,b}$ denotes the criterion function estimated at $j$-th subsample of size $b$. However, it is time-consuming to solve this optimization problem many times. In practice, we set the radius $\alpha_n$ to be the 95% quantile of $\{W(\bar{P}_j^{\mathcal{M}^*}\}(\boldsymbol{V}), P_n^{\mathcal{M}^{\hat{\theta}_n^*}}(\boldsymbol{V}))$, where we use 50 subsamples $\bar{P}_j^{\mathcal{M}^*}, j = 1, \cdots, 50$ from $P^{\mathcal{M}_n^*(\boldsymbol{V})}$ with size $b = \lfloor 15\sqrt{n} \rfloor$.

### D.1  Discrete IV [14]

We consider the noncompliance binary IV example in [14, Section D.1]. The causal graph is shown in Figure 10. We could see from Table 2 that the bound given by NCM is slightly worse than Autobounds but is still close to the optimal bound. Besides, the Autobounds bound does not cover the optimal bound in this example.

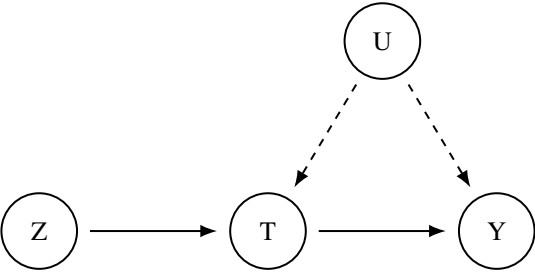

Figure 10: Instrumental Variable (IV) graph. $Z$ is the instrumental variable, $T$ is the treatment and $Y$ is the outcome.

| Algorithm | Average Bound | SD of length | Optimal Bound | True Value |
|---|---|---|---|---|
| NCM (Ours) | [-0.49,0.05] | 0.05 | [-0.45, -0.04] | -0.31 |
| Autobounds ([14]) | [-0.45,-0.05] | 0.02 | [-0.45, -0.04] | -0.31 |

Table 2: The sample size is taken to be 5000. We average over 10 runs with different random seeds. SD is short for the standard derivation.

## D.2 Continuous IV

Now, we turn to the continuous IV setting, where $T$ is still binary, but $Y$ and $Z$ can be continuous. Let $E_i^\lambda \sim \lambda Z_1 + (1 - \lambda)\text{Unif}(-1, 1)$, where $Z_1$ is the Gaussian variable conditioning on $[-1, 1]$ and the structure equations of $\mathcal{M}^\lambda$ to be

$$E_Y = E_1^\lambda, U = E_2^\lambda, Z = E_3^\lambda,$$
$$P(T = 1|Z) = (1.5 + Z + 0.5U)/3,$$
$$Y = 3T - 1.5TU + U + E_Y,$$

where $E_i^\lambda$ are independent. It is easy to see the ATE of this model is 3 regardless of $\lambda$. In the experiment, we randomly choose ten $\lambda \sim \text{Unif}(0, 1)$. For each $\lambda$, we run the algorithms 5 times to get the bound. We choose different latent distributions (indexed by $\lambda$) in the experiments to create more difficulty.

Since Autobounds can only deal with discrete variables, we discretize $Z$ and $Y$. Suppose that $Z \in [l, u]$, we map all points in intervals $[l + i(u - l)/k, l + (i + 1)(u - l)/k], I = 0, 1 \cdots, k - 1$ to the middle point $l + (i + 1/2)(u - l)/k$. We choose $k = 8$ in the following experiments. The choice will give rise to polynomial programming problems with $2^{14}$ decision variables, which is quite large.

Table 3 demonstrate the results. While both algorithms cover the true ATE well, we can see that NCM gives much tighter bounds on average. The main reason may be that the discretized problem does not approximate the original problem well enough. It is possible that a larger discretized parameter $k$ can give a better bound, but since the size of the polynomial problem grows exponentially with $k$, the optimization problem may be intractable for large $k$. On the contrary, NCM does not suffer from computational difficulties.

| Algorithm | Average Bound | SD of length | Success Rate | True Value |
|---|---|---|---|---|
| NCM (Ours) | [2.49,3.24] | 0.49 | 50/50 | 3 |
| Autobounds ([14]) | [1.40, 3.48] | 0.26 | 50/50 | 3 |

Table 3: We take a sample size of 5000. We randomly choose 10 $\lambda \sim \text{Unif}(0, 1)$ and get 10 models $\mathcal{M}^{\lambda_i}$. For each model $\mathcal{M}^{\lambda_i}$, we run the two algorithms for 5 times. The success rate is the number of times when the obtained bounds cover the true ATE divided by the total number of experiments. SD is short for the standard derivation.

## D.3 Counterexample

We test our neural causal method on the counterexample in Appendix C.1. We choose the noise $U_y$ to be the normal variable. The structure equations are

$$P(U_1 = 1) = 3/5, P(U_1 = 0) = 2/5, P(U_2 = 1) = 1/3, P(U_1 = 0) = 2/3,$$

$$X = \begin{cases} -1 & , \text{w.p. } 1/2, \\ 0 & , \text{w.p. } 1/2, \end{cases}$$

$$P(T = 1|X = x) = \begin{cases} 1/2 & , x = -1, \\ 5/8 & , x = 0, \end{cases}$$

$$Y = \begin{cases} T + U_y & , X = -1, \\ U_1 + U_y & , X = 0, T = 1, \\ U_2 + U_y & , X = 0, T = 0, \end{cases}$$

We compare the confidence intervals of the unregularized neural casual method and the Lipschitz regularized one under different architectures. We choose the layerwise Lispchitz constants upper bound to be 5 and 1.2. As a benchmark, we also include the bound obtained by the Double Machine Learning estimator using the EconML package. The result is shown in Appendix D.3. For this identifiable case example, the double ML estimator produces better bounds for the ATE. The intervals given by regularized and unregularized NCM are similar to the regularized one slightly better in the left figure, where we use medium-sized NNs. However, in the right figure, the obtained intervals after regularization are tighter, although slightly biased, compared with the regularized setting. From these two experiments, we conclude that the architecture of NNs will also influence the results and adding regularization during the training process can prevent extreme confidence intervals or inconsistency.

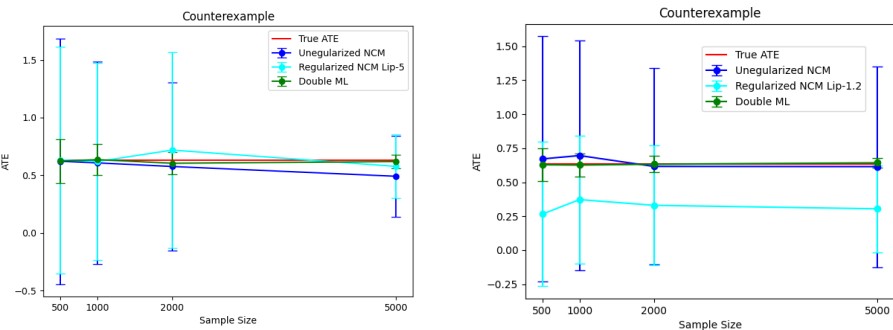

Figure 11: Comparison of Lipschitz regularized and unregularized neural causal algorithm. The two figures show the results of different architectures. The figure on the left side uses a medium-sized NN (width 128, depth 3) to approximate each structural function, while the right figure uses extremely small NNs (width 3, depth 1). In all experiments, we use the projected gradient to regularize the weight of the neural network. For each sample size, we repeat the experiment 5 times and take the average of the upper (lower) bound.

## E    Technical Lemmas

**Lemma 3.** *Let $\mu, \hat{\mu}$ be two measures on compact set $K \subset \mathbb{R}^{d_1}$, for any measurable functions $F, \hat{F} : K \to \mathbb{R}^{d_2}$, if $F$ is $L$-Lipschitz continuous, then*

$$W(F_{\#}\mu, \hat{F}_{\#}\hat{\mu}) \leqslant LW(\mu, \hat{\mu}) + \|F_1 - F_2\|_{\infty}.$$

*Proof.* For any 1-Lipschitz function $g : \mathbb{R}^{d_2} \to \mathbb{R}$,

$$\mathbb{E}_{X \sim F_{\#}\mu}[g(X)] - \mathbb{E}_{X \sim \hat{F}_{\#}\hat{\mu}}[g(X)] = \mathbb{E}_{X \sim \mu}[g \circ F(X)] - \mathbb{E}_{X \sim \hat{\mu}}[g \circ \hat{F}(X)]$$
$$= (\mathbb{E}_{X \sim \mu}[g \circ F(X)] - \mathbb{E}_{X \sim \hat{\mu}}[g \circ F(X)])$$
$$+ (\mathbb{E}_{X \sim \hat{\mu}}[g \circ F(X)] - \mathbb{E}_{X \sim \hat{\mu}}[g \circ \hat{F}(X)]).$$

Note that for any $x, y \in \mathbb{R}^{d_1}$,

$$|g \circ F(x) - g \circ F(y)| \leqslant \|F(x) - F(y)\| \leqslant L\|x - y\|.$$

Thus, $g \circ F$ is $L$-Lipschitz continuous and

$$\mathbb{E}_{X \sim \mu}[g \circ F(X)] - \mathbb{E}_{X \sim \hat{\mu}}[g \circ F(X)] \leqslant LW(\mu, \hat{\mu}). \tag{35}$$

For the second term,

$$\mathbb{E}_{X \sim \hat{\mu}}[g \circ F(X)] - \mathbb{E}_{X \sim \hat{\mu}}[g \circ \hat{F}(X)] \leqslant \mathbb{E}_{X \sim \hat{\mu}}[|g \circ F(X) - g \circ \hat{F}(X)|]$$
$$\leqslant \mathbb{E}_{X \sim \hat{\mu}}[\|F(X) - \hat{F}(X)\|]$$
$$\leqslant \|F(X) - \hat{F}(X)\|_{\infty}. \tag{36}$$

Combine (35) and (36), we have the conclusion. □

**Lemma 4.** *Let* $\mu_1, \cdots, \mu_n, \hat{\mu}_1, \cdots, \hat{\mu}_n$ *be measure on* $\mathbb{R}^d$, *for any* $p_1, \cdots, p_n \in [0, 1]$ *such that* $\sum_{k=1}^n p_k = 1$, *we have*

$$W\left(\sum_{k=1}^n p_k \mu_k, \sum_{k=1}^n p_k \hat{\mu}_k\right) \leqslant \sum_{k=1}^n p_k W(\mu_k, \hat{\mu}_k).$$

*Proof.* For any 1-Lipschitz function $f : \mathbb{R}^d \to \mathbb{R}$, we have

$$\mathbb{E}_{X \sim \sum_{k=1}^n p_k \mu_k}[f(X)] - \mathbb{E}_{X \sim \sum_{k=1}^n p_k \hat{\mu}_k}[f(X)] = \sum_{k=1}^n p_k (\mathbb{E}_{X_k \sim \mu_k}[f(X_k)] - \mathbb{E}_{X_k \sim \hat{\mu}_k}[f(X_k)])$$

$$\leqslant \sum_{k=1}^n p_k W(\mu_k, \hat{\mu}_k).$$

By dual formulation of Wasserstein-1 distance,

$$W\left(\sum_{k=1}^n p_k \mu_k, \sum_{k=1}^n p_k \mu_k\right) = \sup_{f \text{ 1-Lipschitz}} \mathbb{E}_{X \sim \sum_{k=1}^n p_k \mu_k}[f(X)] - \mathbb{E}_{X \sim \sum_{k=1}^n p_k \hat{\mu}_k}[f(X)]$$

$$\leqslant \sum_{k=1}^n p_k W(\mu_k, \hat{\mu}_k).$$

$\square$

**Lemma 5.** *Given two measures* $\mu = \mu_1 \otimes \cdots \otimes \mu_n, \hat{\mu} = \hat{\mu}_1 \otimes \cdots \otimes \hat{\mu}_n$, *we have*

$$W(\mu, \hat{\mu}) \leqslant \sum_{k=1}^n W(\mu_k, \hat{\mu}_k).$$

*Proof.* For any $\epsilon > 0$, let $\pi_k$ be a coupling of $\mu_k, \hat{\mu}_k$ such that

$$\mathbb{E}_{X_k, Y_k \sim \pi_k}[\|X_k - Y_k\|_1] \leqslant W(\mu_k, \hat{\mu}_k) + \epsilon.$$

Then,

$$W(\mu, \hat{\mu}) \leqslant \mathbb{E}_{\pi_1 \otimes \cdots \otimes \pi_n}[\|X - Y\|]$$

$$= \sum_{k=1}^n \mathbb{E}_{X_k, Y_k \sim \pi_k}[\|X_k - Y_k\|_1]$$

$$\leqslant W(\mu_k, \hat{\mu}_k) + \epsilon.$$

Let $\epsilon \to 0$, we get the result. $\square$

**Lemma 6** (Approximation error of Gumbel-Softmax layer). *Given* $1 > \tau > 0$ *and* $p_1, \cdots, p_n > 0$, *let* $\mu^\tau \sim X^\tau = (X_1^\tau, \cdots, X_n^\tau)$ *be*

$$X_k = \frac{\exp((\log p_k + G_k)/\tau)}{\sum_{k=1}^n \exp((\log p_k + G_k)/\tau)}, \tag{37}$$

*where* $G_k \sim \exp(-x - \exp(-x))$ *are i.i.d. standard Gumbel variables. Let* $\mu^0$ *be the distribution of* $X^0 = \lim_{\tau \to 0} X^\tau = (X_1^0, \cdots, X_n^0)$. *Then,*

$$W(\mu^\tau, \mu^0) \leqslant 2(n-1)\tau - 2n(n-1)e^{-1}\tau \log \tau.$$

*Proof.* Without loss of generality, we may assume that $\sum_{k=1}^n p_k = 1$. Otherwise, we can divide the denominator and numerator in (37) by $\sum_{k=1}^n p_k$. Let $\Delta_n = \{(x_1, \cdots, x_n) : \sum_{k=1}^n x_k = 1\}$. We construct a transportation map $T : \Delta_n \to \Delta_n$ as follows. $T(x_1, \cdots, x_n)$ is a $n$-dimensional vector with all zeros except that the $i = \arg\max_j x_j$-th entry being one. If there are multiple coordinates

that is maximum, we choose the first coordinate and let the remaining coordinate be zero. It is easy to verify that $T_{\#}\mu^{\tau}$ and $\mu^0$ have the same distribution [36]. For any $\delta > 0$, we have

$$P(|G_{k_1} + \log p_{k_1} - (G_{k_2} + \log p_{k_2})| < \delta) = \int_{-\infty}^{\infty} \int_{y + \log \frac{p_{k2}}{p_{k_1}} - \delta}^{y + \log \frac{p_{k2}}{p_{k_1}} + \delta} \exp(-x - y - \exp(-y) - \exp(-x)) dx dy$$

$$\leqslant 2\delta e^{-1} \int_{-\infty}^{\infty} \exp(-y - \exp(-y)) dy = 2\delta,$$

where we have used $\exp(-x - \exp(-x)) \leqslant \exp(-1)$. Let event $E_{\delta} = \{\exists i, j \in [n] : |G_i + \log p_i - (G_j + \log p_j)| < \delta\}$, we have

$$P(E_{\delta}) \leqslant \sum_{i \neq j} P(|G_i + \log p_i - (G_j + \log p_j)| < \delta)$$

$$\leqslant \delta n(n - 1)e^{-1}.$$

Now, we calculate the transportation cost

$$\mathbb{E}_{X^{\tau} \sim \mu^{\tau}} \left[\|T(X^{\tau}) - X^{\tau}\|_1\right] \leqslant P(E_{\delta})\mathbb{E}_{X^{\tau} \sim \mu^{\tau}} \left[\|T(X^{\tau}) - X^{\tau}\|_1 | E_{\delta}\right] + P\left(E_{\delta}^c\right) \mathbb{E}_{X^{\tau} \sim \mu^{\tau}} \left[\|T(X^{\tau}) - X^{\tau}\|_1 | E_{\delta}^c\right].$$

Note that $\|T(X^{\tau}) - X^{\tau}\|_1 \leqslant 2$. For the first term, we have

$$P(E_{\delta})\mathbb{E}_{X^{\tau} \sim \mu^{\tau}} \left[\|T(X^{\tau}) - X^{\tau}\|_1 | E_{\delta}\right] \leqslant 2\delta n(n - 1)e^{-1}.$$

For the second term, under event $E_{\delta}^c$, let $k_{\max} = \arg\max_i X_i$, if $j = k_{\max} = \arg\max_i(\log p_k + G_k)$

$$\left|\frac{\exp((\log p_j + G_j)/\tau)}{\sum_{k=1}^n \exp((\log p_k + G_k)/\tau)} - 1\right| = \left|\frac{\sum_{k \neq j} \exp((\log p_k + G_k)/\tau)}{\sum_{k=1}^n \exp((\log p_k + G_k)/\tau)}\right|$$

$$\leqslant \left|\frac{\sum_{k \neq j} \exp((\log p_k + G_k)/\tau)}{\exp((\log p_j + G_j)/\tau)}\right|$$

$$= \sum_{k \neq j} \exp((\log p_k + G_k - (\log p_j + G_j))/\tau)$$

$$\leqslant (n - 1) \exp(-\delta/\tau).$$

If $j \neq k_{\max}$, we have

$$\left|\frac{\exp((\log p_j + G_j)/\tau)}{\sum_{k=1}^n \exp((\log p_k + G_k)/\tau)}\right| \leqslant \exp((\log p_j + G_j - (\log p_{\max} + G_{\max}))/\tau) \leqslant \exp(-\delta/\tau).$$

Therefore,

$$\mathbb{E}_{X^{\tau} \sim \mu^{\tau}} \left[\|T(X^{\tau}) - X^{\tau}\|_1 | E_{\delta}^c\right] \leqslant 2(n - 1) \exp(-\delta/\tau).$$

We get an upper bound of the transportation cost,

$$W\left(\mu^{\tau}, \mu^0\right) \leqslant \mathbb{E}_{X^{\tau} \sim \mu^{\tau}} \left[\|T(X^{\tau}) - X^{\tau}\|_1\right] \leqslant 2\delta n(n - 1)e^{-1} + 2(n - 1) \exp(-\delta/\tau).$$

Let $\delta = -\tau \log \tau$, we get

$$W\left(\mu^{\tau}, \mu^0\right) \leqslant 2(n - 1)\tau - 2n(n - 1)e^{-1}\tau \log \tau.$$

$\square$

**Lemma 7** (Approximation Error of Sinkhorn distance). *For any $\mu \in \Delta_n, \nu \in \Delta_m$ and $\lambda > 0$, we have*

$$0 \leqslant W_{1,\lambda}(\mu, \nu) - W(\mu, \nu) \leqslant (\log(mn) + 1)\lambda,$$

*where $W_{1,\lambda}(\cdot, \cdot)$ is the entropy regularized Wasserstein-1 distance. Moreover, we have the following estimation of approximation error.*

$$|S_{\lambda}(\mu, \nu) - W(\mu, \nu)| \leqslant 2(\log(mn) + 1)\lambda.$$

*Proof.* Let $h(T) = -\sum_{i,j=1}^{n,m} T_{ij} \log T_{ij} + 1$, by [35, Proposition 1], we have

$$0 \leqslant W_{1,\lambda}(\mu,\nu) - W(\mu,\nu) \leqslant c\lambda, \tag{38}$$

where $c = \max\{h(T) : \text{transportation plan } T \text{ is achieve optimal loss } W(\mu,\nu)\}$. Since $T_{ij} \in [0,1]$ and $f(x) = -x \log x$ is concave, we have

$$h(T) = -nm \cdot \frac{1}{nm} \sum_{i,j=1}^{n,m} T_{ij} \log T_{ij} + 1.$$

$$\leqslant -nm \cdot \left( \frac{1}{nm} \sum_{i,j=1}^{n,m} T_{ij} \right) \log \left( \frac{1}{nm} \sum_{i,j=1}^{n,m} T_{ij} \right) + 1$$

$$= 1 + \log(nm).$$

By definition of Sinkhorn distance,

$$S_\lambda(\mu,\nu) = W_{1,\lambda}(\mu,\nu) - W_{1,\lambda}(\mu,\mu)/2 - W_{1,\lambda}(\nu,\nu)/2.$$

By (38), we get

$$|S_\lambda(\mu,\nu) - W(\mu,\nu)| \leqslant 2(\log(mn) + 1)\lambda.$$

$\square$

**Lemma 8.** *Given a measure $\mu$ on $\mathbb{R}^d$ and a real function class $\mathcal{F}$ with output dimension $d$, suppose that the pseudo-dimension of $\mathcal{F}$ is less than $d_\mathcal{F} < \infty$ and there exists $K > 0$ such that $|f(x)| \leqslant K, \forall f \in \mathcal{F}, x \in \mathbb{R}^{d'}$, then with probability at least $1 - \exp\left(O\left(\delta^{-d}\log\left(\delta^{-1}\right) + d_\mathcal{F}\log \epsilon^{-1} - n\epsilon^2\right)\right)$,*

$$\sup_{f \in \mathcal{F}} W(f_\#\mu, f_\#\mu_n) \leqslant \delta + \epsilon$$

*for all $\delta, \epsilon > 0$, where $\mu_n$ is the empirical distribution of $\mu$.*

*Proof.* By the dual formulation of the Wasserstein distance,

$$\sup_{f \in \mathcal{F}} W(f_\#\mu, f_\#\mu_n) = \sup_{h \in \mathcal{F}_1(\mathbb{R}^d, \mathbb{R}^d), h(0)=0} \sup_{f \in \mathcal{F}} \mathbb{E}_{X \sim \mu_n}[h \circ f(X)] - \mathbb{E}_{X \sim \mu}[h \circ f(X)].$$

We define

$$\mathcal{N}_1(\epsilon, \mathcal{F}, n) = \sup_{x_1, \cdots, x_n} \mathcal{N}(\epsilon, \{(f(x_1), \cdots, f(x_n) : f \in \mathcal{F})\}, \|\cdot\|_1),$$

where $\mathcal{N}(\epsilon, S, \|\cdot\|_1)$ is the covering number of set $S$ in $\ell_1$ norm. Obviously, if $h$ is 1-Lipschitz,

$$\mathcal{N}_1(\epsilon, h \circ \mathcal{F}, n) \leqslant \mathcal{N}_1(\epsilon, \mathcal{F}, n).$$

By Theorem 18.4 in [2], for any fixed $h$,

$$\mathcal{N}_1(\epsilon, h \circ \mathcal{F}, n) \leqslant \mathcal{N}_1(\epsilon, \mathcal{F}, n) \leqslant e(d_\mathcal{F} + 1)\left(\frac{2e}{\epsilon}\right)^{d_\mathcal{F}}.$$

By Theorem 17.1 in [2],

$$P\left(\sup_{f \in \mathcal{F}} \mathbb{E}_{X \sim \mu_n}[h \circ f(X)] - \mathbb{E}_{X \sim \mu}[h \circ f(X)] > \epsilon\right) \leqslant \exp\left(O\left(d_\mathcal{F}\log \epsilon^{-1} - n\epsilon^2\right)\right).$$

Let $\mathcal{H}_\delta$ be the $\delta$-net of the set $\mathcal{H} = \left\{h : h \in \mathcal{F}_1\left([-K,K]^d, [-K,K]^d\right), h(0) = 0\right\}$ in $\ell_\infty$ norm. By Lemma 6 in [21],

$$|\mathcal{H}_\delta| \leqslant \exp\left(O\left(\delta^{-d}\log \delta^{-1}\right)\right).$$

Therefore, with probability no more than $\exp\left(O\left(\delta^{-d}\log \delta^{-1} + d_\mathcal{F}\log \epsilon^{-1} - n\epsilon^2\right)\right)$

$$\sup_{h \in \mathcal{H}_\delta, f \in \mathcal{F}} \mathbb{E}_{X \sim \mu_n}[h \circ f(X)] - \mathbb{E}_{X \sim \mu}[h \circ f(X)] > \epsilon.$$

Notice that

$$\sup_{h \in \mathcal{H}, f \in \mathcal{F}} \mathbb{E}_{X \sim \mu_n}[h \circ f(X)] - \mathbb{E}_{X \sim \mu}[h \circ f(X)] \leqslant 2\delta + \sup_{h \in \mathcal{H}_\delta, f \in \mathcal{F}} \mathbb{E}_{X \sim \mu_n}[h \circ f(X)] - \mathbb{E}_{X \sim \mu}[h \circ f(X)],$$

which implies that with probability no more than $\exp\left(O\left(\delta^{-d}\log \delta^{-1} + d_\mathcal{F}\log \epsilon^{-1} - n\epsilon^2\right)\right)$

$$\sup_{h \in \mathcal{H}, f \in \mathcal{F}} \mathbb{E}_{X \sim \mu_n}[h \circ f(X)] - \mathbb{E}_{X \sim \mu}[h \circ f(X)] > 2\delta + \epsilon.$$

$\square$

