# OpenReview forum: "Consistency of Neural Causal Partial Identification"
_NeurIPS.cc/2024/Conference — NeurIPS 2024 poster_

### Official Review · Reviewer_Mk7v · 2024-07-12

**Soundness:** 3
**Presentation:** 2
**Contribution:** 3
**Rating:** 5
**Confidence:** 2

**Summary:**

This paper studies theoretically the partial identification capabilities of neural causal models (NCMs), that is, the extent to which this type of models can approximate the interval on which a certain causal query falls on, $\theta(\mathcal{M}) \in [\underline{F}, \overline{F}]$. To this end, the authors develop a lot of results showing that under ideal scenarios with oracle knowledge on the exogenous distributions, NCMs can approximate any SCM if the proper architecture and regularization is used. Interestingly, the authors show that regularization is key in these settings, as otherwise the observational likelihood can be modelled arbitrarily well, while the interventional distribution do not look alike. It is worth-noting that the theory developed consider general scenarios with confounders and mixed-type data.

**Strengths:**

- **S1.** This work addresses an intellectually interesting question of whether NCMs can properly perform causal partial identification.
- **S2.** The theory developed looks general, sound, and it makes sense on an intuitive level (disclaimer: I have _not_ looked into the proofs).
- **S3.** The work properly studies mixed-type data, which I have rarely seen done.
- **S4.** The fact that NCMs _need_ a form of regularization to properly approximate the SCMs is really interesting.

**Weaknesses:**

- **W1.** My personal main concern is that I do not immediately see any practical implication of the theory developed here. What I mean is that, while I find it interesting, if I have understood it correctly it works under the assumption that we know the exogenous distribution of the ground-truth model, which we never know, and which we could be arbitrarily different from whatever we model (we can always change the exogenous distribution and push the transformation to the generator $f$).
- **W2.** While trying to be general, I similarly feel that some of the assumptions can be quite restrictive in practice. E.g., global Lipschitz-continuity is quite a constraining assumption from my point of view.
- **W3.** I know it is a dense theory paper, but the presentation leaves a lot to be desired. There is a lot of similar notation, non-standard definitions (e.g. SCMs), and a lack of intuition/examples that makes the results hard to parse. The paper is simply not accessible to a large percentage of the community.
- **W4.** The experimental section is extremely short, and it would require of more settings, an ablation study of the different components of the theory, and a proper analysis of the results.

**Questions:**

I have no specific questions, but I want to make clear that a dense paper like this would require many more hours for me to fully understand it and I may have missed some important points. I will reflect this in my confidence score.

**Limitations:**

Limitations are **not** discussed. In the checklist, it is said to be discussed on the Conclusion section, which does not exist.

---

> ### Author Rebuttal · Authors · 2024-08-05
>
> Thank you for your feedback! Below is our response. We hope our clarification can solve your problem.
>
> ***1. The method only works under the assumption that we know the exogenous distribution of the ground-truth model.***
>
> We thank the reviewer for their advice. However, we would like to kindly elaborate on a misunderstanding about a key criticism and we will make sure to make this point even more clearly described in the paper: Our algorithm does not need to know the ground-truth exogenous distribution of the latent factors and this is a key development of our work. We only need to assume that these unknown exogenous distributions satisfy some mild regularity assumptions (Assumption 4,5). This is a key point of partial identification. Without knowing the exact latent distribution, we search over all NCMs that induce a distribution similar to the one observed empirically (measured by a metric distance) and take the maximum and minimum of the causal quantity we are interested in subject to this metric distance constraint.
>
> In our algorithm, we push forward uniform and Gumbel variables using neural networks to simulate exogenous distributions that satisfy Assumption 4 (and may Assumption 5, depending on which architecture to use) and see the maximum and minimum of the causal quantity of causal models whose latent distributions fall in this class (Problem (7) in the paper). A key theoretical insight of our work is that by passing these simple distributions through appropriate neural network architectures, one can represent arbitrary latent variable distributions and hence “fit” any ground truth latent distribution that satisfies the regularity assumptions. Thus arbitrary SCMs, with general unknown latent factor distributions can be represented through NCMs and when fitting to data, one does not need to know the actual distribution of the latent factors. For this reason, our theoretical results have a strong implication for practice and essentially analyze the methodology that prior work had extensively studied empirically.
>
> ***2. While trying to be general, some of the assumptions can be quite restrictive in practice. E.g., global Lipschitz-continuity is quite a constraining assumption.***
>
> Thanks for your feedback. We work with compactly supported distributions (Assumption 3), so a local Lipschitz function is automatically global. While certainly not fully general, the Lipshitz continuity assumption is a standard assumption in the literature on the representation theory of neural network architectures and most neural network architectures (e.g. ReLU) are in fact globally Lipschitz. Moreover, even if there is no confounding in the SCM, to learn the SCM, we need to learn the structural functions $f_i$. To avoid impossibility results, like the No-free-lunch theorem, one needs to make some structural assumptions on the function spaces that the functions $f_i$ lie in. Lipschitz function spaces are a natural choice when providing theoretical statistical consistency and representativeness theorems [3,4,5].
>
> ***3. There is a lot of similar notation, non-standard definitions (e.g. SCMs), and a lack of intuition/examples that makes the results hard to parse.***
>
> Thank you for your feedback. Due to space limitations, we are unable to present all the details in the main body. However, we provide some examples in the appendix.  For the definition of SCM, we follow the definition of Xia et al.[1]. The only difference with the traditional SCM literature is that we allow the latent “noise” variables to enter more than 2 observed nodes. This convention was also followed in Xia et al.[1]. The causal graph induced by an SCM in our definition is a kind of Acyclic Directed Mixed Graph obtained by latent projection [2]. We have provided one example in Appendix A to illustrate our definition.
>
>
> ***4. The experimental section is extremely short, and it would require more settings and a proper analysis of the results.***
>
> Thank you for your comment. As we point out in the paper, the methodology that we analyze has been proposed and analyzed experimentally in prior work. The main contribution of our paper is to provide a theoretical justification of this methodology in general environments. Please see item 1 of our response to the reviewer zYjd for details.
>
> ***5. Limitations are not discussed and the conclusion is missing.***
>
> Thank you for pointing out! We did not include the conclusion part due to space limitations, which can be easily addressed. We will add a ‘Conclusion and Limitations’ section in the camera-ready version, which is shown in the response of reviewer 8521 item 4.
>
> [1] Kevin Xia, Kai-Zhan Lee, Yoshua Bengio, and Elias Bareinboim. The causal-neural connection: Expressiveness, learnability, and inference, October 2022.
>
> [2] Jin Tian and Judea Pearl. On the testable implications of causal models with hidden variables, December 2002.
>
> [3] Yarotsky, Dmitry. "Optimal approximation of continuous functions by very deep ReLU networks." Conference on learning theory. PMLR, 2018.
>
> [4] Yarotsky, Dmitry. "Error bounds for approximations with deep ReLU networks." Neural networks 94 (2017): 103-114.
>
> [5]Wainwright, Martin J. High-dimensional statistics: A non-asymptotic viewpoint. Vol. 48. Cambridge university press, 2019.

---

> > ### Comment · Reviewer_Mk7v · 2024-08-11
> >
> > I thank the authors for their kind and detailed response, it should be clear by now that I am an outsider on this particular problem, and the initial barrier is a bit too tough to pass in the reviewing period. Clearly, other reviewers were much more positive than I initially was.
> >
> > With that said, I understand the practical implications of this work, and I applaud it. However, I still have some questions for which I would appreciate an answer (I apologize I don't have currently the time to go through the proofs in the appendix, yet I'll try to go through the main paper one more time).
> >
> > To my understanding, one key aspect of this work is splitting the estimation error in two terms, the error in the exogenous distribution and that of the causal mechanisms. From there, one can use the theory of NN approximation to estimate the error committed in each part, depending on depth and width. To achieve this result, it is necessary to assume a specific $\mathcal{M}^*$ in its canonical form, which would have a specific Lipschitz constant $L^*$ that the proofs should somehow use.
> >
> > What I don't understand is that, without assuming that one knows the exogenous distribution of $\mathcal{M}^*$, it is possible to find alternative canonical representations by applying arbitrary transformations $\phi$ to the canonical form, i.e., having $(\phi \circ P_U, \phi^{-1} \circ f)$ instead of $(P_U, f)$, with arbitrary Lipschitz constants for the target SCM. As far as I can see, the definition of the canonical form does allow these cases. How does your work rule them out? I guess some of the assumptions do it?
> >
> > Thanks in advance.

---

> > > ### Author Response · Authors · 2024-08-13
> > >
> > > Thank you for your question.
> > >
> > > We want to clarify that the approximation theorems (Theorem 1 and Corollary 1) are tools we use to prove the consistency theorem (Theorem 4), which is the central result of our paper. These approximation theorems state that given an SCM (with known latent distribution and structure functions) that satisfies some assumptions, we can approximate it using NCMs. However, in the consistency theorem, *we do not assume that we know the latent distribution or the structure functions (nor does the consistency theorem claim that we recovered the true latent distribution; just that the true latent distribution is part of the “search space”)*. The consistency theorem proved the consistency of the bounds for the target estimand.
> > >
> > > As we mentioned before, the goal of partial identification is to determine bounds for a causal quantity across all SCMs that have the same observation distribution as the ground-truth model. Since it is difficult to search over all SCMs,  we argue that it is sufficient—and more practical—to search over all NCMs. The approximation theorems are used to prove that the set of NCMs is ‘close’ to the set of SCMs so that the errors caused by this substitution in the partial identification problem can be controlled.
> > >
> > > Therefore, the short answer to your question is that we cannot rule out alternative causal models that result from transformations, as long as they satisfy all the assumptions (particularly Assumptions 3 and 4). Since we only observe a subset of the variables, multiple causal models could potentially lead to the same observational distribution. Without additional information about the true underlying model, it is often difficult to uniquely identify a single causal model from observations alone.
> > >
> > > This is a typical case in causal inference. Even when the target estimand is point identified (e.g. ATE with no unobserved confounding), and hence our bounds converge to a singleton, that does not mean that the latent distribution of the structural causal model nor the structural functions are identifiable, due to the ambiguity you mention. Still, in such unique identification cases, all such transformations yield the same target estimand.
> > >
> > > We hope this clarification helps you better understand our work, and we are happy to address any further questions you might have.

---

> > > > ### Comment · Reviewer_Mk7v · 2024-08-14
> > > >
> > > > Many thanks once again for the detailed response. I think I now get a bit better the intuition of the work.
> > > >
> > > > Since the discussion period is over, I am going to increase my score to borderline accept, as I do not feel comfortable giving more score to a work for which I don't have a full grasp on the results. This is mostly on me, coming from a different area, and the "mathematical heaviness" of the work, which would require many more hours of me to fully grasp.
> > > >
> > > > However, this should be not a problem, as my confidence score is already quite low.

---

### Official Review · Reviewer_VZrh · 2024-07-12

**Soundness:** 3
**Presentation:** 3
**Contribution:** 2
**Rating:** 6
**Confidence:** 3

**Summary:**

This paper provides a novel perspective and solid contribution to the case where continuous and categorical variables both exist.

**Strengths:**

The presentation and organization are concise and clear; the contribution is solid.

**Weaknesses:**

The assumption is somehow strong (e.g., assumption 2).

**Questions:**

How do we argue the importance of solving the problem of ``the approximation error and consistency of the optimization-based approach to partial identification via NCMs''?

**Limitations:**

See above.

---

> ### Author Rebuttal · Authors · 2024-08-05
>
> Thank you for your comments and questions! Below is our response. We hope our clarification can solve your problem.
>
> ***1. The assumption is somehow strong (e.g., assumption 2).***
>
> Thank you for your feedback. Assumption 2 is a relatively new assumption about the data generation process of categorical variables, but which again we view as a mild assumption that is basically an analog of the Lipschitz structural equation function assumption that we place on continuous variables. This assumption basically states that if one conditions on all the latent factors that are impacting a given variable and which also impact other variables (i.e. once one conditions on the latent confounders for this variable), then the probability that this categorical variable takes any given value is a Lipschitz function of this confounding latent factors. We make this intuition formal using the following formal convention. The classical definition of SCM assumes that the structural equations have the form $ V_i = f_i(\text{Pa}(V_i),U_i) $, where $\text{Pa}(V_i)$ is the set of parents of node $V_i$. However, if we follow this definition and $V_i$ is categorical, this means $f_i$ can only take finitely many values. If one of the inputs is a continuous variable, the only way for $f_i$ to be continuous (which is a quite mild assumption) is that $f_i$ is a constant function, which is trivial. To avoid this situation, we assume that the latent variables have two parts, the confounding part $U$ and the independent noises part $G$. The categorical variable $V_i$ is generated by the following process. First, we calculate the propensity function $f(\text{Pa}(V_i),U_i) \in \Delta =\\{x\in\mathbb{R}^d:x_i \geq 0, \sum x_i = 1\\}$. Then, we generate a categorical variable according to this probability using the independent noise $G_i \in G$. In this way, we only need to approximate the propensity in the learning process.
>
> We also want to emphasize that the Gumbel variables in assumption 2 are chosen for convenience to generate categorical distributions in the second step mentioned above. One can replace the Gumbel variables with any random variables that can generate categorical distributions. Therefore, this assumption is basically a natural analog of the “Lipschitz structural equation” assumption, to the data generation process of categorical variables.
>
> ***2. How do we argue the importance of solving the problem of “the approximation error and consistency of the optimization-based approach to partial identification via NCMs”?***
>
>
> Thank you for your question. As we mentioned in our paper, identifying causal quantities from observational data is an important problem. However, in the presence of unobserved confounding, it is usually impossible to accurately identify many causal quantities. In that situation, partial identification gives a remedy to the problem.
>
> There has been rich literature on partial identification in causal inference and the optimization-based approach has demonstrated good empirical performance [1,2,3,5].  However, in the general continuous variables setting (and the mixed setting), previous methods [1,2,3] do not have any theoretical result on the soundness of their methods. Moreover, the NCM approach is the only generic methodology for partial identification with general variables. Prior general methodologies for automated Partial Identification [5], that don’t go through the NCM method, are only available for discrete variables and, as our experiments showcase, they translate poorly to the continuous variable setting (if one invokes discretization) and don’t have natural continuous counterparts that one could analyze theoretically. Thus our paper fills this gap: provides a theoretical justification of the only existing candidate for an automated partial identification procedure for general variables.
>
> [1]Kocaoglu, Murat, et al. "Causalgan: Learning causal implicit generative models with adversarial training." arXiv preprint arXiv:1709.02023 (2017).
>
> [2]Vahid Balazadeh, Vasilis Syrgkanis, and Rahul G. Krishnan. Partial identification of treatment effects with implicit generative models, October 2022
>
> [3]Kirtan Padh, Jakob Zeitler, David Watson, Matt Kusner, Ricardo Silva, and Niki Kilbertus. Stochastic causal programming for bounding treatment effects.In Conference on Causal Learning and Reasoning pages 142–176. PMLR, 2023
>
> [4] Florian Gunsilius. A path-sampling method to partially identify causal effects in instrumental variable models, June 2020
>
> [5] Guilherme Duarte, Noam Finkelstein, Dean Knox, Jonathan Mummolo, and Ilya Shpitser. An automated approach to causal inference in discrete settings, September 2021

---

> ### Comment · Reviewer_VZrh · 2024-08-11
> **Thanks**
>
> It is a concise and high-quality response. Thank you! I raise my score to 6.
>
> Besides, in fact, I'm quite familiar with Partial Identification, one of my main research topics. I also share two nice new papers and recommend you cite them as follows.
>
> Of course, I acknowledge that this literature is somehow different from your settings, but they all make meaningful contributions to general PI. For instance, after my careful proofreading, Literature [1] 's main IFF theorem holds in discrete/continuous/mixed cases.
>
> Finally, my last question is: Why is the extension from the discrete case to the continuous case highly non-trivial? For instance, when we get the continuous variable, why not conduct naive discretization, and the corresponding result may not change significantly? If this holds true, it will not be interesting. So, I wonder about some more insightful reasons/counterexamples.
>
> Thank you! I will continuously consider raising my score.
>
>
>
> [1] tight partial identification of causal effects with marginal distribution of unmeasured confounders. (Zhang, ICML2024 spotlight)
>
> [2] Model-Agnostic Covariate-Assisted INference on Partially Identified Causal effects (Wen et al)

---

> > ### Author Response · Authors · 2024-08-11
> >
> > Thank you so much for your feedback and for sharing the recent literature with us. We sincerely appreciate your contributions to improving our work.
> >
> > Regarding the two papers you mentioned, they indeed focus on a specific type of causal graph, which differs from the general causal graph setting our algorithm addresses. However, we agree that these papers offer valuable insights for the partial identification literature. We will cite them in the camera-ready version of our paper.
> >
> > As for your question about discretization, we appreciate the opportunity to clarify this point. As we mention in the paper, for a general discrete SCM, the Partial Identification (PI) problem becomes a Polynomial Programming (PP) problem [1], which is NP-hard in general. While discretizing continuous data and solving the PP problem to obtain bounds is possible, the problem's size grows *exponentially* with the cardinality of the support. This implies that the finer the discretization of the continuous data, the larger and more computationally expensive the resulting PP problem becomes.
> >
> > In our experiments, we found that the PP problem quickly becomes intractable when we increase the support's cardinality of continuous variables. For instance, in a continuous IV setting, we compared our algorithm with Autobounds [1], the state-of-the-art algorithm for solving discrete PI problems, by discretizing all continuous variables such that their support's cardinality is 8 (details are in Section D.2). This produces a PP problem of size approximately $2^{14}$. However, even we solved such a large problem, the bound obtained by this approach was not as tight as the one obtained using the NCM approach. While finer discretization could potentially improve the bound, the resulting PP problem becomes so large that we were unable to obtain a solution after running the Autobounds algorithm for over a day. Therefore, we believe it is crucial to generalize the NCM approach to include continuous (and mixed) variables, as it proves to be more efficient and powerful compared to the discretization approach in these settings.
> >
> > Thank you again for your detailed review and valuable questions. We are glad to take any further questions.
> >
> >
> > [1] Guilherme Duarte, Noam Finkelstein, Dean Knox, Jonathan Mummolo, and Ilya Shpitser. An automated approach to causal inference in discrete settings, September 2021

---

### Official Review · Reviewer_8521 · 2024-07-12

**Soundness:** 4
**Presentation:** 3
**Contribution:** 4
**Rating:** 7
**Confidence:** 4

**Summary:**

The paper extends neural causal models (NCMs) to the continuous and mixed-type variables. NCMs is a neural-networks based tool to perform an automated point/partial identification and estimation of causal queries. The authors provided new theoretic results (1) on how to construct a canonical representation of an SCM, (2) on how to approximate it with sufficiently expressive Lipshitz neural networks, and (3) on additional assumptions to perform the consistent identification/estimation of an average treatment effect (ATE) from an arbitrary causal digram and an observational distribution.  The paper provides experimental results for the partial identification of the ATE in the discrete and continuous outcomes settings.

**Strengths:**

To the best of my knowledge, this paper is the first one to advance neural causal models (NCMs) to the mixed-type variables setting. NCMs are a universal tool for general automated partial identification and estimation of causal quantities (a very important problem in causal inference and treatment effect estimation). The theory provided in the paper is very general and extends over simple linear SCMs. The paper is clearly written (as far as a massive theoretic contribution allows) and well-structured. The authors rigorous proofs for all the theoretic statements.

**Weaknesses:**

I didn’t find any major or minor weaknesses in the paper. Yet, I have several suggestions on how to improve the paper’s understandability for the general public of the NeurIPS conference:
1. I encourage the authors to provide more examples of how to construct canonical representations for some simple SCMs and then the corresponding neural architectures of the NCMs.
2. I would provide a more precise explanation of Figures 1 and 5, e.g., the number of layers in the yellow and blue blocks. Also, multiple notation elements could potentially be added to the Figures, e.g., the number of connected components in the latent space or the number of confounded components.
3. I also encourage the authors to provide the code of the proposed method.
4. Seems like the conclusion is missing in the main part of the text.

I am willing to increase my score if the authors implement the above-mentioned suggestions.

**Questions:**

I appreciate the provided continuous IV experiment. I wonder, what is the performance of the method in other confounded ATE settings where the ground-truth is available, e.g., a no-assumptions bound for the ATE with hidden confounding?

**Limitations:**

The authors have fully stated the limitations of their theory.

---

> ### Author Rebuttal · Authors · 2024-08-05
>
> We sincerely thank the reviewer for the helpful suggestions, which make our paper clearer and more readable. Below are the changes we make.
>
> ***1. Provide more examples of how to construct canonical representations for some simple SCMs and then the corresponding neural architectures of the NCMs.***
>
> Thanks for the suggestion. We will provide more examples in the appendix in the later version. Specifically, we plan to use the example in Appendix A as an example.
>
> The causal graph of this example is shown in Figure 2b in our paper. Following the construction in Proposition 4, we use one latent variable for each $C^2$ component. As we explain in Appendix A, this causal model has three $C^2$ components, $ \\{V_1,V_2,V_3\\}, \\{V_3,V_4\\}, \\{V_4,V_5\\} $. In the canonical representation, exactly the latent variables enter their corresponding $C^2$ component. The canonical model is shown in Figure 2(a) in the global response.
>
> Now, we show how to construct the NCM architecture from a canonical representation. As we mentioned in Section 3, we approximate the latent distribution by pushing forward uniform and Gumbel variables. The structure equations of the NCM are
> \begin{align}
>     &V_1 = f^{\theta_1}_1(V_2,g_1^{\theta_1}(Z_1)),\\\\
>     &V_2 = f^{\theta_2}_2(g_1^{\theta_1}(Z_1)),\\\\
>     &V_3 = f^{\theta_2}_3(V_1,V_2,g_1^{\theta_1}(Z_1),g_2^{\theta_1}(Z_2)), \\\\
>     &V_4 = f_4^{\theta_4}(V_3.g_2^{\theta_1}(Z_2)),\\\\
>     &V_5 = f_5^{\theta_5}(V_4,g_2^{\theta_3}(Z_3)),
> \end{align}
> where $f_i^{\theta_i}, g_j^{\theta_j}$ are neural networks, $Z_i$ are join distribution of independent uniform and Gumbel variables, and $g_j^{\theta_j}$ has the special architecture described in Section 3.1 and Section 3.2.  Figure 2(b)  in the global response shows the architecture of the NCM. Each in-edge represents an input of a neural net.
>
>
> ***2. I would provide a more precise explanation of Figures 1 and 5, e.g., the number of layers in the yellow and blue blocks. Also, multiple notation elements could potentially be added to the Figures, e.g., the number of connected components in the latent space or the number of confounded components.***
>
> Thank you for your helpful advice. We have modified figure 1 and 5 according to your advice. We include the figures after modification in the global response PDF. Figure 5 is changed similarly and captions about the number of layers will be added too. We will implement these changes in the later version.
>
>
> ***3. I also encourage the authors to provide the code of the proposed method.***
>
> Thanks for the suggestion! We will include the link to the code in the camera-ready version if our paper gets accepted.
>
>
> ***4. The conclusion is missing in the main part of the text.***
>
>  Thank you for pointing out. We did not include it to avoid some repetition given space constraints, but we will add it for the camera-ready version. We will add the following ‘Conclusion and Limitations’ section to the paper.
>
>  In this paper, we provide theoretical justification for using NCMs for partial identification. We show that NCMs can be used to represent SCMs with complex unknown latent distributions under mild assumptions and prove the asymptotic consistency of the max/min estimator for partial identification of causal effects in general settings with both discrete and continuous variables. Our results also provide guidelines on the practical implementation of this method and on what hyperparameters are important, as well as recommendations on values that these hyperparameters should take for the consistency of the method. These practical guidelines were validated with a small set of targeted experiments, which also showcase superior performance of the neural-causal approach as compared to a prior main contender approach from econometrics and statistics, that involves discretization and polynomial programming.
>
> An obvious next step in the theoretical foundation of neural-causal models is providing finite sample guarantees for this method, which requires substantial further theoretical developments in the understanding of the geometry of the optimization program that defines the bounds on the causal effect of interest. We take a first step in that direction for the special case, when there are no unobserved confounders and view the general case as an exciting avenue for future work.
>
>
>
> ***5. What is the performance of the method in other confounded ATE settings where the ground-truth is available, e.g., a no-assumptions bound for the ATE with hidden confounding?***
>
> Thank you for asking. We do an extra experiment on the leaky mediation setting [1]. The structure equations of this causal model are
> 	\begin{align*}
> 	T &= C + U_T, \\\\
> 	X &= T + U_X + U, \\\\
> 	Y &= 2X + U +C +U_Y, \\\\
> 	\end{align*}
> where $C,U,U_T,U_X,U_Y \sim \text{Unif}(-1,1)$ are latent variables and $T,X,Y$ are observed variables. Like the settings in our paper, we compare our algorithm with the Autobounds package. The true ATE of this model is 2. The following results are the averages of 10 experiments for each algorithm. The bounds obtained by NCM is [-4.19,4.12], while the bound obtained by Autobounds is [-10.97, 12.56].
>
> The bounds obtained by both algorithms seem non-informative because for this partial identification problem, the ATE can be an arbitrary real number. This is because the following causal model, with $\alpha$ taking over all real numbers, has the same observation distribution as the ground-truth model.
> \begin{align*}
> 	T &= C + U_T, \\\\
> 	X &= T + U_X + U, \\\\
> 	Y &= \alpha X + (3 - \alpha)(U +C) + U_Y + (2 - \alpha)U_X, \\\\
> 	\end{align*}
> 	The results of the NCM approach correctly reflect this fact.
>
> [1] Padh, Kirtan, et al. "Stochastic causal programming for bounding treatment effects." Conference on Causal Learning and Reasoning. PMLR, 2023.

---

> > ### Comment · Reviewer_8521 · 2024-08-12
> >
> > Thank you for the clarifications.
> >
> > I still have an important concern regarding the no-assumptions bound experiments (5).
> >
> > For the outcome $Y \in [a, b]$, the no-assumption (Manski) bounds on the ATE always have the width of $b - a$ [1]. In the example, provided by the authors, $Y \in [-11, 11]$ and, thus, the ground-truth width is $b - a = 22$. Therefore, it seems that the method proposed by the authors yields invalid bounds with a width of $8.32$ (whereas the Autobounds package provides a more valid width of $23.53$). Could the authors explain the source of the invalidity?
> >
> > **References**:
> > - [1] Manski, Charles F. "Nonparametric bounds on treatment effects." The American Economic Review 80.2 (1990): 319-323.

---

> > > ### Author Response · Authors · 2024-08-13
> > >
> > > Thank you for your question.
> > >
> > > We would like to point out that the Manski bound [1] is derived specifically for binary treatment settings, whereas in our example, the treatment is continuous. Besides, the causal graph of [1] is different from the leaky mediation graph. For these reasons, it is not immediately clear whether the same bound can be applied directly in our context. A more comparable setting is the binary IV example in our paper, where both algorithms yield similar bounds. In the binary IV example, we use the algorithm from [2] to verify that the bounds we get are correct.
> > >
> > > Additionally, we impose Lipschitz constraints on the structure functions when solving the PI problem, which may result in tighter bounds. However, the analytic characterization of the optimal bounds under Lipschitz constraints has not been established and seems hard to establish in analytic form.
> > >
> > >
> > > We are happy to address any further questions you may have.
> > >
> > >
> > > [1] Manski, Charles F. "Nonparametric bounds on treatment effects." The American Economic Review 80.2 (1990): 319-323.
> > >
> > > [2] Balke, Alexander, and Judea Pearl. "Bounds on treatment effects from studies with imperfect compliance." Journal of the American statistical Association 92.439 (1997): 1171-1176.

---

> > > > ### Comment · Reviewer_8521 · 2024-08-13
> > > >
> > > > Thank you, now it is clear to me. I maintain my original positive score.

---

### Official Review · Reviewer_zYjd · 2024-07-13

**Soundness:** 3
**Presentation:** 2
**Contribution:** 3
**Rating:** 6
**Confidence:** 2

**Summary:**

This paper develop consistency results for partial identification via neural causal model with both continuous and categorical variables. Their results shed light on the impact of the neural network architecture and Lipschitz regularization during training. The resulting method can be trained via gradient-based optimization algorithms, and is validated on synthetic data.

**Strengths:**

- The paper is clearly written. The assumptions used are clearly stated.
- The identification result is useful and a good complement to existing ones, by making them more general.
- The identification result and algorithm are technically sound.

**Weaknesses:**

- The empirical validation of the algorithm/result is not extensive, but I understand that the goal of the work is to establish identification results and a thorough empirical study might not be necessary.
- The paper would benefit from giving some proof sketches (or a brief overview of the proof strategy) in the main paper.
- Many of the assumptions and results are not well explained, making it not straightforward to understand the assumptions and implications of the results.

**Questions:**

- A conclusion section does not seem to be provided.
- It would enrich the paper to briefly discuss the connection with optimization-based approaches in causal learning (many of which also used similar techniques including augmented Lagrangian and gumbel techniques), although the task is not completely the same (see e.g. https://arxiv.org/abs/2007.01754, https://arxiv.org/abs/1910.08527, https://openreview.net/forum?id=HsSLdHuAmnY).
- It would be helpful to elaborate more on the assumptions required and discuss some examples where the assumptions are satisfied/violated.
- Other related work that could be worth mentioning: https://arxiv.org/abs/2105.12891

**Limitations:**

The limitations have not been well discussed.

---

> ### Author Rebuttal · Authors · 2024-08-05
>
> Thank you for your advice to improve our paper! Below is our response. We hope our clarification can solve your problem.
>
> ***1. The empirical validation of the algorithm/result is not extensive.***
>
> Thank you for your feedback. As we point out in the paper, the methodology that we analyze has been proposed and analyzed experimentally in prior work [2,3,4]. The main contribution of our paper is to provide a theoretical justification of this methodology in general environments. Given the prior experimental validation we did not perform extensive experiments and we opted to perform a small set of targeted experiments that highlight the key design elements of the procedure (e.g. neural architecture, Lipschitz regularization) that our theory predicts are important. Moreover, we note that unlike prior work, our work is the first to compare experimentally the neural-causal approach to the polynomial programming approach of [5], which is a key contender in the prior literature in partial identification in econometrics and statistics (see Appendix D for more details on our experimental setup).
>
>
> ***2. The paper would benefit from giving some proof sketches (or a brief overview of the proof strategy) in the main paper.***
>
> Thanks for your suggestion. We will add some proof sketches in the main body and before the proof in the appendix. Here, we briefly discuss our proving strategy of the main consistency theorem (Theorem 4).
>
> We first establish a general result concerning the convergence of the optimal values of a sequence of optimization problems (Proposition 6). More specifically, we consider a sequence of constrained optimization problems $\\{P_n\\}$. We prove that if the objective and constrained functions satisfy some regularity assumptions (i.e. Lipschitz), the domain is compact, and the sequence of constrained functions converges uniformly to a function, the upper limit and lower limit of the sequence of optimal values can be bounded. Then, apply Proposition 6 to this setting, verifying the partial identification problem satisfies the assumptions of the proposition using previous approximation results (Theorem 1,2,3 and Corollary 1).
>
> ***3. A conclusion section does not seem to be provided.***
>
> Thanks for the feedback. We did not include it to avoid some repetition given space constraints, but we will add it for the camera-ready version. The conclusion we plan to add in the final version is shown in the response of reviewer 8521 item 4.
>
> ***4. It would be helpful to elaborate more on the assumptions required and discuss some examples where the assumptions are satisfied/violated.***
>
> Thank you for your advice. We will include extra discussions on the assumptions we use in the later version. Here, we briefly discuss the assumptions we use.
>
> Assumption 1 is about the independence of latent variables, which is also used in  [1,2]. Using this assumption, we can model confounding between two observed nodes in a causal model by letting one latent variable affect both nodes. This is primarily a convention and does not limit the applicability of the result.
>
> Assumption 2: We discuss Assumption 2 in the reply to reviewer VZrh. Please see the item 1 of our response there.
>
> Assumption 3 is about the boundedness and Lipshictz continuity of the structural functions, which is quite standard in the literature on representation theorems of non-parametric functions with neural networks.
>
> Assumption 4 and Assumption 5 are about the latent distribution. These two assumptions enable us to approximate the latent distribution by pushing forward uniform and Gumbel variables. These assumptions may be violated if the support of the latent distribution has infinitely many connected components or if at least one component of the support is not homomorphic to the unit cube. However, we expect most natural distributions to obey these assumptions, as a violation of these assumptions can be viewed as a form of pathological distribution.
>
>
> ***5. Mention other related works.***
>
> We appreciate the reviewer’s effort to improve our paper. The provided literature is really helpful. We will add these works to our Related Work part in the camera-ready version.
>
>
> [1]Zhang, Junzhe, Elias Bareinboim, and Jin Tian. "Partial identification of counterfactual distributions." (2021).
>
> [2] Xia, Kevin, et al. "The causal-neural connection: Expressiveness, learnability, and inference." Advances in Neural Information Processing Systems 34 (2021): 10823-10836.
>
> [3] Balazadeh Meresht, Vahid, Vasilis Syrgkanis, and Rahul G. Krishnan. "Partial identification of treatment effects with implicit generative models." Advances in Neural Information Processing Systems 35 (2022): 22816-22829.
>
> [4] Hu, Yaowei, et al. "A generative adversarial framework for bounding confounded causal effects." Proceedings of the AAAI Conference on Artificial Intelligence. Vol. 35. No. 13. 2021.
>
> [5]  Guilherme Duarte, Noam Finkelstein, Dean Knox, Jonathan Mummolo, and Ilya Shpitser. An automated approach to causal inference in discrete settings, September 2021

---

> > ### Comment · Reviewer_zYjd · 2024-08-11
> >
> > Thank you for the helpful replies. Most of my concerns have been addressed and I would like to maintain my rating.

---

### Author Rebuttal · Authors · 2024-08-05

Thanks for all the reviewers' helpful feedback. We include all the figures in our response in this PDF file.

---

### Decision · Program_Chairs · 2024-09-25

**Decision:**

Accept (poster)

**Comment:**

This work extends neural causal models (NCMs) to the continuous and mixed-type variables and develops consistency results for partial identification.

Overall, all reviewers were positive about the contribution of the paper and agreed that the theoretical contribuitions of the work are novel and make NCM's more general. I agree with the assessemnt and recommend acceptance.